# Mid-old cells are a potential target for anti-aging interventions in the elderly

Young Hwa Kim[1,9], Young-Kyoung Lee[1,2,9], Soon Sang Park [1,2,3,9], So Hyun Park[4], So Yeong Eom[1,3,4], Young-Sam Lee [5], Wonhee John Lee [6], Juhee Jang[6], Daeha Seo [6], Hee Young Kang[1,7], Jin Cheol Kim[1,7], Su Bin Lim [1,2,3], Gyesoon Yoon[1,2,3], Hong Seok Kim[8], Jang-Hee Kim [1,4,10] ✉ & Tae Jun Park [1,2,3,10] ✉

The biological process of aging is thought to result in part from accumulation of senescent cells in organs. However, the present study identified a subset of fibroblasts and smooth muscle cells which are the major constituents of organ stroma neither proliferative nor senescent in tissues of the elderly, which we termed "mid-old status" cells. Upregulation of pro-inflammatory genes (*IL1B* and *SAA1*) and downregulation of anti-inflammatory genes (*SLIT2* and *CXCL12*) were detected in mid-old cells. In the stroma, SAA1 promotes development of the inflammatory microenvironment via upregulation of MMP9, which decreases the stability of epithelial cells present on the basement membrane, decreasing epithelial cell function. Remarkably, the microenvironmental change and the functional decline of mid-old cells could be reversed by a young cell-originated protein, SLIT2. Our data identify functional reversion of mid-old cells as a potential method to prevent or ameliorate aspects of aging-related tissue dysfunction.

Senescent cell accumulation in tissues is a well-known driver of organ aging and the overall aging process[1–5]. Multiple studies, including p16INK4A reporter mice, p16INK4A–3MR mice, and aged human, have consistently revealed the accumulation of senescent cells with the progression of aging[1–5]. Accumulated senescent cells play a significant role as they cause a halt in the proliferation of functional cells, ultimately resulting in organic dysfunction[2–5]. Moreover, senescent cells significantly affect the surrounding microenvironment by inducing sterile chronic inflammation through the secretion of senescence-associated secretory phenotypes (SASPs), which are known as "inflammaging" phenomena[6,7]. While it has been known that the accumulation of senescent cells in the tissues of the elderly is related to

tissue aging, it does not constitute the majority of cells within the tissue[1–5]. Moreover, it is understood that non-senescent cells within the elderly tissue still proliferate. However, the reason for the decline in organic function in the elderly as they age remains unclear. Therefore, we hypothesized that there might be a subset of cells in an intermediate stage of the cellular senescence process within the tissue, significantly impacting and ultimately leading to organic dysfunction in the elderly. Here, we propose the existence of intermediate stage cells that are neither youthful nor senescent. We termed these cells as "mid-old cells." Moreover, these cells would likely accumulate in the stromal region of the tissues for the following reasons. Epithelium, which undergo differentiation rather than self-replication, have

[1]Inflamm-Aging Translational Research Center, Ajou University Medical Center, Suwon 16499, Korea. [2]Department of Biochemistry and Molecular Biology, Ajou University School of Medicine, Suwon 16499, Korea. [3]Department of Biomedical Sciences, Ajou University Graduate School of Medicine, Suwon 16499, Korea. [4]Department of Pathology, Ajou University School of Medicine, Suwon 16499, Korea. [5]Department of New Biology, Daegu Gyeongbuk Institute of Science & Technology, Daegu 42988, Korea. [6]Department of Physics and Chemistry, Daegu Gyeongbuk Institute of Science & Technology, Daegu 42988, Korea. [7]Department of Dermatology, Ajou University School of Medicine, Suwon 16499, Korea. [8]Department of Molecular Medicine, College of Medicine, Inha University, Incheon 22212, Korea. [9]These authors contributed equally: Young Hwa Kim, Young-Kyoung Lee, Soon Sang Park. [10]These authors jointly supervised this work: Jang-Hee Kim, Tae Jun Park. ✉e-mail: drjhk@ajou.ac.kr; park64@ajou.ac.kr

shorter lifespans and undergo periodic replacement[8,9]. In contrast, mesenchymal cells, known to have longer lifespans and self-replicate more frequently than epithelial cells, are prone to accumulate in elderly tissues[10]. These stromal cells, located deeper within the tissues and protected by epithelial cells[11], might be relatively less exposed to external or environmental insults. Therefore, we focused on the fibroblasts and smooth muscle cells which are main constituent of organic mesenchymal cells and actively produce extracellular matrix components supporting functioning epithelial cells[12]. Here, we found that the major population of local organic fibroblasts or smooth muscle cells are mid-old status. Moreover, we investigated the cellular characteristics of mid-old fibroblasts and smooth muscle cells in vitro and in vivo, leading us to propose mid-old cells as a new potential target for anti-aging therapy.

## Results

### Characteristics of mid-old cells in vitro

In order to identify mid-old cells in elderly tissue in vivo, we aimed to elucidate the markers and characteristics of mid-old cells through an in vitro culture model. Primary human fibroblasts were sub-cultured in increasing passages, observing flattened morphology and decreased proliferation along with reduced Ki67 expression and EdU incorporation (Supplementary Fig. 1a). Sub-cultured fibroblasts could be categorized into three groups based on doubling time (DT) and SA-β-Gal positivity: early passage named 'young' (1-2 days DT, <1% SA-β-Gal), middle passage named 'mid-old' (5-7 days DT, <5% SA-β-Gal), and later passage named 'old' (>14 days DT, >65% SA-β-Gal). Notably, these three cell groups exhibit distinct expression patterns of conventional senescence markers, p16[INK4A], p21[Waf1], and p53. Fibroblasts in the early passage did not express those senescent markers (Fig. 1a). However, in the later passage showed higher expression of p16[INK4A] and p21[Waf1]. Moreover, heterochromatin foci formation and higher SA-β-Gal positivity were observed (Supplementary Fig. 1b–d). Intriguingly, fibroblasts in the middle passage exhibited completely different genes expression pattern; p53-p21[Waf1]-dependent cell cycle inhibition, and this pattern is confirmed in publicly available microarray data GSE41714[13] (Fig. 1a and Supplementary Fig. 1c lower panel). RNA-sequencing revealed gene expression patterns, showing mid-old cells more closely related to young cells than old cells (Fig. 1b). We analyzed 17,686 genes and identified those with gradual expression changes from young to mid-old to old cells, including genes highly expressed in mid-old cells (Supplementary Fig. 2a and Supplementary Data 1). To investigate the functional changes in fibroblasts during the aging process, gene set enrichment analysis (GSEA) was conducted using the Hallmark Gene Sets[14] (Supplementary Fig. 2b). GSEA indicated a clear senescence phenotype only in old cells based on the "FRIDMAN: Senescence Up[15]" (Fig. 1c). To assess the metabolic change with aging, we conducted GSEA on metabolism-related genes of macromolecules. The data revealed a decrease in gene expression related to DNA and mRNA metabolism in old cells compared to young and mid-old cells (Supplementary Fig. 2c–d). Interestingly, the gene expression related to peptide metabolism was found to be highest in mid-old cells. This phenomenon suggests that protein metabolism upregulation may be another characteristic of the mid-old cells (Supplementary Fig. 2e).

Next, we examined the functional capacity of fibroblasts and identify changes associated with cellular senescence. Fibroblasts can be classified into four functional categories: self-replication, extracellular matrix (ECM) formation, tissue organization/regeneration, and immune response regulation[16] (Fig. 1d and Supplementary Data 2). GSEA using gene sets related to these functions revealed decreased ECM formation and tissue organization in mid-old and old cells, which was confirmed by cDNA microarray (GSE41714)[13] (Fig. 1d and Supplementary Fig. 3a). The proliferative function showed a decreasing trend in mid-old cells, but did not reach statistical significance, while old cells exhibited decreased functional indices and proliferation capacity,

along with a significant upregulation of inflammatory response (Fig. 1e and Supplementary Fig. 3b). Conversely, although mid-old cells showed no significant differences in inflammatory response compared to young cells, the IL1β pathway was specifically upregulated, as indicated by the presence of *IL1B* in the leading edges of the inflammation-related gene set and confirmed through GSEA (Fig. 1f). Additionally, IL1β signaling pathway is significantly upregulated in mid-old cells, and SAA1, an acute response protein, was identified as a secretory factor in the IL1β pathway (Fig. 1g). In contrast, old cells exhibited distinct upregulation of TNFα and IL6 signaling pathways, separate from the IL1β-specific inflammation observed in mid-old cells, as confirmed by real-time PCR and ELISA (Fig. 1h, i). Conversely, to examine the expression of anti-inflammatory genes, real-time PCR was performed in young, mid-old, and old cells combining gene sets from GSEA (GO:0070100) and the previous study[17] (Fig. 1j), as there was no appropriate gene set available for GSEA encompassing both anti-inflammatory chemokines and cytokines. The real-time PCR revealed upregulation of most anti-inflammatory genes in young cells, including *SLIT2*, which is known to inhibit NFκB, a key regulator of inflammatory responses[18]. These findings demonstrate distinct gene expression patterns in mid-old cells compared to young and old cells in vitro (Fig. 1k).

Another important feature of mid-old cells was their capacity for functional recovery. It is common for old cells not to respond to external stimuli[19]. Several mechanisms could potentially explain these phenomena[19]. However, the most likely regulatory mechanism is disruption or distortion of signaling transduction in cells with the later passage. In our data, signal-transporting proteins, including importins[20] and caveolins[21], were downregulated in old cells, but their expression levels were maintained in mid-old cells (Fig. 1l, m). To assess the signal transduction capacity of the cells, we examined the localization of the proliferation-related transcription factor Erk1/2[22], which serves as an indicator of cellular aging signaling. In old cells, cytoplasmic p-Erk1/2 was detected, but it failed to translocate into the nucleus. Conversely, mid-old cells exhibited efficient translocation of p-Erk1/2 into the nucleus, potentially facilitating gene transcription (Fig. 1n). Real-time analysis of p-Erk1/2 nuclear translocation demonstrated that the translocation velocity was preserved in mid-old cells, comparable to that of young cells (Fig. 1o, Supplementary Fig. 3c, and Supplementary Movie 1). These findings suggest that mid-old cells retain their capacity to respond to external stimuli.

### A substantial proportion of mid-old cells in elderly organs

To investigate the presence of mid-old cells, we employed mid-old cell markers that were derived in vitro and quantified their expression through immunohistochemistry (IHC) in elderly tissues. Previous studies have shown that the proportion of p16[INK4A]-positive cells remains relatively low, even as it increases with age[1,5]. Rather, our data clearly showed marked increase of p21[Waf1] in stromal cells of elderly tissues, despite scarce expression of p21[Waf1] in epithelial cells (Fig. 2a). Furthermore, mid-old-specific inflammatory genes, *IL1β* and *SAA1*, significantly upregulated in mid-old fibroblasts based on our in vitro bulk RNA-seq data (Fig. 1g), showed robust upregulation in fibroblast-rich organs, such as the colon and lung in elderly subjects (Fig. 2b, c). Additionally, SAA1 expression was elevated not only in fibroblasts but also in smooth muscle cells of the muscular mucosa, arteries, and bronchioles in tissues from elderly subjects compared to young subjects (Fig. 2b) but absent in the epithelium (Supplementary Fig. 4). Although not all gene expressions displayed the same patterns as fibroblasts, the expression of mid-old cell markers was consistently observed in an in vitro culture model of smooth muscle cells (Supplementary Fig. 5). We confirmed these findings using the mid-old cell markers (SAA1, IL1β, and p21[Waf1]) obtained from in vitro studies, as well as markers for fibroblasts (vimentin) and smooth muscle cells (smoothelin). Immunofluorescent (IF) staining revealed that SAA1 and

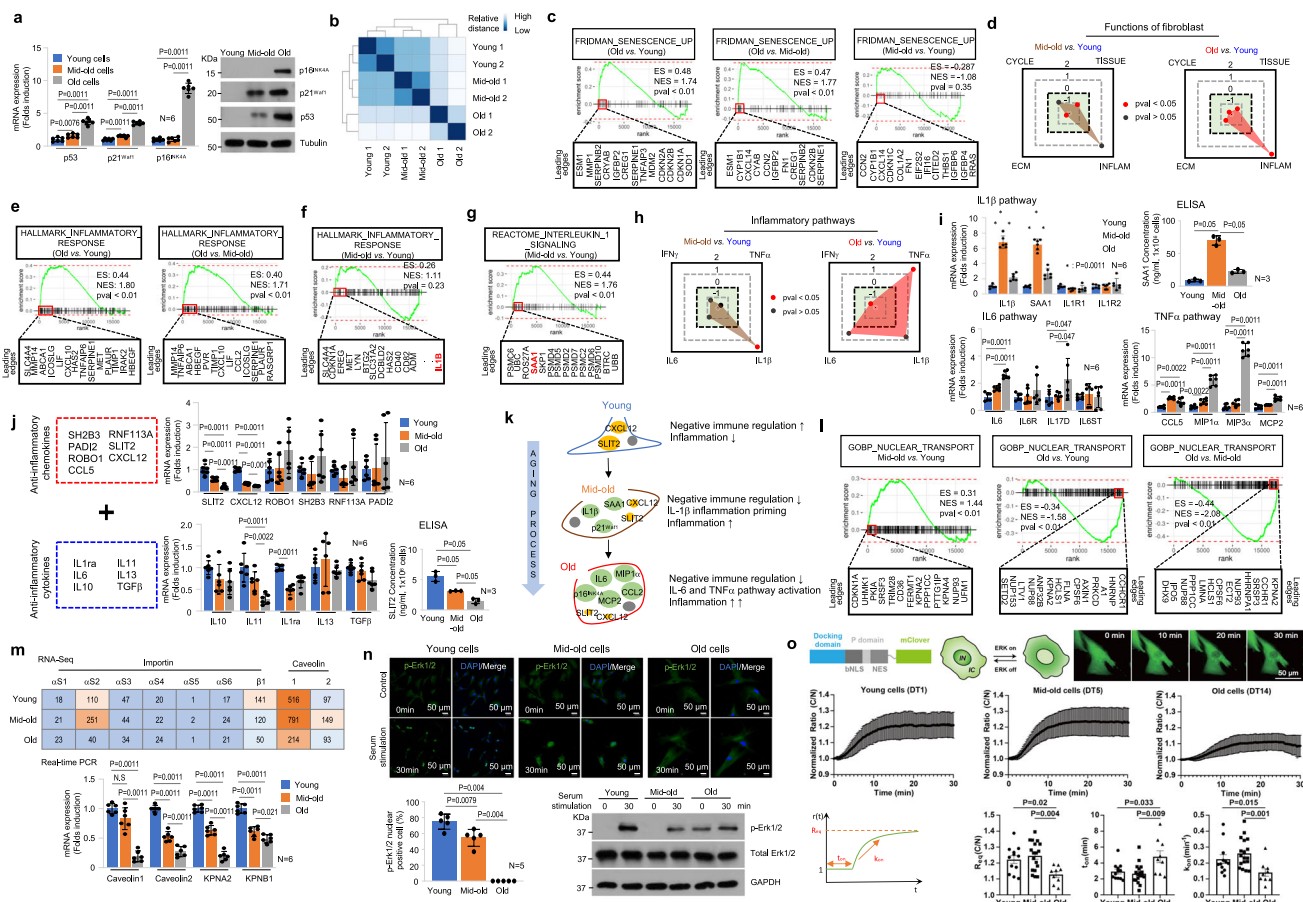

**Fig. 1 | Characteristics of young, mid-old, and old cells in vitro. a** mRNA (left panel) and protein (right panel) levels of representative senescence markers (p53, p21^Waf1, and p16^INK4A) in young, mid-old, and old cells are shown. **b** Relative distances represented as VST using count data from young, mid-old, and old samples are shown (*n* = 2, each). **c** GSEA using the "FRIDMAN: senescence up" gene set in old *vs.* young (left panel), old *vs.* mid-old (middle panel), and mid-old *vs.* young cells (right panel) (adjp = 0.0029, 0.016, and 0.53, respectively). **d** NES calculated from GSEA using gene sets representing fibroblast functions are presented. The gene sets include "CYCLE" representing self-replicative ability, "ECM" representing ECM productive ability, "TISSUE" representing tissue regeneration ability, and "INFLAM" representing inflammation-regulating ability. Used gene sets and related statistics for this analysis can be found in Supplementary Data 2. **e, f** GSEA of old *vs.* young, old *vs.* mid-old, and mid-old *vs.* young using the "HALLMARK: inflammatory response" gene set is shown (adjp = 0.0075, 0.0087 for (**e**), and 0.31 for (**f**), respectively). **g** GSEA of mid-old *vs.* young using the "REACTOME: interleukin 1 signaling pathway" gene set is shown (adjp = 0.018). **h** NES calculated from GSEA using gene sets representing the inflammatory pathway are shown. Used gene sets and related statistics in this analysis can be found in Supplementary Data 2. **i** The mRNA levels of multiple inflammatory pathways related genes and SAA1 protein level were analyzed by real time PCR and ELISA analysis. **j** A list of analyzed genes that act as anti-inflammatory chemokines and cytokines used in this study is shown (left panel). The mRNA levels of the listed genes are shown (middle panel). ELISA analysis of SLIT2 in young, mid-old, and old cells (right lower panel). **k** A schematic image illustrating the characteristics of young, mid-old, and old cells with

representative gene expression. **l** GSEA using the "GOBP: nuclear transport" gene set in mid-old *vs.* young (left panel), old vs. young (middle panel) and old *vs.* mid-old (right panel) cells is shown (adjp = 0.085, 0.046, and 0.042, respectively). **m** The mRNA expression of nuclear transport-related genes in RNA-seq (upper panel). The figures in the box represent FPKM counts. S1-6 represents the subunits of the importin α. Real-time PCR results of nuclear transport-related genes are shown (lower panel). **n** Erk1/2 phosphorylation and nuclear translocation were analyzed after serum stimulation for 30 min (upper panel). Quantification of the number of cells with p-Erk1/2 nuclear translocation (lower left panel). p-Erk1/2 level was analyzed by western blot (lower right panel). **o** Investigation of p-Erk1/2 translocation dynamics. Schematic illustration of Erk1/2-KTR-mClover system and kinetic parameters (left upper and left lower). Young (*n* = 4), mid-old (*n* = 6), and old (*n* = 3) cells were assessed with the system, respectively. Three independent experiments were repeated. Representative images of Erk1/2-KTR-mClover translocation (right upper). Erk1/2-KTR-mClover translocation C/N curves for each cell type (middle). Cells were starved for 24 h with serum free media prior to experiments. Fluorescent images were acquired every 15 sec for 30 min after serum stimulation. Comparison of translocation kinetic parameters between cell types (lower panel). The *p* value in (**a**), (**i**), (**j**), and (**m–o**) was calculated using the one-tailed Mann–Whitney *U*-test. The *p* values in (**c–h**), and (**i**) are obtained using GSEA statistics in 'fgsea' package. ES enrichment score, NES normalized enrichment score, adjp = adjusted *p* value. The graph in (**a**), (**i**), (**j**), and (**m–o**) was represented as mean ± SD. A thousand times of permutations were performed for adjustment in (**c–h**) and (**l**).

---

IL1β were expressed by mesenchymal cells, rather than epithelial cells (Supplementary Fig. 6). Furthermore, serum SAA1 level was increased in the elderly even though the tested human serum was from patients with benign or malignant disease (Supplementary Fig. 7a). Although SAA1 is involved in the acute inflammatory response[23], and is secreted by hepatocytes mainly after micro-organism infection[24], its expression was not increased in hepatocytes from elderly subjects (Supplementary Fig. 7b). In mouse experiments, SAA1 expression was increased in tissues of 18-month-old mice compared to 4-month-old mice.

However, its expression pattern was slightly different from that of human tissue; fibroblasts and colon epithelial cells, rather than smooth muscle cells, strongly expressed SAA1 (Supplementary Fig. 7c). Furthermore, serum SAA1 levels were elevated in old-aged mice (Supplementary Fig. 7d).

Cumulatively, the specific inflammation associated with mid-old cells, characterized by increased SAA1 and IL1β expression, was observed in tissues from elderly subjects. Subsequently, we analyzed anti-inflammatory factors and found that SLIT2 expression was

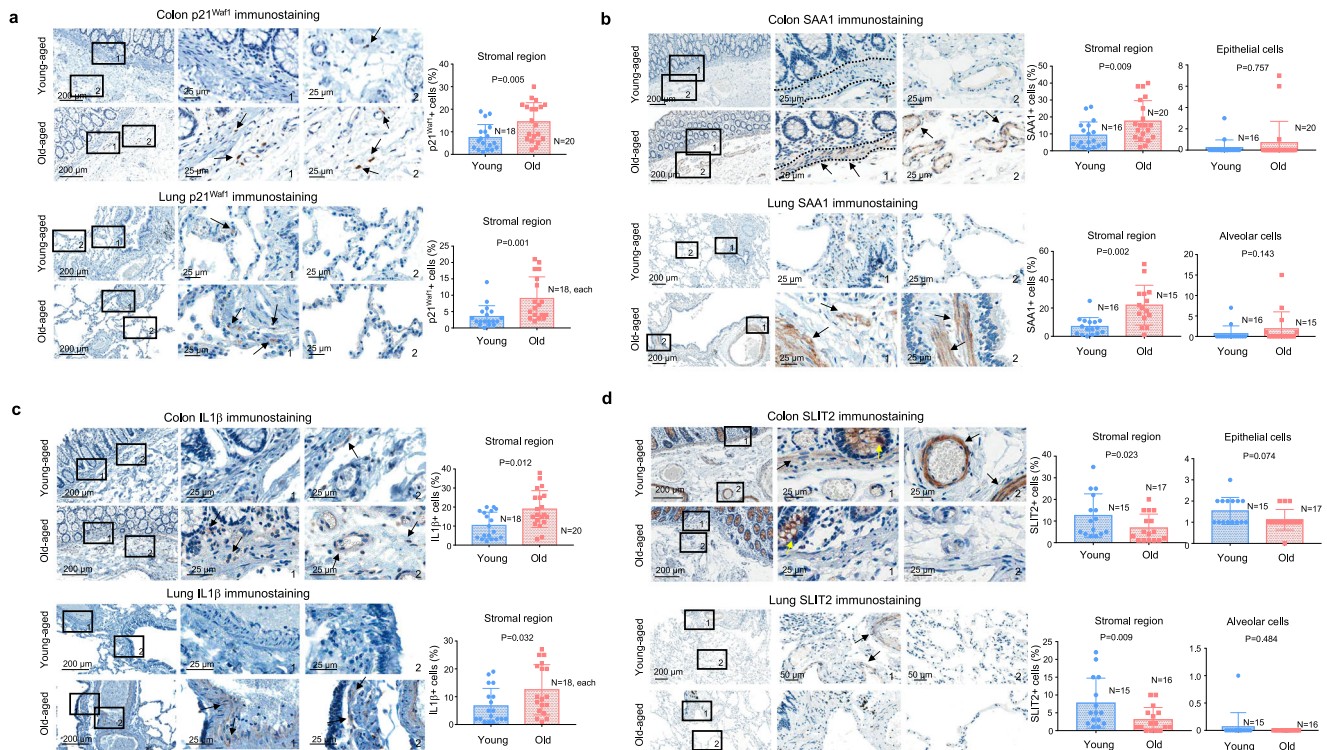

**Fig. 2 | Mid-old specific gene expression in old-aged tissues. a–d** Representative images of p21[Waf1], SAA1, IL1β and SLIT2 IHC analysis in young and old-aged human colon and lung tissues are shown. "1" and "2" indicate high-magnification views of the original figure. Image "1" primarily depicts the epithelium and smooth muscle region, while image "2" depicts a stromal-rich region. Black arrows indicate stained cells in the stromal region. Yellow arrow in (**d**) indicate SLIT2-positive cells in epithelial region. Corresponding quantification dot plots are shown in the right panel of each figure. IHC results were presented as the percentage of positive cells in the stromal or epithelial region. The *p* value was calculated using the one-tailed Mann–Whitney *U*-test. The graphs in (**b**) (right tables) and (**d**) (lower right table) were represented as mean + SD. The graphs in (**a**), (**b**) (left tables), (**c**), and (**d**) (left tables) were represented as mean ± SD.

significantly downregulated in tissues from elderly subjects and aged mice (Fig. 2d and Supplementary Fig. 8). Previously, we demonstrated that CXCL12 functions as an anti-inflammatory cytokine by reducing CD8+ T-cell infiltration[25], and CXCL12 can be an anti-aging factor in the skin[26]. Interestingly, in current study, we also observed downregulation of CXCL12 in elderly tissues (Supplementary Fig. 9). These data suggest that elderly tissues undergo a shift towards a chronically inflammatory state, characterized by the upregulation of SAA1 and IL1β and the downregulation of SLIT2 and CXCL12.

## The functional role of SAA1 in elderly tissues

Based on our in vitro and in vivo data, SAA1 was found to be highly upregulated in mid-old cells, closely associated with IL1β signaling (Figs. 1g and 2b). We analyzed the effect of SAA1 on the types of cells comprising the analyzed tissues, including epithelial cells, fibroblasts, and smooth muscle cells (Fig. 3a). Well-known aging-related symptoms of the colon and lungs, such as distorted absorption and peristalsis, are exacerbated by impaired epithelial cell and smooth muscle function[27,28]. Therefore, constipation, dry cough, and frequent infection are common ailments in the elderly[27,28]. SAA1 is well-known as an inducer of pro-inflammatory cytokines[23]. However, in our studies, recombinant human SAA1 (rhSAA1) increased expression of MMPs only, especially *MMP9*, in fibroblasts and muscular mucosa smooth muscle cells (Fig. 3b, c) via NFκB signaling (Supplementary Fig. 10a). Thus, the expression of MMP9 was considered a characteristic of mid-old and old cells (Supplementary Fig. 10b). These findings suggest that SAA1 may contribute to the development of aging-related microenvironments by promoting the degradation of ECM components. Indeed, MMP9 protein expression was robustly increased in elderly tissues that expressed SAA1

(Fig. 3d). Furthermore, SAA1 has been known as skeletal muscle atrophy-inducing factor[29]. Indeed, muscle atrophy-related genes, known as 'Atrogenes[30]', including *FBXO30*, *FBXO32*, and *TRIM63*, were significantly upregulated with the decrease in cell proliferation in rhSAA1-treated primary smooth muscle cells (Fig. 3e), while expression of muscle contraction-related genes was unchanged (Supplementary Fig. S11). The thickness of the muscular mucosa, which exhibited high expression of SAA1 in the colon, was significantly reduced in tissues from elderly subjects (Fig. 3f). Although we attempted to assess the thickness of the bronchial smooth muscle layer, irregularities in histological sectioning hindered precise measurements in lung tissue. We then investigated the impact of SAA1 on epithelial cells, specifically their function in ionic exchange and water reabsorption capacity. Interestingly, SAA1 did not alter the expression of genes associated with these functions in colonic epithelial cells (Supplementary Fig. 12). These findings suggest that SAA1 directly affected mesenchymal cells, such as fibroblasts and smooth muscle cells, but not epithelial cells. Based on a previous study showing that SAA1 stimulates the upregulation of MMP9[31], and since MMP9 can affect epithelial cell function by modifying the ECM and basement membrane (BM)[32], there is a high possibility that increased expression of stromal SAA1 indirectly affects epithelial cell function by degrading the BM. Maintaining the integrity of the BM is crucial for proper epithelial cell function, and remnants of degraded ECM components, known as 'Matrikines[33]', can also influence epithelial cell gene expression. Therefore, we evaluated epithelial cell function based on BM stability. Although the mRNA levels of type IV collagen, a major component of the BM, did not show significant difference between elderly and young-aged tissues (GSE178341[34]), the protein level of type IV collagen was

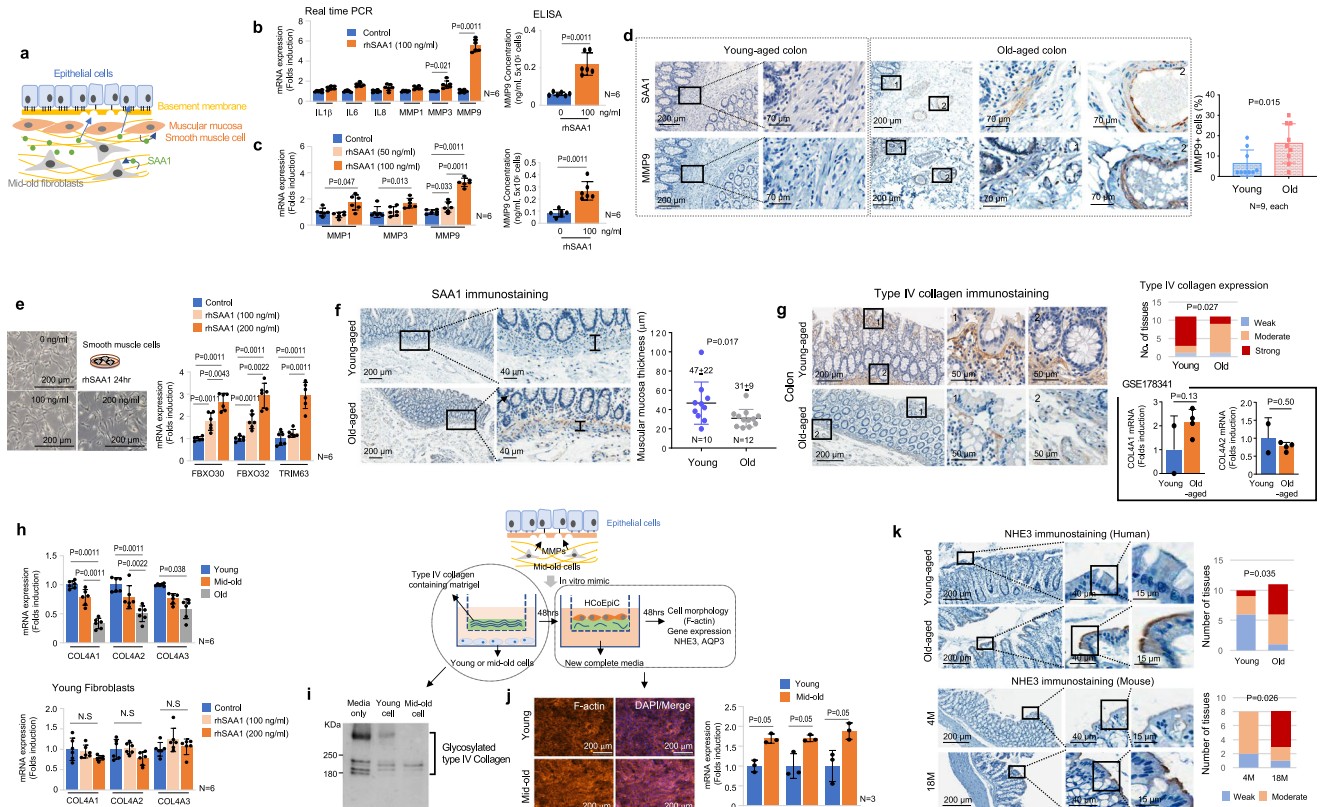

**Fig. 3 | The Effect of SAA1 on old-aged tissues. a** Schematic drawing of the effect of SAA1 on tissue. **b** Young fibroblasts or **c** smooth muscle cells (PASMCs) were treated with rhSAA1 for 24 h and SASP expression was analyzed using real-time PCR (left panel). MMP9 protein expression was analyzed by ELISA (right panel). **d** Serially dissected human colon tissues were subjected to IHC analysis for SAA1 and MMP9. Corresponding quantification data is shown in the right panel. IHC results were presented as the percentage of positive cells in the stromal region. **e** PASMCs were treated with rhSAA1 at the indicated concentration for 24 h and real-time PCR was performed using primers for atrophy-related genes. **f** IHC analysis for SAA1 in colon tissue from young and elderly subjects, and muscular mucosa thickness was measured. **g** IHC analysis for type IV collagen of colon tissues from young and elderly subjects is shown. "1" and "2" indicate high-magnification views of the original figure (left panel). Each protein expression was presented as weak, moderate, and strong (right upper panel). mRNA expression level of *COL4A1* and *COL4A2* in fibroblasts/smooth muscle cells was analyzed between young and old subjects in scRNA-seq data set (GSE178341)[34] (right lower panel). **h** *COL4A1*,

*COL4A2* and *COL4A3* gene-expression were analyzed by real-time PCR in young, mid-old, and old fibroblasts (upper panel). Young fibroblasts were treated rhSAA1 for 24 h, and mRNA expression level of type IV collagen was analyzed (lower panel). **i** In vitro Matrigel degradation assay results are shown. Matrigel-coated membrane was co-cultured with young or mid-old fibroblasts for 2 days. Degradation of type IV collagen was analyzed using western blotting. Three independent experiments were performed to get similar results. **j** Cell morphology was analyzed using F-actin staining (left panel) and functional marker expression was measured using real-time PCR analysis (right panel) in HCoEpiC. **k** NHE3 immunostaining in human (upper panel) and mouse (lower panel) colon tissue. NHE3 was stained in the apical region of the colon epithelium. Each protein expression status indicates weak, moderate, and strong. The *p* values of figure (**b**–**f**), (**g**) (lower graph), (**h**), and (**j**) are obtained from the one-tailed Mann–Whitney *U* test. The *p* values of figure (**g**) (upper graph) and (**k**) are obtained from Chi-square test. The graph in (**b**–**h**), and (**j**) was represented as mean ± SD.

downregulated in elderly tissues (Fig. 3g). Type IV collagen mRNA expression was downregulated only in old fibroblasts, and treating with rhSAA1 did not decrease type IV collagen mRNA levels in young fibroblasts (Fig. 3h). Hence, it is considered that the downregulation of type IV collagen in the BM of elderly organs is likely due to its degradation by MMP9 rather than reduced production of type IV collagen. To test this hypothesis in vitro, we cultured young or mid-old fibroblasts in Matrigel containing type IV collagen and found that collagen IV degradation was more pronounced in the presence of mid-old fibroblasts (Fig. 3i). Furthermore, co-culture of HCoEpiC with mid-old fibroblasts exhibited upregulation of *NHE3* and *AQP3* expression, which are functional dysregulation marker genes for the large intestine[35,36] (Fig. 3j). Intriguingly, MMP9 knockdown in mid-old fibroblasts rescued the upregulation of *NHE3* and *AQP3* expression (Supplementary Fig. 13a). When we co-culture of HCoEpiC with old fibroblasts exhibited upregulation of *NHE3* expression only (Supplementary Fig. 13b). We further confirmed it using SW480 colon cancer cells (Supplementary Fig. 13c). Indeed, NHE3 expression was increased in colonic epithelial cells of both elderly human

and aged-mice relative to young counterparts (Fig. 3k and Supplementary Fig. 13d).

Analogous to the findings in the colon, we observed upregulation of MMP9 expression and downregulation of type IV collagen in lung tissue derived from elderly subjects (Fig. 4a, b). To maintain proper gas exchange, adequate alveolar structure and fluid transportation should be maintained[37]. Oxygen and carbon dioxide are mutually exchanged in type I alveolar cells in dose-dependent manner[38], and sufficient alveolus expansion and adequate water exchange by type II alveolar cells are required for efficient gas exchange[39] (Fig. 4c). Like skin fibroblasts, IMR90 lung fibroblasts demonstrated increased expression of SAA1 and p21[Waf1] upon transitioning to the mid-old stage (Fig. 4d). Notably, it was observed that MMP9 expression was enhanced by rhSAA1. However, in alveolar cells, rhSAA1 treatment did not influence expression of genes related to surfactants and water reabsorption, including *SFTPD, SCNN1A, AQP3*, and *ATP1A1* (Fig. 4e). However, when we co-cultured primary alveolar cells with mid-old IMR90 on the Matrigel, we observed an increase in the expression of the *SCNN1A* (Fig. 4f). It is also confirmed in elderly's human and mouse

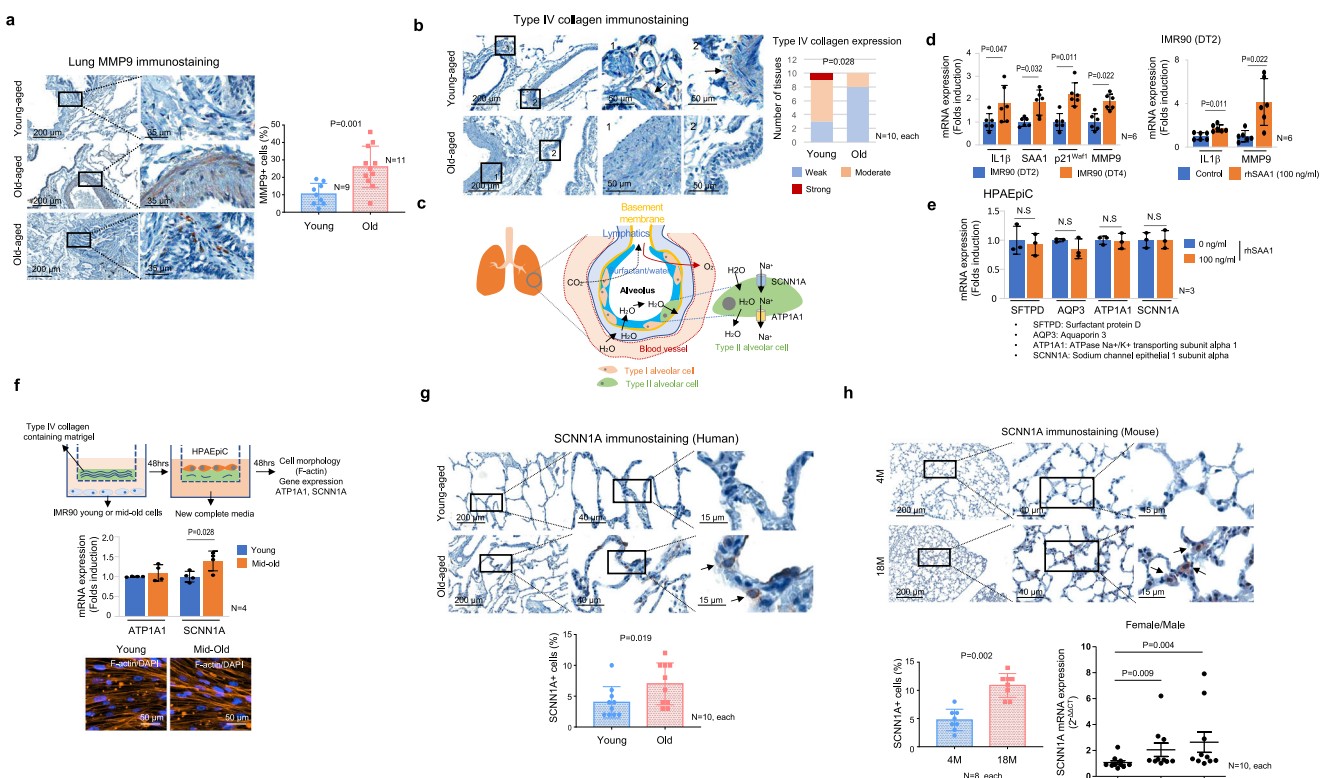

**Fig. 4 | The Functional role of BM degradation in the lung tissue of the elderly.** **a** IHC analysis of human lung tissues were subjected for MMP9. Quantification data of IHC results is shown in the right panel. **b** IHC analysis of Type IV collagen was performed on lung tissue obtained from elderly and young subjects. High-magnification views of the original figure are represented as "1" and "2". **c** The schematic image illustrates the process of water exchange in the lung. Alveolar water absorption is regulated by the uptake of Na⁺ ions through the apical SCNN1A channel and the basolateral ATP1A pump in type II alveolar cells. **d** IMR90 lung fibroblasts were sub-cultured to generate mid-old cells (DT 4 days). The mRNA expression of mid-old cell markers in young and mid-old IMR90 cells is shown in the left panel. Young IMR90 cells were treated with rhSAA1 for 24 h, and the mRNA expression of IL1β and MMP9 was analyzed in the right panel. **e** HPAEpiC were treated with rhSAA1 (100 ng/ml) for 24 h, and the expression of *SFTPD*, *AQP3*, *ATP1A1*, and *SCNN1A* was analyzed using real-time PCR. **f** HPAEpiC were cultured on the Matrigel-coated membrane for 2 days, and the expression levels of *ATP1A1* and *SCNN1A* were analyzed using real-time PCR. **g** IHC analysis of SCNN1A in human lung tissue is shown. Arrows indicate SCNN1A-expressing cells. Quantification data of IHC results is shown in the lower panel. **h** IHC analysis of SCNN1A in mouse lung tissue is shown. Arrows indicate SCNN1A-positive cells. Quantification data of IHC results is shown in the left lower panel. The mRNA expression level of *SCNN1A* was analyzed in the mouse lung tissue is shown (right lower panel). IHC results were presented as the percentage of positive cells in the stromal or epithelial region. The *p* values of (**a**), (**d–g**), and (**h**) are obtained from the one-tailed Mann–Whitney *U* test. The *p* value of (**b**) is obtained using Chi-square test. The graphs in (**a**) and **d–h**) were represented as mean ± SD.

---

lung tissues (Fig. 4g, h). These data suggest SAA1 may affect to deterioration of water circulation in the alveolus.

## Young cells' origin, SLIT2, inhibits the senescence program

The aging process at the whole-organism level is age-dependent and exhibits a step-like pattern rather than a linear one[40] (Fig. 5a). At certain ages, such as around 35-, 63-, and 78-year-old, the senescence program in the human body accelerates rapidly[40], but the underlying mechanism is poorly understood. To address this question, we proposed that young cells may prevent the onset of senescence in adjacent mid-old cells. However, if the senescence-driving factors outweigh the inhibitory ability of young cells, a large population of mid-old cells would rapidly enter the senescence program (Fig. 5a). This effect is likely mediated by multiple factors released from young cells. Co-culturing old cells with young cells did not elicit a response from the old cells (Supplementary Fig. 14a). However, the proliferative ability of mid-old cells significantly increased when co-cultured with young cells (Fig. 5b, c). Similar results were observed when conditioned media (CM) from young to mid-old fibroblasts were used (Supplementary Fig. 14b). To exclude the possibility of migration from the upper chamber, we introduced a GFP system in mid-old fibroblasts, demonstrating that the proliferation of GFP-positive mid-old fibroblasts increased (Supplementary Fig. 14c). These data suggest that factors secreted from

young cells has effect of reverse aging not only the self-replication ability but also other functions of mid-old fibroblasts. RNA-sequencing revealed a significant increase in all functional criteria of fibroblasts, except for the inflammatory response (Fig. 5d, e). Although the inflammatory response did not reach statistical significance (*p* = 0.056), mid-old cells co-cultured with young cells showed a decreasing tendency of inflammation, particularly in the expression of IL1β (Fig. 5e). Real-time PCR and ELISA validation confirmed the functional recovery (Fig. 5f), indicating the presence of potential factors released from young cells that reverse the aging phenotype of mid-old cells, termed "Juvenile-Associated Secretory Phenotypes (JASPs)."

To identify the factors responsible for restoring the proliferative capacity of mid-old cells, we analyzed the composition of CM from young fibroblasts, including lipids, metabolites, proteins, and long non-coding RNAs (Lnc-RNAs). None of the detected small molecules were known p53-p21^Waf1 pathway inhibitors, including lipids and metabolites (Supplementary Fig. 15). However, Lnc-RNAs, including SBLC[41], SAL-RNA1[42], and ROR[43], known to inhibit p53, were expressed abundantly in young fibroblasts, but only SBLC identified in exosomes derived from young fibroblasts (Fig. 5g). Overexpression of SBLC in mid-old fibroblasts decreased the protein level of p21^Waf1 by regulating MDM2[41] (Fig. 5h), but it did not affect the expression of inflammatory

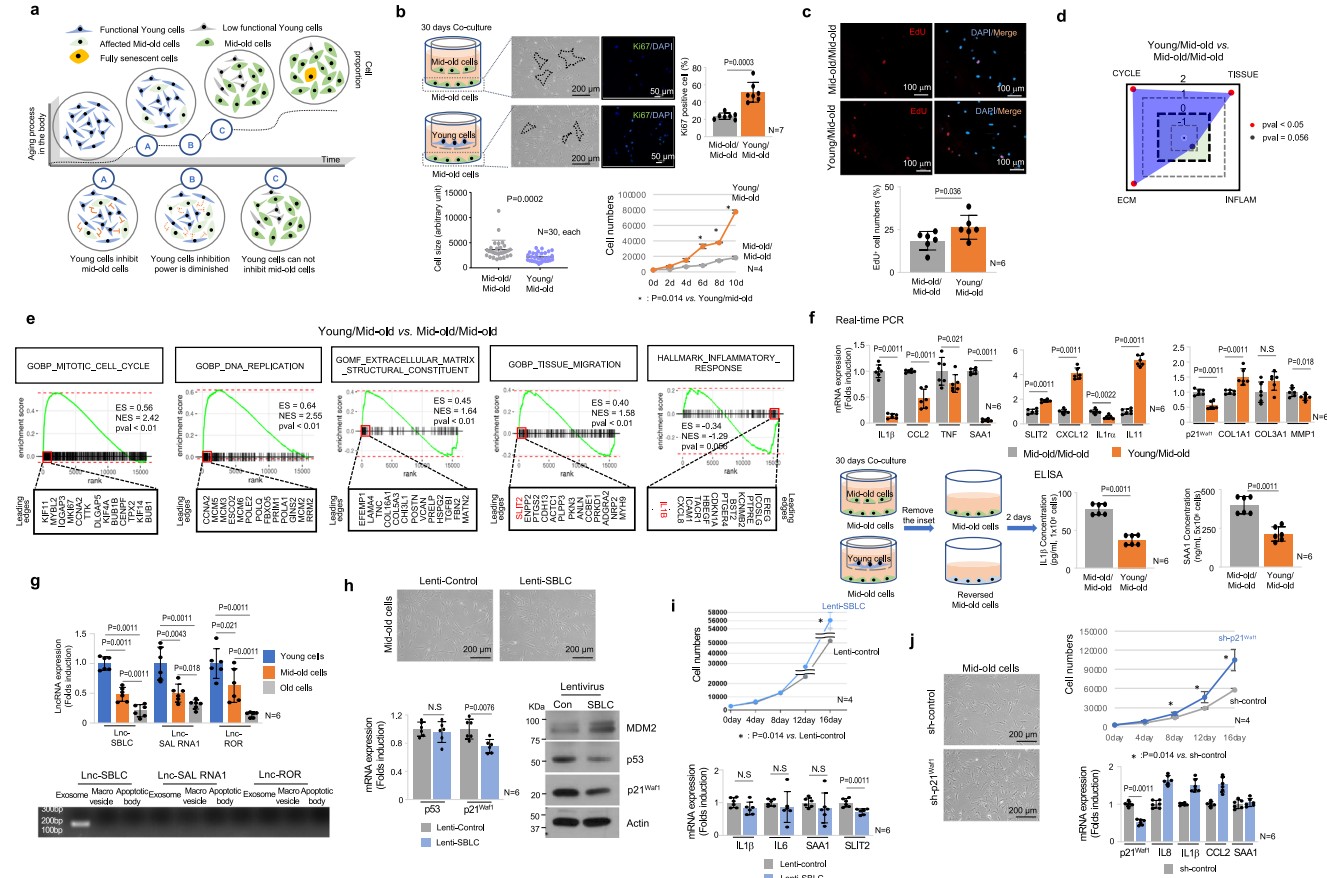

**Fig. 5 | Young cell-driven factors reverse the phenotype of mid-old cells. a** A schematic drawing of the stromal cell population according to the aging process is shown. **b** Mid-old cells are co-cultured with young or mid-old cells for 30 days, respectively. IF staining for Ki67 was performed (left upper panel). Quantification data for Ki67 is shown in the right upper panel. The cell size was compared and quantified (left lower panel). After 30 days of co-culture, cell growth rates were measured over a period of 10 days (right lower panel). **c** EdU incorporation assay. Mid-old cells co-cultured with young or mid-old cells for 30 days and then analyzed EdU incorporation. Quantification data is shown in the lower panel. **d** GSEA was performed in mid-old cells co-cultured with young and mid-old cells, respectively ($n = 2$, each). Functional gene sets of fibroblasts including "CYCLE" representing self-replicative ability, "ECM" representing ECM productive ability, "TISSUE" representing tissue regeneration ability, and "INFLAM" representing inflammation-regulating ability were analyzed. Used gene sets and related statistics for this analysis can be found in Supplementary Data 2. **e** GSEA was performed for "GOBP: mitotic cell cycle", "GOBP: DNA replication", "GOMF: extracellular matrix structural constituent", "GOBP: tissue migration", and "Hallmark: inflammatory response" (adjp = 0.038, 0.038, 0.038, 0.038, and 0.11). The $p$ value and adjp are obtained using GSEA statistics in 'fgsea' package. ES: enrichment score; NES: normalized enrichment score; adjp: adjusted $p$ value. **f** The mRNA expression of inflammatory, anti-inflammatory genes, and ECM-related molecules in mid-old cells co-cultured with young or mid-old cells (upper panel). ELISA analysis of IL1β and SAA1 in mid-old cells co-cultured with young or mid-old cells (lower panel). **g** The expression level of Lnc-RNAs was measured using real-time PCR in young, mid-old, and old cells (upper panel). Lnc-RNAs were isolated from multiple cellular compartments and shown using conventional real time-PCR (lower panel). **h** Mid-old cells were infected with lentivirus harboring Lnc-SBLC for 4 days. Cell morphology (upper panel) was observed and p53 and p21$^{Waf1}$ expression were analyzed using real-time PCR (left lower panel) and western blot analysis (right lower panel). **i** The upper panel shows the cell growth rate, which was measured by counting the cells every 4 days. The lower panel displays the expression levels of inflammatory genes, which were measured using real-time PCR. **j** Mid-old cells were infected with lentiviruses carrying sh-p21$^{Waf1}$ for 4 days. Cell morphology was examined and depicted in the left panel, while the cell growth rate was monitored by counting the cells every 4 days (right upper panel). The expression of various inflammatory genes was assessed using real-time PCR (right lower panel). The $p$ values of (**b**) (right upper and lower), (**c**), and (**f**), (**j**) are obtained from the one-tailed Mann–Whitney $U$ test. The $p$ value of (**b**) (left lower) is obtained from two-tailed Student's $t$ test. The $p$ values of (**e**) (left lower) are obtained using GSEA statistics in 'fgsea' package in R. The graphs in (**b**), (**c**), and (**f**–**j**) were represented as mean ± SD. A thousand times of permutations were performed for GSEA adjustment.

genes (Fig. 5i). Downregulation of p21$^{Waf1}$ by shRNA increased fibroblast proliferation but did not improve the inflammaging phenomenon (Fig. 5j). These data indicate that increased proliferative ability alone is not sufficient for reversal of aging phenotype, suggesting the involvement of other primary mechanisms. Previously, we proposed CXCL12 as a juvenile factor, demonstrating its anti-inflammatory effect by inhibiting T-cell infiltration[25] and promoting skin whitening[26]. However, CXCL12 itself did not affect the expression of inflammatory genes in mid-old fibroblasts (Supplementary Fig. 16). In contrast, SLIT2, identified as a representative young cell secretory factor in our in vitro studies, is a known anti-inflammatory protein that progressively decreases as cells become senescent. To assess the impact of

young cell driven SLIT2 on mid-old cells, we treated mid-old fibroblasts with recombinant human SLIT2 protein (rhSLIT2). Treatment with rhSLIT2 resulted in decreased expression of inflammatory genes and proteins, including SAA1 and IL1β, through inhibition of Pyk2-NFκB signaling[18] (Fig. 6a, b). SLIT2 overexpression also showed similar effects (Fig. 6c). Surprisingly, long-term treatment with rhSLIT2 protein restored the morphology of mid-old fibroblasts, causing them to revert to a small and spindle-shaped morphology reminiscent of young cells. To exclude the possibility of proliferation in young cells within the mid-old cell population in response to rhSLIT2, we treated young cells with rhSLIT2 and observed that young cells did not undergo increased proliferation (Supplementary Fig. 17a). Additionally, the

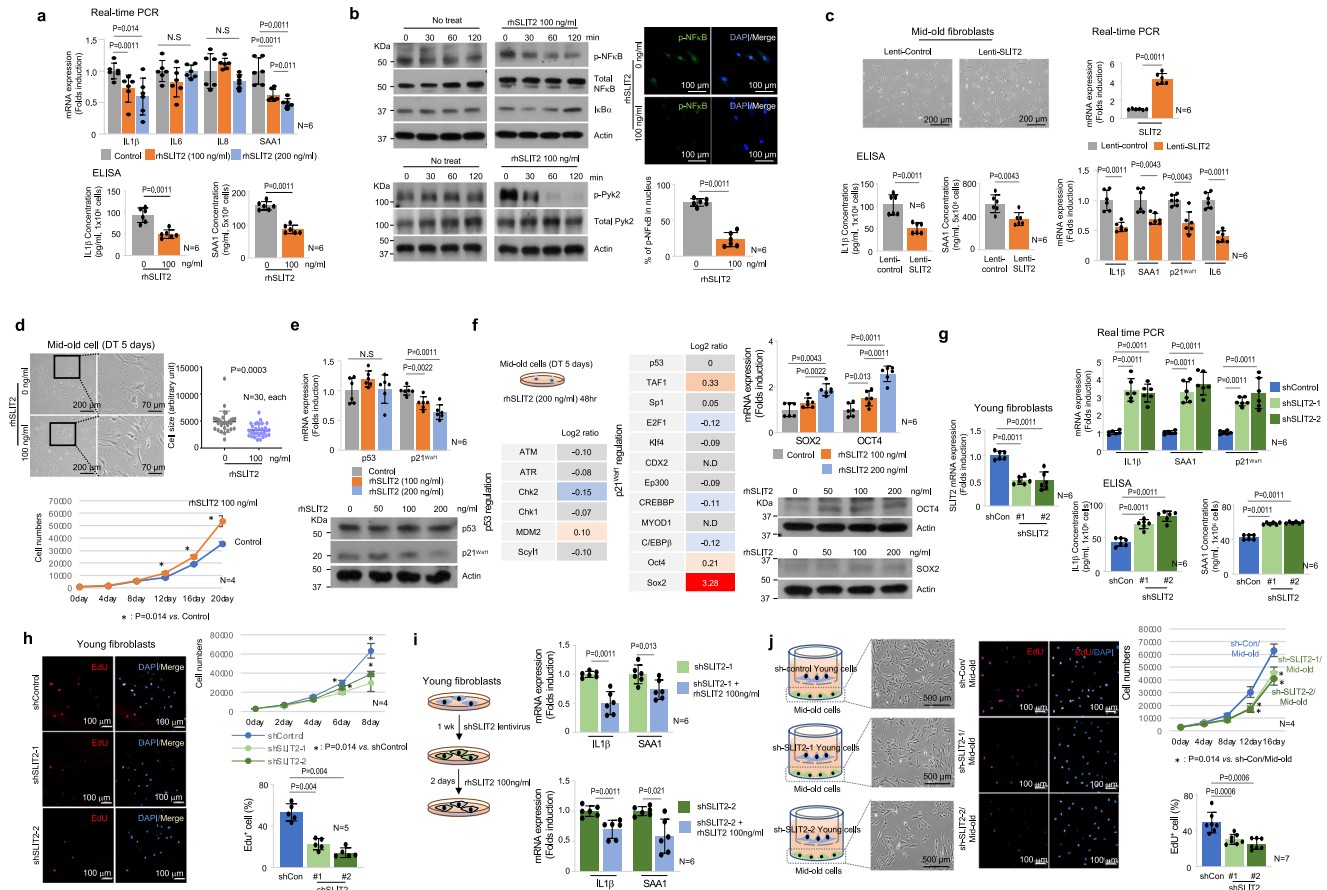

**Fig. 6 | SLIT2 has an impact on mid-old cells. a** Mid-old cells were treated for 24 h with rhSLIT2, and inflammatory gene expression was analyzed using real-time PCR (upper panel). IL1β and SAA1 protein level was analyzed by ELISA in rhSLIT2-treated mid-old cells (lower panel). **b** Mid-old cells were treated with rhSLIT2 for the indicated times and analyzed for Pyk2-NFκB signaling by western blot analysis (left panel). The number of nuclear p-NFκB-positive cells was quantified in mid-old cells using IF staining (right panel). **c** SLIT2-expressing lentivirus infected into mid-old fibroblasts, and inflammation related genes expression including SAA1 and IL1β were analyzed using real-time PCR (right lower panel). IL1β and SAA1 protein level was analyzed by ELISA in SLIT2-overexpressing mid-old cells (left lower panel). **d** The morphology and cell size of mid-old cells were analyzed after treating rhSLIT2 for 20 days. Cell growth rate was analyzed every 4 days for 20 days by counting the number of cells. **e** The expression of p53 and p21^Waf1 were measured in mid-old cells after administration of rhSLIT2 at the indicated concentration for 2 days using real-time PCR (upper panel) and western blot analysis (lower panel). **f** The changes in gene expression of p53 and p21^Waf1-regulating genes were evaluated after treating cells with rhSLIT2 using RNA-seq FPKM count (left panel). The mRNA expression of *SOX2* and *OCT4* were analyzed using real-time PCR (right

upper panel) and western blot analysis (right lower panel) in rhSLIT2-treated mid-old cells. **g** Young cells were infected with shSLIT2 lentivirus and stabilized for 2 days, and subsequently cultured for 8 days. The mRNA expression level of SAA1 and IL1β was analyzed using real-time PCR (upper panel), while the protein expression level was assessed using ELISA (lower right panel). **h** The cell growth rate was evaluated by counting the number of cells every 2 days over a period of 8 days (right upper panel). EdU incorporation assay. Same experiment scheme as (**g**) was applied for EdU incorporation study (left panel). **i** SLIT2 was knocked down in young cells infecting shSLIT2 lentivirus. The knockdown was then rescued by treating the cells with rhSLIT2 protein at a concentration of 100 ng/ml for 2 days. SAA1 and IL1β mRNA expression levels were analyzed with real-time PCR. **j** Mid-old cells were co-cultured with sh-control or sh-SLIT2 lentivirus infected young cells for 30 days. At the experimental endpoint, the morphology of mid-old fibroblasts was observed (left panel), and EdU incorporation assay (middle panel) and cell proliferation (right upper panel) was measured. The *p* values of (**a–c**), (**d**) (lower panel), and (**e–j**) are obtained from one-tailed Mann–Whitney *U* test. The *p* value of (**d**) (right upper panel) is obtained from two-tailed Student's *t* test. The graphs in (**a–j**) were represented as mean ± SD.

proliferative capacity of mid-old fibroblasts was recovered, accompanied by decreased levels of p21^Waf1 mRNA and protein in a p53-independent manner (Fig. 6d, e). Notably, the expression of p21^Waf1 transcriptional regulating factors did not change significantly, except for *SOX2* and *OCT4*, well-known stem cell factors[44]. Treatment with rhSLIT2 in mid-old cells resulted in upregulated expression of *SOX2* and *OCT4* (Fig. 6f). The expression of known SLIT2 receptors, including *ROBO2* and *ROBO4*, was not significantly downregulated in mid-old cells (Supplementary Fig. 17b), suggesting that the effect of SLIT2 is not solely due to decreased receptor activity. To further investigate the role of SLIT2, we downregulated SLIT2 expression using shRNA in young fibroblasts. SLIT2 downregulation induced the expression of inflammation-related genes and proteins, including IL1β and SAA1, and decreased the proliferation ability of the cells (Fig. 6g, h). These

findings suggest that a decrease in SLIT2 itself could promote senescence process. We further confirmed the effect of SLIT2 by adding back rhSLIT2 protein to shRNA-infected young cells, which resulted in the downregulation of the expression of upregulated inflammatory genes (Fig. 6i). Finally, we applied SLIT2 downregulation in a co-culture system and found that it abolished the reversal of aging phenotype effect on mid-old cells (Fig. 6j).

## Anti-aging effects of SLIT2 on aged mouse

Above our findings suggest that stromal cells in organs drive the senescence program at a whole-organism level, and that the Lnc-RNA-SBLC and SLIT2 protein derived from young cells function as anti-aging factors for mid-old fibroblasts. We then applied it to an aged-mouse model. However, SBLC could not be applied due to non-

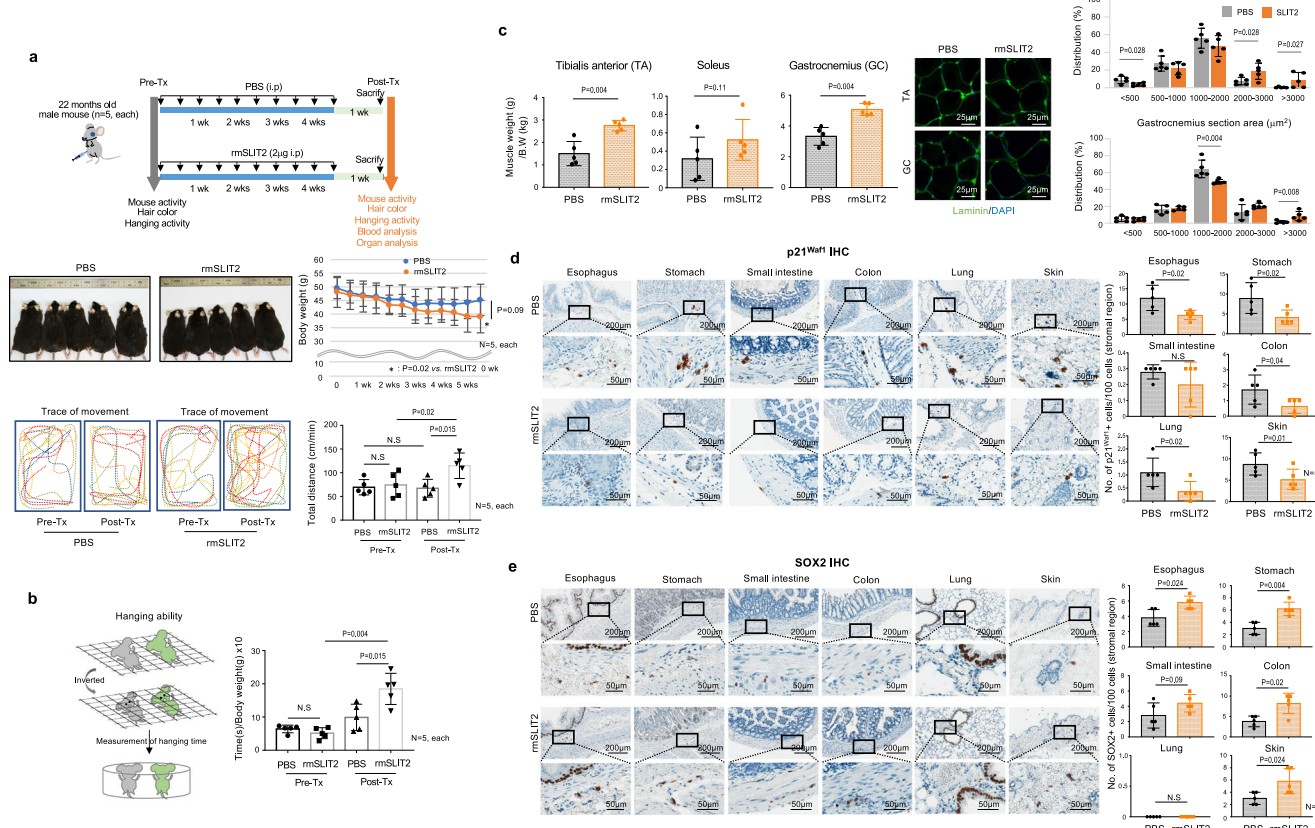

**Fig. 7 | Administration of rmSLIT2 increased activity in aged mice. a** PBS (control) or rmSLIT2 (2 µg/injection) was intraperitoneally injected twice weekly for 5 weeks in 23-month-old male mice (*n* = 5; PBS, *n* = 5; rmSLIT2, respectively) (upper panel). The body weight (middle panel) and activity in the cage (lower panel) were evaluated at the experimental beginning and endpoint. **b** The hanging ability of the rmSLIT2 or PBS-treated mouse was compared (left panel). The quantification data is shown in the right panel. **c** The muscle weight of the rmSLIT2 or PBS-treated mouse was compared (left panel). Laminin IF staining (middle panel). The tibialis anterior and gastrocnemius muscle were transversely sectioned to measure the area of muscle fibers (right panel). Representative images of laminin (green) and DAPI (blue) staining (middle panel). p21[Waf1] (**d**) and SOX2 (**e**) expression were analyzed by IHC in stromal region of esophagus, stomach, small intestine, colon, lung, and skin, respectively (*n* = 5; PBS, *n* = 5; rmSLIT2). Each protein expression status was indicated positive cells number/100 stromal cells. The *p* values of (**a–e**) are obtained from the one-tailed Mann–Whitney *U* test. The graphs in (**a–e**) were represented as mean ± SD.

conserved sequence between humans and mice[41]. Thus, we hypothesized that SLIT2 could be an anti-aging factor in both species. We conducted two independent experiments using aged mice. In the first experiment, we treated 22-month-old C57BL/6 J mice (*n* = 4 PBS and *n* = 5 rmSLIT2-treated, male and female each, one mouse in the group treated with PBS died during the experiment) with recombinant mouse SLIT2 protein (rmSLIT2) 6 times. This resulted in increased activity in the cage and decreased levels of the inflammatory marker SAA1 in the blood (Supplementary Fig. 18a–b and Supplementary Movie 2). However, we did not observe significant morphological changes in various organs (Supplementary Fig. 18c). In the in vitro experiments (Fig. 6), rhSLIT2 treatment showed effects when applied for a long period. Therefore, in the second round of animal experiments, we treated 23-month-old male C57BL/6 N male mice (*n* = 5 PBS and *n* = 5 rmSLIT2-treated) with rmSLIT2 10 times over a duration of several weeks. In this second experiment, we analyzed additional indices including activity, muscle weight, the number of mid-old cells in tissues, and blood analysis. Activity in the cage and hanging ability were measured before and after rmSLIT2 treatment to reduce animal to animal bias. During the experiment, body weight showed decreasing tendency in the rmSLIT2-treated groups. After 10 rounds of treatment, mouse aging-related characteristics significantly improved. Specifically, activity in the cage and hanging ability increased (Fig. 7a, b, and Supplementary Movie 3). However, blood analysis showed no significant difference between the control and rmSLIT2-treated groups (Supplementary

Fig. 19). To check the possibility that SLIT2 may also affect young mouse function due to its known effect on CNS development[45], rmSLIT2 was treated to young mice. However, rmSLIT2 did not affect activity in young mice (Supplementary Fig. 20 and Supplementary Movie 4). Necropsy results of rmSLIT2-treated aged mice demonstrated an inhibition of muscle mass reduction in leg muscle weight, including gastrocnemius and tibialis anterior (Fig. 7c, left panel). Laminin IF data showed larger muscle fiber size (Fig. 7c, right panel). Interestingly, the number of p21[Waf1]-positive mid-old cells decreased in stroma-rich organs, including the stomach, esophagus, colon and skin in rmSLIT2-treated group (Fig. 7d). Furthermore, the number of SOX2-expressing cells was increased in rmSLIT2-treated mice (Fig. 7e). Additionally, it was observed that the expression of functional markers, NHE3 and SCNN1A, decreased upon rmSLIT2 treatment (Supplementary Fig. 21). In summary, these findings demonstrate that SLIT2 protein functions as an anti-aging molecule in aged mice.

## Discussion

The regulatory mechanisms for functional decline in human organs are incompletely understood. The accumulation of senescent cells in the organs of the elderly may provide a plausible explanation for this decline. Multiple studies employing various senescence models have consistently demonstrated an increase in p16[INK4A]-positive cells, which are believed to be senescent, in aged mice[1,5]. Omori and colleagues revealed that the proportion of tdTomato (p16[INK4A])-positive cells

increased in aged mice; in the lung, skin, liver and kidney tissues of 12-month-old mice compared to 2-month-old mice using p16[INK4A]-CreERT2-tdTomato mice[1]. While senescent cells can exert a significant impact on their surrounding environment through the expression of SASPs, we have conjectured that this phenomenon, which is associated with the accumulation of senescent cells, may not be the sole explanation for the functional decline observed in aged organs. This is because p16[INK4A]-negative cells still make up the majority of the organ's population according to the abovementioned studies[1,5]. Instead, we put forth a hypothesis that the accumulation of cells in an intermediate stage of the cellular senescence process, referred to as 'mid-old' cells in the current study, might be another relevant factor in understanding this decline in organ function. Interestingly, our current data show that a considerable number of mesenchymal cells in aged organs exhibit the intermediate stage of the cellular senescence process rather than epithelial cells. This might be related to the relatively rapid turnover of epithelial cells, which predominately undergo apoptosis or are phagocytized by macrophages prior to induction of the senescence process[8,9]. The full extent of the effects of senescent and mid-old cells on the functional decline observed in aged organs remains uncertain. However, there is a strong indication that these cells may have a synergistic effect, meaning that their combined presence and activities could play a significant role in contributing to this decline. Further research is needed to better understand the interplay between senescent and mid-old cells and their collective impact on organ function during the aging process, and this interaction will be another interesting future studies.

Our in vitro transcriptomic studies showed that mid-old cells are functionally impaired compared to young cells; ECM formation, proliferation capacity, and tissue regeneration ability are significantly decreased. Nevertheless, it was observed that mid-old cells undergo a certain degree of functional recovery through the paracrine effect of substances released by young cells. Our study revealed that SLIT2 had a positive impact on the functional capacity of mid-old cells, although it remains uncertain whether it increased their maximum functional limit or enhanced their remaining functional potential. This is consistent with a prior study which demonstrated that Slit2-transgenic mice had improved organ function during aging[46]. Surprisingly, SLIT2 decreased inflammatory response and improved proliferative capacity only in mid-old, but not in old fibroblasts. Moreover, SAA1 expression was regulated by SLIT2, but SAA1 is not related to cell proliferation (Supplementary Fig. 22). Therefore, SLIT2 is a potential cell cycle regulator, and this could be attributed to the upregulation of stem cell factors, including SOX2 and OCT4. Previous reports demonstrated that SOX2 and OCT4 regulate the downregulation of p21[Waf1] expression via direct binding to its promoter or DNA methyltransferase[47,48]. Based on our cautious interpretation, we suggest that SLIT2 may enhance cell proliferation by upregulating stem cell factors, particularly SOX2. However, it is important to note that effects of SLIT2 are not universally applicable. Specifically, senescent fibroblasts did not exhibit a response to SLIT2, and mid-old fibroblasts showed diminished effectiveness when DT exceeded 8 days (Supplementary Fig. 23). Here, another important question arises: Can SLIT2 extend lifespan? Although our mouse sample size was small ($n = 4$, each), we administered PBS or rmSLIT2 injections once a week to 23-month-old mice and measured longevity. However, we did not observe evidence that SLIT2 extends lifespan, which is a major drawback of this study. Nonetheless, we believe that further investigation using a larger sample size of aged mice is necessary to draw conclusive results.

Regarding the changes in colon function with aging, we observed alterations in gene expression related to water absorption and the smooth muscle function involved in bowel movements in the large intestine. Specifically, we observed that the muscular mucosa layer became thinner in elderly colon tissue, and this was associated with increased expression of SAA1. Additionally, the expression of

'Atrogenes' was heightened in smooth muscle cells due to the presence of SAA1. Our findings suggest that SAA1 can induce atrophy in smooth muscle cells and may contribute to systematic muscle atrophy as circulating SAA1 levels increase with age. These results imply that targeting the master regulators of SAA1 directly could potentially prevent muscle mass loss in elderly individuals. Furthermore, we noted an increase in the expression of NHE3 and SCNN1A, genes related to water absorption, in colon and lung epithelial cells in aged tissues. Notably, this increase in NHE3 and SCNN1A expression was not attributed to SAA1 but rather appeared to be induced by the effects of degraded ECM on the stromal microenvironment. These degraded ECM fragments were found to function as 'Matrikines,' which upregulated signaling pathways such as Erk1/2, JNK, and NFκB. It is through these pathways that the expression of NHE3 and SCNN1A is regulated. Therefore, collagen degradation by MMPs could serve as 'Matrikines,' potentially leading to increased expression of NHE3 and SCNN1A.

This study offers new insights into the heterogeneous nature of the senescence ecosystem. The current concept and definition of senescence can be somewhat broad and vague. While the term 'cellular senescence' is often used interchangeably with 'irreversible cell cycle arrest[49]', there is growing evidence that certain aspects of a senescent cell can be reversed to a proliferative state[50–52]. Furthermore, the 'mid-old cells' identified in our study could represent an intermediate state of cellular senescence, as they are positive for the conventional markers, such as p21[Waf1] and IL1β. We demonstrated that the reduced or halted proliferation capacity of these mid-old cells can be reversed with appropriate stimulation. However, the borderline of cellular traits and the definition between 'mid-old' and 'early senescence' is still vague, and the additional study is needed to elucidate this question. Moreover, previously mentioned facts suggest that there could be various subtypes within the cellular senescence which makes us hard to understand the overall senescence ecosystem. Senescence encompasses various stages in cellular development, including early phases where proliferation ability can be restored, as well as later stages where cells take on specialized roles in molecule secretion or play a role in either promoting or inhibiting immune clearance processes. Furthermore, the trait of mid-old cells in pathologic conditions can be different from those in normal tissues, and the role of mid-old cells in pathologic conditions, such as age-related diseases will be another interesting future study.

Conclusively, our results demonstrated that most fibroblasts and smooth muscle cells in the organs of elderly subjects were in the mid-old status. Furthermore, SLIT2 protein originating from young cells could act as an anti-aging protein for mid-old cells. Collectively, the study proposes a new strategy for anti-aging, including targeting mid-old cells and restoring young cell factors to improve quality of life in elderly persons.

## Methods

### Human tissue experiments

This study complies with the Declaration of Helsinki and was approved by the Institutional Review Board of Ajou University Hospital (AJIRB-BMR-OBS-20-552, Suwon, Korea), and informed consent was obtained from all patients, parents, or legal guardians as appropriate. We obtained serval normal tissues (colon, lung, liver, and skin) from surgically resected specimens adjacent to pathogenic regions by an experienced pathologist. Although histologically normal tissues adjacent to tumors were present within 1 cm from the margins of the tumor, to exclude possibility potential contamination with tumor tissue, we obtained normal tissues from histologically normal (non-pathologic) tissues around the resection margin of surgically resected specimens which were located at 5–12 cm from the pathologic regions. Normal tissues were separately sampled in representative areas by an experienced pathologist, fixed immediately and processed to be formalin-fixed paraffin-embedded (FFPE), according to the tissue

specimen regulations of Ajou University Hospital (Supplementary Table 1-5). The standard age was classified into two groups: young (≤40 years) and elderly for colon (≥80 years), lung (≥75 years), liver (≥80 years), serum (≥70 years), and skin (≥70 years).

## Animal experiments

Male and female C57BL/6J mice (4-month-old and 12-month-old) were purchased from Korea Basic Science Institute (KBSI, Gwangju, Korea). The 12-month-old mice were raised under clean room system until they became 18 and 22 months old. C57BL/6N aged mice were obtained from ORIENT BIO (Seongnam, Korea) and subsequently raised until they reached 4 months or 23 months old. All animal procedures were approved by the institutional animal research ethics committee at Ajou University Medical Center (approval number: 2020-0051). Blood was immediately collected by intracardial puncture. Blood gas and electrolytes were analyzed by I-Smart 300 Blood Gas & Electrolyte Analyzer (I-SENS Inc., Seoul, Korea). Serum was collected by centrifuging blood samples. Colon, small intestine, stomach, lung, liver, esophagus, and skin were isolated and embedded in paraffin. To determine whether SLIT2 administration caused an anti-aging phenotype, recombinant mouse SLIT2 (5444-SL, R&D Systems, Minneapolis, MN, 2 μg/mouse) was intraperitoneally injected twice weekly for 3 weeks in 22-month-old, 5 weeks in 23-month-old, or 5 weeks in 4-month-old mice. The movement of the mice was recorded on video for a duration of a minute, two times before reaching the experimental endpoint. Movement activity was determined using measurement of the total movement distance for 1 min. Each mouse movement recorded on video was tracked manually. For the four limb-hanging tests, mice were positioned on a metal wired mesh and inverted. Hanging time was normalized to body weight. The results were averaged from 5 trials for each mouse.

## Muscle weight and cross-sectional area (CSA) measurements

The skeletal muscles containing gastrocnemius, tibialis anterior and soleus muscle were dissected from the hindlimb and measured a weight and embedded in an OCT compound (Sakura Finetek, Torrance, CA). Sections were cut transversely at 7-μm thickness. For measurement the cross-sectional area (CSA), Laminin antibody (1:500, ab11575, Abcam, Cambridge, UK) was incubated overnight at 4 °C. Sides were washed with PBS and incubated with Alexa-488 conjugated secondary antibody (1:400, A21206, Invitrogen, Carlsbad, CA) for 1 h at room temperature and then stained with DAPI (ab228549, Abcam) for staining nuclei. Images were visualized with a Zeiss LSM 710 confocal laser microscope (Zeiss, Oberkochen, Germany). The CSA was measured as the internal laminin unstained area by the ImageJ software (NIH, Bethesda, MD, freeware imaging software). Two hundred fifty fibers per muscle were measured.

## Cell culture

Normal human dermal fibroblasts were isolated from foreskin. Isolated single cells were cultured in Dulbecco's modified Eagle's medium (Gibco, Carlsbad, CA) supplemented with 10% fetal bovine serum (FBS) (Invitrogen) and antibiotics (Invitrogen) at 37 °C in a humidified incubator with 5% $CO_2$. Young cells were defined as DT < 2 days, mid-old cells as DT = 5–7 days, old cells as DT > 14 days. Primary pulmonary artery smooth muscle cells (PASMCs) were purchased from American Tissue Culture Collections (ATCC, Rockville, MD) and were cultured in vascular cell basal medium (PCS-100-030, ATCC) supplemented with vascular smooth muscle cell growth kit (PCS-100-042, ATCC) at 37 °C in a humidified incubator with 5% $CO_2$. Human pulmonary alveolar epithelial cells (HPAEpiC) and human primary colonic epithelial cells (HCoEpiC) were purchase from ScienCell Research Laboratories, Inc. (Carlsbad, CA) and Cell Biologics, Inc. (Chicago, IL), respectively, and were cultured in Alveolar Epithelial Cell Medium (ScienCell Research Laboratories, Inc.) and Human Epithelial Cell Basal Medium (Cell

Biologics, Inc.), respectively. SW480 cells were purchased from ATCC and were cultured in RPMI1640 medium (WELGENE Inc., Gyeongsan, Korea) supplemented with 10% FBS (Invitrogen) at 37 °C. The IMR90 cells, purchased from ATCC, were cultured in DMEM media with 10% FBS at 37 °C. Heterochromatin foci were assessed by staining the cells with a 1:10,000 diluted DAPI solution (ab228549, Abcam) and examining them using a Zeiss LSM 710 confocal laser microscope.

## Real-time PCR

First-strand cDNA was produced from 1 μg total RNA using the RevertAid First Strand cDNA Synthesis Kit (Thermo Fisher Scientific, Waltham, MA) and oligo(dT) primer mix. Real-time PCR was conducted with Power SYBR Green PCR Master Mix (Bio-Rad, Hercules, CA) using the following conditions: Initial activation at 95 °C 5 min, by 40 cycles of 95 °C for 15 s and 60 °C for 1 min. The primers used for real-time PCR are specified in Supplementary Table 6.

## Lentivirus preparation

Lentiviral particles were generated by co-transfection of the lentiviral expression vector (lentivirus plasmid, shRNA lentiviral vector) with lentiviral packaging plasmids (pGagpol, pVSV-G). HEK-293TN cells were transfected with the lentiviral vectors, and the virus-containing supernatant was harvested after 48 h. The SBLC and SLIT2 lentivirus plasmid was constructed by Vector Builder (Chicago, IL) and packaged. Human CXCL12 was cloned from normal human fibroblasts in the laboratory. The coding sequence of CXCL12 was inserted into the pCDH-CMV-MCS-EF1-Puro lentiviral vector (System Biosciences, Mountain View, CA). shRNA sequences are specified in Supplementary Table 7.

## Immunocytochemistry

Cell culture plates were placed in an incubator at 37 °C, 5% $CO_2$ for 24 h. Cells were fixed with 4% paraformaldehyde for 15 min and permeabilized with 0.05% Triton X-100 in 1× PBS (0.1% TWEEN-20) for 15 min. Plates were washed twice with 1× PBS-T (0.5% TWEEN-20) and blocked with 1% BSA in 1× PBS for 30 min. Primary antibodies were applied overnight, including Ki67, 1:500 (M7240, Dako, Santa Clara, CA); p-Erk1/2, 1:100 (9101, Cell Signaling Technology, Danvers, MA); p-NFκB, 1:100 (3033, Cell Signaling Technology); p21$^{Waf1}$, 1:100 (ab109520, Abcam); p16$^{INK4A}$, 1:300 (ab189034, Abcam). Cells were washed two times with 1× PBS-T and incubated with appropriate conjugated secondary antibodies for 1 h at room temperature. Secondary antibodies were as follows: Alexa Fluor 488, 1:600 (A-21206, Thermo Fisher Scientific); Alexa Fluor 555, 1:600 (A-31572, Thermo Fisher Scientific).

## EdU proliferation assay

EdU proliferation assay was performed using EdU proliferation kit (ab222421, Abcam) following the protocol. Cells were plated on cover glass and incubated overnight. The cell culture media were then replaced with 10 μM EdU and incubated overnight. Afterward, cells were fixed with 4% formaldehyde and permeabilized using permeabilization buffer. Subsequently, cells were washed with Tris-based saline buffer and treated with the reaction mix containing iFluor 647 azide and EdU additive solution for 30 min. Image acquisition was conducted using a Zeiss LSM 710 confocal laser microscope.

## RNA sequencing analysis

Total RNA was extracted from three independent samples using a Macherey-Nagel RNA kit (Macherey-Nagel GmbH & Co. KG, Düren, Germany). Briefly, sample quality was checked using a Bioanalyzer RNA Chip (Agilent Technologies, Santa Clara, CA) and RNA-seq was conducted with a Nextseq 500 (Illumina, San Diego, CA). To prepare the data for analysis, Skewer version 0.2.2 was used to trim potentially existing sequencing adapters and raw quality bases from the raw reads.

The resulting high-quality reads were then mapped to the reference genome using STAR version 2.5 software. A sequencing library was prepared by LAS Inc. (Gimpo, Korea). Gene models based on the gene annotation of the reference genome hg38 from the UCSC genome in GTF format were used to calculate gene expression values in fragments per kilobase of transcript per million fragments mapped (FPKM) units. Heatmaps drawn in this study used FPKM data to visualize the gene expression. For variance stabilizing transformation (VST) and GSEA, raw sequencing data are trimmed and mapped using AltAnalyze (v.2.1.4.3). Raw count data are loaded in the R (v 4.0.3) DESeq2 package (v.1.38.3). DESeq data set was generated using 'DESeq' function. Sample distance was generated by calculating VST and drawn using 'pheatmap' function. GSEA was performed using the "fgsea" package (v.1.24.0) from R. Permutations were performed 1000 times, and $p$ values were calculated using statistics from GSEA. The weighted Kolmogorov-Smirnov statistics were calculated to a running sum of a ranked gene list.

### Analysis of public scRNA-seq datasets for human tissues
In the analysis of cells from normal tissue, the publicly available scRNA-seq datasets, GSE178341[34] (peri-lesional normal colon) was utilized. The gene expression of young (35 and 42 years old) and Old (81, 81, 82, and 90) patients were compared and analyzed. The offered count data and metadata were loaded into 'Seurat Object' using the R (version 4.0.3) 'Seurat' package (version 4.3.0) Following normalization was performed using the 'NormalizeData' function, and principal component analysis (PCA) was performed using the 'RunPCA' function. To determine dimensionality, 'JackStraw' function was performed with 100 times of replications. Cells with the number of featured RNA >200 & <10,000 and <50% mitochondrial gene percentage were selected. *FN1* and *ACTA2* were used to annotate fibroblasts and smooth muscle cells, respectively.

### IHC and IF staining
Immunohistochemical staining of FFPE tissue sections (4-μm) was performed on Benchmark XT automated IHC stainer (Ventana Medical Systems Inc., Tucson, AZ). The primary antibodies used were as follows: human p16INK4A, predilution (805-4713, Roche, Tucson, AZ); human Ki67, 1:3000 (M7240, Dako); human H3K9me3, 1:500 (ab176916, Abcam); human p21Waf1, 1:300 (2990-1, Epitomics, Nanterre, France); human IL1β, 1:100 (ab9722, Abcam); human SAA1, 1:75 (MAB30191, R&D Systems); human SLIT2, 1:80 (HPA023088, Atlas Antibodies, Bromma, Sweden); human type IV Collagen, 1:300 (ab6586, Abcam); human MMP9, 1:100 (GTX100458, Genetex Inc., Irvine, CA); human CXCL12, 1:100 (MAB350, R&D Systems); human CCL5, 1:200 (MAB278, R&D Systems); mouse p16INK4A, 1:300. (ab189034, Abcam); mouse Ki67, 1:200 (ab16667, Abcam); mouse SAA1/2, 1:1000 (ab199030, Abcam); mouse SLIT2, 1:200 (PA5-31133, Invitrogen); human/mouse NHE3, 1:500 (NBP1-82574, Novus Biologicals, Centennial, CO); human/mouse SCNN1A, 1:100 (ab272878, Abcam); mouse SOX2, 1:100 (ab92494, Abcam). IHC results were presented as the percentage of positive cells in the stromal or epithelial region in the three random areas at high power. The Benchmark XT automated immunohistochemistry stainer was used to perform double immunohistochemistry. The first antibodies were detected using the UltraView universal DAB detection kit (#760-500, Ventana Medical Systems Inc), followed by the detection of the second antibodies using the UltraView Universal Alkaline Phosphatase Red Detection kit (#760-501, Ventana Medical Systems Inc). For IHC analysis of type IV collagen and NHE3, an experienced pathologist evaluated the intensity of immunostaining and graded it as weak, moderate, or strong. Counterstaining of each stained slide was performed using Hematoxylin II (790-2208, Ventana Medical Sytems Inc.). For IF staining of FFPE tissue sections, the following primary antibodies were incubated overnight at 4 °C: SAA1, 1:50 (MAB30191, R&D Systems); p21Waf1, 1:100 (ab109520,

Abcam); IL1β, 1:100 (ab9722, Abcam); SLIT2, 1:200 (PA5-31133, Invitrogen); p16INK4A, predilution (805-4713, Roche); Vimentin, 1:400 (ab92547, Abcam); Vimentin, 1:100 (AF2105, R&D Systems); Smoothelin, 1:100 (OMA1-06020, Invitrogen); Smoothelin, 1:500 (NBP2-37931, Novus Biologicals); E-cadherin, 1:200 (610181, Becton Dickinson); E-cadherin, 1:100 (ab15148, Abcam); CDX2, 1:100, (235R-15, Cell Margue, Rocklin, CA). Slides were washed twice with PBS and incubated with appropriately conjugated secondary antibodies for 1 h at room temperature. Secondary antibodies for IF staining were as follows: Alexa Fluor 488, 1:300 (ab150113, Abcam); Alexa Flour 488, 1:300 (A-11055, Invitrogen); Alexa Flour 405, 1:300 (A48255, Invitrogen); Alexa Fluor 594, 1:300 (ab150080, Abcam); Alexa Fluor 594, 1:300 (ab150116, Abcam). The stained cells were visualized with a Zeiss LSM 710 confocal laser microscope.

### Image processing for Erk1/2 activity reconstruction
Registration and subsequent background subtraction of fluorescence images was performed using ImageJ software. We adopted same strategy as CellProfiler[53] for nuclear and cytosol segmentation and quantified the average fluorescence level of Erk-KTR-mClover[54] for each segment using custom MATLAB codes. Briefly, cell lines were incubated with 500 ng/ml Hoechst (H1399, Invitrogen) for 10 min to label the nuclei. Subsequently, the cytoplasmic area was defined by a 5-pixel (~1.1 μm) annular ring from the nuclear region. Tracking of each cell was conducted using a Munkres assignment algorithm to record the trace of mean fluorescence intensity ratio (cytoplasmic over nuclear, C/N). The extracted C/N ratio was then normalized to frame just prior to stimulation. Normalized C/N ratio curves were further fitted with the following equation to obtain kinetic parameters (trans-localization rate $k_{on}$, time delay $t_{on}$ and C/N ratio at equilibrium $R_{eq}$).

$$r(t) = \begin{cases} 1 & for\ t \le t_{on} \\ R_{eq} + \left(1 - R_{eq}\right)\exp\left(-k_{on}(t - t_{on})\right) & for\ t > t_{on} \end{cases}$$

If the C/N ratio pattern was recovered (such as old cells in Supplementary Figure 3c), we fitted the C/N curve using the equation below.

$$r(t) = \begin{cases} 1 & for\ t \le t_{on} \\ R_{eq} + \left(1 - R_{eq}\right)\exp\left(-k_{on}(t - t_{on})\right) & for\ t_{on} < t \le t_{off} \\ 1 + \left(r\left(t_{off}\right) - 1\right)\exp\left(-k_{off}\left(t - t_{off}\right)\right) & for\ t > t_{off} \end{cases}$$

Model fitting was conducted using the nlinfit function in MATLAB R2021.

### Live cell imaging
Fluorescence (DAPI and FITC filter, Nikon, Tokyo, Japan) and bright field difference interference contrast (BF-DIC, Nikon, TI2-C-DICP-I) microscopy was performed using an inverted microscope (Nikon, ECLIPSE Ti2-E) equipped with a perfect focus system (PFS, Nikon, TI2-N-NDA-P), a motorized stage (Nikon, TI2-S-SE-E), an electron multiplying charge-coupled device (EM-CCD, Andor Technology, Belfast, Northern Ireland, DU-897U-CS0) and a 60 x oil immersion objective lens (Nikon, CFI Apochromat TIRF 60XC Oil, NA 1.49). An epifluorescence illuminator (Nikon, Intensilight C-HFGI, 130 W) was used as a light source for fluorescence. Temperature (37 °C), $CO_2$ (5%), and humidity (80%) were maintained during the live cell imaging by using stage top incubator (UNO-T-H-PREMIXED, Okolab, Ottaviano, Italy).

### In vitro Matrigel degradation assay
SPLInsert™ (SPL Life Sciences Co., Ltd, Pocheon, Korea) with 6.5-mm polycarbonate membrane filters of 0.4 μm pore size were used to form dual compartments. Young or mid-old cells were seeded onto the

lower chamber of the SPLInsert™ and the upper chamber of the SPLInsert™ was coated with Matrigel™ (Becton Dickinson) to mimic the conditions of the BM, individually. The upper chamber was transferred into the lower chamber and maintained for 2 days. Two experiments were then individually performed using a Matrigel-coated upper chamber, as follows; for the in vitro Matrigel degradation assay, Matrigel-coated membrane in the upper chamber was cut and boiled with 2× sample buffer (24 mM Tris-HCl,pH 6.8, 10% glycerol, 0.8% SDS, 2% 2-mercaptoethanol and 0.04% bromophenol blue). The supernatant was used for Western blotting to detect type IV collagen (1:1000, ab6586, Abcam) using standard procedures. To analyze the epithelial cell functional marker, HCoEpiC ($5 × 10^4$) or SW480 ($5 × 10^4$) or HPAEpiC ($1.5 × 10^4$) cells were seeded on the Matrigel-coated upper chamber for 2 days and, two experiments were then performed. First, real-time PCR was performed to measure mRNA levels of functional marker genes. Second, immunocytochemistry staining was performed with F-actin staining to compare cell morphology. Cells were fixed in 4% paraformaldehyde for 10 mins and treated with TRITC- phalloidin (1:500, P1951, Sigma, St. Louis, MO) for 1 h and DAPI was counterstained (1:10000, D3571, Invitrogen) for 10 min. Stained cells were visualized with a Lionheart FX automated microscope (BioTek Instruments, Winooski, VT).

## Metabolomics analysis

Young (DT1) and mid-old (DT4) fibroblasts ($n = 3$, each) were cultured in 150 mm culture dishes (353025, Corning Inc., Corning, NY). Both cell types were seeded with the aim of reaching a final cell count of $3 × 10^6$ at the time of harvest, which occurred 2 days later. The conditioned media (CM) was harvested and then centrifuged at $200 × g$ for 5 min. The supernatant was gently collected, and the subsequent analysis was performed by Basil Biotech (Incheon, Korea). The samples were added with 80% methanol. After vortexing for 1 min and centrifugation at $2000 × g$ for 10 min, supernatant was transferred to a new 1.5 mL tube and completely dried using a HyperVAC-MAX VC2200 centrifugal vacuum concentrator (Hanil Scientific Inc., Gimpo, Korea). Dried metabolite contents were reconstituted in 100 μL of 0.1% formic acid in water and then subjected to liquid chromatography-mass spectrometry (LC-MS) analysis. LC-MS analysis for metabolomics was performed using a Q Exactive™ Hybrid Quadrupole-Orbitrap MS (Thermo Fisher Scientific) coupled with a 1290 Infinity ultra-high performance liquid chromatography (UHPLC) (Agilent Technology). Metabolite mixtures were loaded using a ZORBAX Eclipse Plus C18 Rapid Resolution High Definition (RRHD) column (2.1 × 50 mm, 1.8 μm particles). The mobile phase solvents consisted of 0.1% formic acid in water and 0.1% formic acid in 80% acetonitrile, and the flow rate was fixed at 0.2 mL/min. The gradient of mobile phase was as follows: 2.5% solvent B in 5 min, 2.5–12.5% solvent B in 29 min, 12.5–25% solvent B in 11 min, 25–37.5% solvent B in 11 min, 37.5–80% solvent B in 0.1 min, holding at 80% of solvent B in 13.9 min, 80–2.5% solvent B in 0.1 min, 2.5% solvent B for 19.9 min. The electrospray source was equipped with a Heated Electrospray ionization (HESI-II) Probe combined with the standard Thermo Scientific™ Ion Max source. Parameters were set as follows: positive mode, spray voltage; 3.8 kV, capillary temperature; and 320 °C. Properties of full MS/dd-MS2 were set up as follows: full-MS scans, 150–2000 $m/z$ of scan range, 70,000 of resolution at 400 $m/z$, 1 × $10^6$ of AGC target, and maximum IT of 60 ms. For MS2 scans, the following parameters were used: 17,500 of resolution at 400 $m/z$, 2 × $10^5$ of AGC target, maximum IT was 250 ms, ±2 $m/z$ of isolation width, and NCE for dd-MS2 of 30%. Obtained UHPLC-Orbitrap-MS/MS RAW files were processed using Compound Discoverer 3.1.1.12™ (Thermo Fisher Scientific). Untargeted metabolomics workflow was used to perform retention time alignment and compound identification. Identification of compounds was performed using mzCloud™ and ChemSpider.

## Lipidomics analysis

The analysis was performed by Basil Biotech. CM was harvested in a same method with the metabolomics analysis (see above). For lipid extraction from the harvest CM samples (young and mid-old, $n = 3$ each), a two-step method involving neutral and acidic extraction were used. At first, in neutral extraction, lipids from the samples were extracted according to the Folch method using a mixture of chloroform and methanol (2:1, v/v). The samples were vortexed and incubated on ice for 10 min. The samples were centrifuged at $13,800 × g$ for 2 min at 4 °C, supernatant was transferred to a new 1.5 mL tube. Next, in Acidic extraction, 750 μL of chloroform:methanol:HCl (1 N, 37%) (40:80:1, v/v/v) was added to the remaining samples. After incubating for 15 min at room temperature, 250 μL of cold chloroform and 450 μL of cold 0.1 M HCl were added to the sample. The mixture was vortexed for 1 min and centrifuged at $6500 × g$ for 2 min at 4 °C. The lower organic phase was collected and combined with the prior extract. The sample was then dried using HyperVAC-MAX VC2200 centrifugal vacuum concentrator (Hanil Scientific Inc.). Dried lipid contents were reconstituted in 50 μL of mobile phase solvent A:solvent B (2:1, v/v) and then subjected to LC-MS/MS analysis. LC-MS analysis for lipidomics was performed using a Q Exactive™ Hybrid Quadrupole-Orbitrap MS (Thermo Fisher Scientific) coupled to a 1290 Infinity UHPLC (Agilent Technology) with heated electrospray ionization (HESI). Lipids mixtures were loaded using a e Hypersil GOLD™ C18 HPLC column (2.1 × 100 mm; 1.9 μm particle size; Thermo Fisher Scientific). The mobile phase solvents consisted of (A) Acetonitrile:Methanol:Water (19:19:2, v/v/v) + 20 mmol/L ammonium formate + 0.1% formic acid (v/v) and (B) 2-propanol + 20 mmol/L ammonium formate + 0.1% formic acid (v/v), and the flow rate was fixed at 0.2 mL/min. The gradient of mobile phase was as follows: 0–5 min, 5% B; 5–15 min, 5–30% B; 15–22 min, 30–90% B; 22–25 min, 90% B; 25–26 min, 90–5% B; 26–30 min, 5% B. Parameters were set as follows: positive mode, spray voltage; 3.8 kV, capillary temperature; and 320 Celsius degree, and S-lens radio frequency(RF); 60%. Properties of full MS/dd-MS2 were set up as follows: full-MS scans, 150 to 2000 $m/z$ of scan range, 70,000 of resolution, 1 × $10^6$ of AGC target, and maximum IT of 60 ms. For MS2 scans, the following parameters were used: 35,000 of resolution, 1 × $10^5$ of AGC target, maximum IT was 250 ms, ±1 $m/z$ of isolation width, and NCE for dd-MS2 of 30%. Obtained UHPLC-Orbitrap-MS/MS RAW files were processed using Lipid Search 4.2TM (Thermo Fisher Scientific). Lipid profiling under the following conditions: parent search; 0.2 Da, product search; 5.0 ppm, precursor ion mass tolerance; 8.0 ppm, M-score threshold; 2.0.

## In vitro anti-aging experiment

Mid-old cells were co-cultured with young cells for 30 days. For co-culture, mid-old cells were seeded in a transwell (0.4-μm pore size, 6 well, Corning Inc.) low chamber (1 × $10^4$ cells /well) in a six-well plate. Young or mid-old cells were then seeded in the transwell upper chamber (1 × $10^4$ cells). The upper chamber was replaced weekly. For CM experiments, mid-old cells were treated with CM of young cells for 30 days. Mid-old cells were seeded onto six-well plates (1 × $10^4$ cells/well) and allowed to adhere for 24 h. Culture medium was then replaced with CM, which was harvested from the culture medium of young or mid-old cells (1 × $10^4$ cells) and centrifuged for 3 min at $200 × g$. The CM was replaced every 2 days. For SLIT2 experiment, mid-old cells were seeded in a 60-mm dish (5 × $10^4$ cells). Medium was then replaced with serum-free medium with or without recombinant SLIT2 protein (150-11, PeproTech, Rocky Hill, NJ) for 24–48 h.

## Immunoblotting

Cells were scraped off the plates, and lysates were collected in 1× RIPA buffer (20 mM Tris-HCl pH 7.5, 150 mM NaCl, 1 mM $Na_2EDTA$, 1 mM

EGTA, 1% NP40 and 1% sodium deoxycholate) containing protease inhibitor cocktail (K272, Biovision, Milpitas, CA) and phosphatase inhibitor cocktail (K282, Biovision). Samples were resolved by electrophoresis an SDS-PAGE gel and PVDF membranes probed with the antibodies indicated; p16INK4A, 1:1000 (LS-B1347, LSbio, Seattle, WA); p21Wafl, 1:1000 (ab109520, Abcam); p53, 1:1000 (MA5-12557, Invitrogen); p-Erk1/2, 1:1000 (9101, Cell Signaling Technology); Total Erk1/2, 1:1000 (9107, Cell Signaling Technology); MDM2, 1:1000 (ab16895, Abcam); Actin, 1:3000 (Abc-2004, Abclon, Seoul, Korea); p-NFκB, 1:1000 (3033, Cell Signaling Technology); Total NFκB, 1:1000 (8242, Cell Signaling Technology); IκBα, 1:1000 (4814, Cell Signaling Technology); p-Pyk2, 1:1000 (3291, Cell Signaling Technology); Total Pyk2, 1:1000 (3292, Cell Signaling Technology); SOX2, 1:500 (ab92494, Abcam); OCT4, 1:1000 (ab19857, Abcam); MMP9, 1:1000 (GTX100458, Genetex Inc.); Tubulin, 1:1000 (sc-32293, Santa Cruz Biotechnology inc., Dallas, TX); GAPDH, 1:1000 (60004-1-lg, Proteintech, Rosemont, IL).

### SA-β-Gal staining
Cells were fixed with 4% paraformaldehyde and incubated with SA-β-Gal (1 mg/ml X-gal, 40 mM citric acid/sodium phosphate pH 5.8, 5 mM potassium ferrocyanide, 5 mM potassium ferricyanide, 150 mM NaCl and 2 mM MgCl$_2$) solution for 12 h at 37 °C. After washing with PBS, SA-β-Gal-positive cells were then analyzed with inverted light microscopy (DMi1, Leica Microsystems, Wetzlar, Germany).

### ELISA analysis
Mouse Serum Amyloid A (SAA) Quantikine ELISA Kit (MSAA00, R&D systems) was used for quantitative measurement of circulating SAA levels according to the manufacturer's instructions. Human SAA1 concentration from plasm obtained from Ajou University Hospital, was measured using a Human Serum Amyloid A1 DuoSet ELISA (DY3019-05, R&D systems). Human SLIT2 (LS-F4840, LSBio), human MMP9 (DMP900, R&D system) and human IL1β (437004, Biolegend, San Diego, CA) concentration were measured from harvested medium of cells.

### Long non-coding RNA analysis
To isolate exosomes, conditioned medium was harvested, and differential centrifugation was performed using Ultracentrifugation. Cellular debris was removed by centrifugation at $300 \times g$ for 10 min, followed by centrifugation at $2000 \times g$ for 20 min to remove apoptotic bodies. Macrovesicles were removed by centrifugation at $10,000 \times g$ for 20 min, while exosomes were pelleted by centrifugation at $100,000 \times g$ for 70 min. Pelleted exosomes were resuspended in RNA extraction solution (Macherey-Nagel GmbH & Co. KG) for total RNA extraction. cDNA synthesis was performed using an oligo(dT) primer mix and random hexamer primer mix and PCR was performed using primers (see Supplementary Table 6). Amplified DNA product was analyzed by agarose gel electrophoresis.

### Single cell tracking
Cells, labeled with GFP, were tracked over time and through cell divisions using a Lionheart FX automated microscope (BioTek Instruments). Initially, $1 \times 10^2$ cells were seeded in a six-well culture plate (30006, SPL Life Sciences Co., Ltd.), and their individual coordinate locations were saved for subsequent tracking.

### Microscope image acquisition
IHC staining image acquisition was conducting using a Leica Aperio Scanscope CS and Aperio ImageScope software (Leica Microsystems.). Cell images were acquired using a Leica DMi1 Inverted Microscope and Leica Application Suite (LAS) software (Leica Microsystems). IF images were captured on a Zeiss LSM 710 microscope and analyzed with Zeiss Axio Imager software.

### Statistical analysis
Numerical data are presented as mean ± SD of independent determinations. Statistical differences were examined using the Student's $t$ test and the Mann–Whitney $U$ test. A two-tailed Student's $t$ test was conducted for samples following a normal distribution. For samples that did not follow a normal distribution, a one-tailed Mann–Whitney $U$ test was performed. The proportional difference between two groups is calculated by Chi-square test.

### Reporting summary
Further information on research design is available in the Nature Portfolio Reporting Summary linked to this article.

## Data availability
The data presented in this study are publicly available in NCBI's Gene expression Omnibus (GEO) and Metabolomics Workbench under the accession codes: GSE232425, GSE232427, GSE201457, GSE41714 (cDNA microarray), GSE178341 (scRNA-seq), ST002960 (metabolomics) [http://dev.metabolomicsworkbench.org:22222/data/DRCCMetadata. php? Mode=Study&StudyID=ST002960&Access=ZirF6602], and. ST002961 (lipidomics) [http://dev.metabolomicsworkbench.org: 22222/data/DRCCMetadata.php? Mode=Study&StudyID=-ST002961&Access=IrhC2910]. Source data are provided with this paper.

## Code availability
The R code used to generate figures for bulk RNA-sequencing, GSEA, scRNA-seq, and MATLAB code for live cell imaging used in this study is available upon request to the corresponding author, T.J.P. (park64@ajou.ac.kr).

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

## Acknowledgements

This experiment was supported by National Research Foundation of Korea (NRF-2019R1A2C2086127, NRF-2020R1A6A1A03043539, NRF-2020M3A9D8037604 to T.J.P.; RS-2023-00211049 to Y.-K.L.).

## Author contributions

Y.H.K., Y.-K.L., S.S.P., S.H.P., S.Y.E, W.J.L. and J.J. conducted the experiments. Y.S.L., D.S., H.Y.K., S.B.L., J.C.K., G.Y., and H.S.K. analyzed the data. T.J.P. and J.-H.K. conceptualized and supervised the study. Y.H.K., Y.-K.L., S.S.P., J.-H.K and T.J.P. wrote the original and revised manuscript.

## Competing interests

The authors declare no competing interests.
