## [Peer Review File · Nature Communications]

Mid-Old Cells are A Potential Target for Anti-aging Interventions in the ElderlyReviewer #1 (Remarks to the Author):

This is an interesting study by Young Hwa Kim and colleagues in which they report that cultured cell types that are subject to cellular senescence in vitro (including fibroblasts and smooth muscle cells) enter a pre-senescent stage referred to as a "mid-old" stage. This stage is characterized by extended duplication times, elevated levels of stress response proteins p53 and p21, and various transcriptomic changes that include several pro- and anti-inflammatory genes. A subset of these genes was studied in greater detail, including the pro-inflammatory genes IL1 β and SAA1 and the anti-inflammatory genes CXCL12 and SLIT2. Of these, SAA1 was shown to drive the expression of proteins associated with degeneration, including MMP9 and the atrophy-related genes FBXO30, FBXO32, and TRIM64. Furthermore, the authors demonstrate that early passage fibroblasts produce so-called juvenile-associated secretory phenotypes (JASPs) that act to "rejuvenate" mid-old cells in that they become more proliferative and less pro-inflammatory. They go on to show that SLIT2 is a key factor driving this rejuvenation program through the upregulation of the stem cell factors SOX2 and OCT4.

The authors explore the in vivo relevance of their findings using p21, IL1 β , SAA1, and SLIT2 as markers of "mid-old" cells. They report that tissues of old people or old mice typically contain more cells with elevated levels of p21, IL1 β , or SAA1 than the corresponding young tissues. Moreover, the tissues of old people or old mice contained more cells with low SLIT2 levels than their young counterparts. These findings led the authors to conclude that a substantial portion of aged tissues consists of "mid-old" cells. To examine the extent to which SLIT2 supplementation rejuvenates old mice, 22-month-old mice were injected with recombinant SLIT2 for three weeks and then monitored for improvements in aging-related characteristics. The authors report that treated mice were more active, had more hair, thicker epidermis, and more SOX2-positive cells, leading them to conclude that SLIT2 acts as a rejuvenator in vivo.

While the in vitro experiments are generally of high quality and provide meaningful new insights into the characteristics of "mid-old" stage cells, the corresponding in vivo work aimed at examining whether these insights have in vivo relevance in the context of aging is less convincing as detailed below. Improvements to these studies will be necessary to support the study's central conclusion (1) that most fibroblasts and smooth muscle cells in aged human and mouse tissues reside in a "mid-old" status in which cells assume pro-inflammatory and ECM-modifying properties that cause tissue dysfunction, and (2) that systemic supplementation of old mice with SLIT2 protein reverses this status and rejuvenates old tissues.

Main points:

- RE Fig 2: The authors should include a more comprehensive assessment of the transcriptomic changes between young, "mid-old", and senescent cells. How many genes are significantly altered as cells transition between these three stages and what are the functional clusters associated with these transcriptomic changes?

- RE FIG 3: The authors should be complimented for mapping "mid-old" cells in human tissues. However, the approach they use to characterize these cells has interpretational limitations.

First, they use IHC to perform single staining for p21, SAA1, IL1beta, and SLIT2 in human and mouse tissues but this approach does not provide any evidence that these mid-old markers are expressed in the same cell. For that, it would be necessary to use combinations of these markers in an immunofluorescence labeling approach.

Second, the intensity of the counterstain of the IHC sections makes it difficult to see the antibody-specific staining. I find it therefore very difficult to locate positive cells in most of the tissues examined and evaluate these results as currently presented.

Third, while for the quantitation of p16-positive cells the authors expressed the amount of these cells as a percentage of the total cells, for the quantitation of "mid-old" cells they use 0, +1, and +2 as measures. I was unable to find how these measures are defined. The authors should replace these values with percentages of total cells. The figure legends should provide detail about how

the analysis was conducted (number of samples, number of sections per sample, etc).

Fourth, it is not clear how the authors make accurate distinctions between various cell types in the various tissues without the use of cell markers.

Thus, multiple experimental improvements are necessary to justify the authors' conclusion that most fibroblasts and smooth muscle cells in aged human and mouse tissues reside in a "mid-old" status.

- RE Fig. 4 and 5: The authors conclude that the downregulation of type IV collagen in basement membranes of elderly organs is due to its degradation by MMP9 without providing evidence for this conclusion. What they could say is that increased SAA1 in old tissues correlates with decreased collagen type IV which would be consistent with the hypothesis that SAA1 may act to alter the integrity of the basement membrane through MMP9-mediated degradation of collagen type IV.

- RE Fig. 8: While supplementation of old animals with recombinant SLIT2 is a good approach to directly test the extent to which it acts to rejuvenate tissues and organs, the depth and execution of the analyses are suboptimal, which leads to interpretational problems: the findings do not suggest that SLIT2 protein functions as a rejuvenation molecule in aged mice.

First, baseline values (prior to SLIT2 supplementation) were not assessed for animals in the study. What was the extent of the hair loss or reduction in the activity level of each of the animals prior to treatment and how much did things improve with treatment (on an animal-to-animal basis)? Also, the studies require a young control group to assess the impact of SLIT2 on animal activity. For instance, SLIT2 is a known modulator of the central nervous system and as such could impact activity.

Second, histological evaluations of a number of tissues only yielded a significant effect on skin, on the epidermis. It is not clear why the small intestine, liver, lung, and colon are presented in the main figure.

Third, the assessment of SCNN1A and SOX2 expression levels in tissues are not convincing.

Fourth, a tissue-wide assessment of the extent to which mid-old cell numbers decline (using the markers established in this study) would be the most logical experiment to test the authors' theory that low SLIT2 promotes the formation of "mid-old" cells.

- RE Fig. 1: p16 staining alone is not a reliable (exclusive) marker of senescence but needs to be stained in combination with other senescence markers to properly define a cell as a senescent cell. The authors do use histone H3K9 trimethyl as a second marker, but it would be more informative if the authors would conduct a double immunofluorescence stain for p16 and histone H3K9 trimethyl (or p21 or HMGB1) to identify senescent cells in tissues. It is well established that senescent cell numbers increase with aging in various tissues and organs. So, the authors would have to conduct a more comprehensive characterization of senescence to substantiate their statement about rates of senescence with aging.

Minor point:

The title "The Elderly's Organs are Ready for Functional Rejuvenation" is not an appropriate title.

Reviewer #2 (Remarks to the Author):

In this manuscript entitled "The Elderly's Organs are Ready for Functional Rejuvenation", Kim et al. propose the existence of an intermediate cellular state between young/healthy cells and senescent cells, which they term "mid-old" cells, consisting in fibroblasts and smooth muscle cells that were

neither proliferative nor fully senescent. Surprisingly, and in contrast to extensive scientific literature, they claim that full-senescent cells are not increased in normal elderly tissue, including from human individuals and mice. The authors provide some evidence of upregulations of pro-inflammatory genes, such as IL1B or SAA1, and downregulation of anti-inflammatory genes, such as SLIT2 or CXCL12, in mid-old cells. They also propose that SAA1 promotes an inflammatory microenvironment via upregulation of MMP9 that affect the integrity of epithelial cells by altering the stability of the basement membrane. Finally, the authors suggest that the functional decline of mid-old cells can be rejuvenated by SLIT2 and provide an in vivo validation where mice were treated with recombinant SLIT2 protein.

Overall, this is a voluminous manuscript including a variety of molecular biology analyses, co-culture approaches, mouse model validations, and the use of a remarkable collection of clinical samples. However, the experimental support for many of the conclusions drawn by the authors is still weak, and they often fall into overinterpretations. I consider the proposed findings and mechanistic hypotheses are still preliminary and largely disconnected, and there are major concerns, as outlined below, that should be addressed at this stage.

Major concerns:

1. Figure 1 and Suppl Figs. 1-3: The central hypothesis of the authors (i.e. accumulation of non-senescent mid-old stromal cells during ageing) based on the comparison between old-aged and mid-aged tissues seems incomplete and fragmentary. The results largely depend on the efficiency of the p16 antibody and the quality of the images provided do not allow to test any conclusions. It should be noted that immuno histochemistry for p16 is known to be challenging in most tissues. In contrast to cumulative evidence by using both p16 reporter and/or ablation mouse models and clinical samples (e.g. Baker et al. Nature 2011, 2016; Jeon et al. Nat Med 2017; Xue et al. Nat Med 2018; Liu et al. PNAS 2018; Grosse et al. Cell Metab 2020, Martinez-Zumudio et al. Aging Cell, 2021 etc.; for review see Di Micco et al. Nat Rev Mol Cell Biol 2021), the authors state that fully senescent cells (commonly p16-positive) are uncommon in old-aged tissues. A much more comprehensive study should be done to support this conclusion. Importantly, and as the authors must know, there is not a universal biomarker of senescence but a collection of markers defining the senescence programme. As such, senescence assessment in tissues would need of colocalisation analyses to ensure that senescence biomarkers apply to the same cell (either by IF and/or IHC), for instance for p16/Ki67 (to ensure that p16 cells are not proliferative), p16/H3K9m3 or p16/p21. Sup Fig. 1b lacks of quantifications and statistics. Other biomarkers such as SenTraGor (equivalent to SABgal activity for archived materials) and gene expression profiles (Cdkn1a, Cdkn1b, LaminB, IL-6, IL1b, TNF, etc.) would be desirable.
2. Fig. 2 and Supp Fig 4. Based on my experience with fibroblasts, mid-old cells (DT5-DT7 days) will contain a percentage of young/healthy cells, really mid-old cells and senescent cells. Again, the assessment of senescence results deficient, as p16/Ki67 or p21/ki67 should be determined in co-staining but not separately (H3K9m3 or HP1gamma would help as heterochromatin marker too). I consider that EdU incorporation assays would be a more refined methodology to assess DNA replication vs Cell Cycle arrest and also to determine the heterogeneity of cell populations across the criteria of the authors to define young, mid-old and old cells.
3. Figure 3. The authors test SAA1 and SLIT2 in colon and lung samples, based on the transcriptomic results of Figure 2 with fibroblasts. But, these fibroblast are human dermal fibroblasts isolated from foreskin, which do not seem to be an appropriate model. For a lung model the authors should use IMR90 or WI-38 pulmonary fibroblasts, which are well-characterised as a model of senescence in response to multiple triggers.
4. Figure 4. Figure 4b and 4c indicating higher levels of MMP9 upon rhSAA1 should be validated by Western Blot and/or ELISA. Also, do mid-old or old fibroblasts and smooth muscle cells express more mRNA levels of MMP9? In addition, the design of the co-culture of Figure 4i and 4j does not seem to be representative of colon tissue, as the authors appear to use humand dermal fibroblasts and a colon cancer cell line (SW480) rather than epithelial cells (HCoEpiC). Then, the link that the authors make with NH3 protein expression in mouse and human samples in Figure 4K is an over

interpretation rather than a demonstration: "As a result, increased water reabsorption and smooth muscle atrophy of muscular mucosa can impair colonic function in the elderly, which could be due to SAA1 expressed in mid-old fibroblasts".

5. Figure 5. Following my comments above, experiment 5e should have been done with young and mid-old pulmonary fibroblasts, but not with human dermal fibroblasts. Also, the increased fold induction of SCNN1A seems to be small. Is it also increased in old fibroblasts?

6. Figure 6. As stated above, EdU incorporation assays would be more informative and quantitative in Figure 6b. Could the authors provide GSEA of pathways of DNA replication and Cell Division in Figure 6c?

7. Figure 7. A more concluding and convincing validation would require to generate SLIT2 KO young fibroblasts and mid-old fibroblasts and then supplement the culture with rhSLIT2, while proliferation assays and assessment of inflammatory genes and, most importantly, secreted factors at the level of protein is done (which is currently missing). In fact, the decreased mRNA expression levels of inflammatory genes upon Lenti-SLIT2 in Fig.7c seem really marginal.

8. Figure 8. The authors show some evidence that the administration of rmSLIT2 increased activity in aged mice. However, a direct (rather than circumstantial) role in organ rejuvenation, if any, is missing. There are not functional tests (e.g. spirometry or gas transfer test) and the SCNN1A immunostaining, or NHE3 or Sox2 immunostaining, represent minor evidence. Also, the effect size is low. Many questions remain pending, for example if the treatment expands the lifespan of the mice or whether it protects mice from damage/inflammation (e.g. mice subjected to irradiation).

Other relevant points:

1. There is lack of relevant and more detailed information regarding the clinical samples. A Table is missing including ages, pathological conditions, treatments, etc. Concerningly, the samples seem to come from surgically resected specimens from cancer patients. Even if the tissues were dissected at 5- 12 cm from the pathologic regions the physiological tissue microenvironment might be strongly impacted by the clinical conditions of the patient and treatments (e.g. neoadjuvant chemotherapy or radiotherapy).

2. The quality of the IHC images is poor across the manuscript (Figs. 1, 3, 4, 5 and 8, and Suppl Fig 1, 3, 8 and 9), and convincing magnifications are missing. It is very difficult for the reader to discern between positive or negative cells for the particular markers.

3. Sometimes the criteria of the authors seem biased. For example, the quantification of number of tissues in the graphs, where protein expression status was classified as grade 0, 1+ or 2+. Overall, the number of positive events (positive cells or stained surface) seem very low or marginal, partially explaining the criteria of the authors. In my opinion, a more objective quantification analysis might be obtained by analyzing the relative percentage of positive cells for the representative images. For the reproducibility of the data, all quantifications should have been done in an automated manner. Also, I don't think the calculation of the p value by using one-way ANOVA is the most refined approach, but a two-tailed student's t-test instead, that might be subjected or not to a correction (e.g. Welch's correction).

4. Figure 2. Could the authors confirm whether the genes presented in Figure 2b are differentially expressed with statistical significance? What is the criteria for the presented list of inflammatory/anti-inflammatory genes? This is not a defined dataset pathway but rather a "partial" list of genes.

5. Supp Fig. 16. The authors assume that factors restoring the proliferative capacity of mid-old cells should be p53-p21 pathway inhibitors but why not p16-Rb pathway inhibitors?

6. Figure 6d. The "shortlisting" of Lnc-SBLC, Lnc-SAL RNA1 and Lnc-ROR is unclear. Why do the authors focus on these particular LncRNAs?

Reviewer #1

This is an interesting study by Young Hwa Kim and colleagues in which they report that cultured cell types that are subject to cellular senescence *in vitro* (including fibroblasts and smooth muscle cells) enter a pre-senescent stage referred to as a “mid-old” stage. This stage is characterized by extended duplication times, elevated levels of stress response proteins p53 and p21, and various transcriptomic changes that include several pro- and anti-inflammatory genes. A subset of these genes was studied in greater detail, including the pro-inflammatory genes IL1 β and SAA1 and the anti-inflammatory genes CXCL12 and SLIT2. Of these, SAA1 was shown to drive the expression of proteins associated with degeneration, including MMP9 and the atrophy-related genes FBXO30, FBXO32, and TRIM64. Furthermore, the authors demonstrate that early passage fibroblasts produce so-called juvenile-associated secretory phenotypes (JASPs) that act to “rejuvenate” mid-old cells in that they become more proliferative and less pro-inflammatory. They go on to show that SLIT2 is a key factor driving this rejuvenation program through the upregulation of the stem cell factors SOX2 and OCT4.

The authors explore the *in vivo* relevance of their findings using p21, IL1 β , SAA1, and SLIT2 as markers of “mid-old” cells. They report that tissues of old people or old mice typically contain more cells with elevated levels of p21, IL1 β , or SAA1 than the corresponding young tissues. Moreover, the tissues of old people or old mice contained more cells with low SLIT2 levels than their young counterparts. These findings led the authors to conclude that a substantial portion of aged tissues consists of “mid-old” cells. To examine the extent to which SLIT2 supplementation rejuvenates old mice, 22-month-old mice were injected with recombinant SLIT2 for three weeks and then monitored for improvements in aging-related characteristics. The authors report that treated mice were more active, had more hair, thicker epidermis, and more SOX2-positive cells, leading them to conclude that SLIT2 acts as a rejuvenator *in vivo*.

While the *in vitro* experiments are generally of high quality and provide meaningful new insights into the characteristics of “mid-old” stage cells, the corresponding *in vivo* work aimed at examining whether these insights have *in vivo* relevance in the context of aging is less convincing as detailed below. Improvements to these studies will be necessary to support the study’s central conclusion (1) that most fibroblasts and smooth muscle cells in aged human and mouse tissues reside in a “mid-old” status in which cells assume pro-inflammatory and ECM-modifying properties that cause tissue dysfunction, and (2) that systemic supplementation of old mice with SLIT2 protein reverses this status and rejuvenates old tissues.

Response: We deeply appreciate for the reviewer 1 to give us the opportunity to revise our manuscript. Accordingly, we changed and added additional data in **Fig. 2b-j, 2l, Fig. 3, Fig. 4b-d, 4g, 4j, 4k, Fig. 5a, 5d, 5f-h, Fig. 6c-f, Fig. 7a, 7c, 7g-j, Fig. 8 and Supplementary Fig. S1b-d, Fig. S4a-f, Fig. S5a-c, Fig. S6a-c, Fig. S7a-b, 7d, Fig. S8a, Fig. S12b, Fig. S15a-b, Fig. 19a, Fig. S21, Fig. S22, Fig. S23a-b**. We sincerely hope that our amendments meet your expectations.

[Main points]

1. RE Fig 2: The authors should include a more comprehensive assessment of the transcriptomic changes between young, “mid-old”, and senescent cells. How many genes are significantly altered as cells transition between these three stages and what are the functional clusters associated with these transcriptomic changes?

Response 1: Thank you for the important feedback provided by the reviewer. Following the reviewer's recommendation, we conducted additional analyses using RNA sequencing data. Firstly, we conducted a variance stabilizing transformation (VST) to discern differences in overall gene expression patterns across three groups: young, mid-old, and old (**Fig. 2b**). The clustering based on VST revealed that samples were coherently grouped according to their passage number, which corresponded to the categories of young, mid-old, and old. Furthermore, the gene expression pattern of mid-old cells was found to be more similar to that of young cells rather than that of old cells. This finding suggests that the aged traits of mid-old cells may be more easily remedied than those of fully senescent (old) cells with appropriate stimulation.

Additionally, we conducted gene set enrichment analysis (GSEA) to identify the functional traits of each group. Top gene sets featured by *p* value and normalized enrichment score (NES) in each group are listed in **Supplementary Fig. S7b**. We evaluated the senescence-related phenotype using the representative senescence-related gene set, "FRIDMAN: senescence up" (**Fig. 2c**). Our analysis disclosed that the absolute senescence phenotype was exclusively present in fully senescent (old) cells, not in mid-old cells. This finding also suggests that mid-old cells are more amenable to functional reversal given the appropriate stimulus.

A detailed analysis using RNA-seq data was performed on the function of fibroblasts, based on the category defined by the previous study [Plikus et al., *Cell* (2021)]. It was found that the functions of fibroblasts, such as extracellular matrix production and tissue reorganization, declined both in mid-old and old cells. Although the overall inflammatory response of mid-old cells did not differ from young cells, the specific inflammatory pathway of IL1 β was upregulated in mid-old cells. Additionally, SAA1, an acute response protein, was identified in the IL1 β pathway as a leading-edge gene. These findings suggest that age-related inflammation may be primed by the IL1 β pathway. However, the degree of inflammation was highly upregulated in old cells compared to young and mid-old cells, with TNF α and IL6-related pathways being major inflammatory responses in old cells. Based on this data, it was found that young, mid-old, and old cells have completely different cellular characteristics from each other. We added related data in **Fig. 2b-j** and **Supplementary Fig. S7a-d**.

2. RE FIG 3: The authors should be complimented for mapping "mid-old" cells in human tissues. However, the approach they use to characterize these cells has interpretational limitations.

2-1. First, they use IHC to perform single staining for p21, SAA1, IL1beta, and SLIT2 in human and mouse tissues but this approach does not provide any evidence that these mid-old markers are expressed in the same cell. For that, it would be necessary to use combinations of these markers in an immunofluorescence labeling approach.

Response 2-1: Thank you for your insightful comment. In response to the reviewer's suggestion, we performed double immunofluorescence staining on the tissues; SAA1 (green)/p21^{Waf1} (red), and SAA1 (green)/IL1 β (red), respectively. We discovered that mid-old cells exhibiting positivity for p21^{Waf1} also showed positivity for SAA1. Furthermore, we found that IL1 β positive cells in old-aged colon and lung tissues also expressed SAA1. In addition, we detected the expression of SLIT2 in fibroblasts (vimentin positive cell) and smooth muscle cells (smoothelin positive cells) in young-aged colon and lung tissues. These results suggest that SAA1, IL1 β , and SLIT2 are respectively expressed in mid-old cells and young cells. We have included these findings in **Supplementary Fig. S8a**.

2-2. Second, the intensity of the counterstain of the IHC sections makes it difficult to see the antibody-specific staining. I find it therefore very difficult to locate positive cells in most of the tissues examined and evaluate these results as currently presented.

Response 2-2: We apologize for the poor quality of the image. In the revised version of the manuscript, we have added high-quality images with reduced counterstaining. We added high magnification and high-quality image in **Fig. 1a-d, Fig. 3a-d, Fig. 4d, 4k, Fig. 5a, 5b, 5g, 5h, Fig. 8d, 8e, Supplement Fig. S1b, 1c, 1e, S3, S4f, S10, S11, and S23a, 23b.**

2-3. Third, while for the quantitation of p16-positive cells the authors expressed the amount of these cells as a percentage of the total cells, for the quantitation of “mid-old” cells they use 0, +1, and +2 as measures. I was unable to find how these measures are defined. The authors should replace these values with percentages of total cells. The figure legends should provide detail about how the analysis was conducted (number of samples, number of sections per sample, etc).

Response 2-3: In response to the reviewer's comment, we have presented the number of mid-old cells in percentage form in the IHC. Additionally, we have included the measurement method and information of tissues in the **Materials and Methods** section; IHC results were presented as the percentage of positive cells in the stromal or epithelial region of three random areas at high power. For IHC analysis of type IV collagen and NHE3, an experienced pathologist evaluated the intensity of immunostaining and graded it as weak, moderate, or strong.

2-4. Fourth, it is not clear how the authors make accurate distinctions between various cell types in the various tissues without the use of cell markers.

Response 2-4: Thank you for your valuable feedback. Heeding your suggestion, we carried out double immunofluorescence staining on colon and lung tissues from young and old individuals; SAA1 (green)/vimentin (red), smoothelin (green)/IL1 β (red), smoothelin (green)/SAA1 (red), smoothelin (green)/p21^{Waf1} (red), vimentin (green)/p21^{Waf1} (red), vimentin (green)/IL1 β (red), smoothelin (green)/SLIT2 (red), and vimentin (green)/SLIT2 (red), respectively. Our findings indicated that cells positive for smoothelin (indicative of smooth muscle cells) and vimentin (indicative of fibroblasts) expressed SAA1, IL1 β and p21^{Waf1} in tissues from older individuals. Conversely, smoothelin and vimentin-positive cells expressed SLIT2 in tissues from younger individuals. We have incorporated these findings into **Supplementary Fig. S8a.**

3. RE Fig. 4 and 5: The authors conclude that the downregulation of type IV collagen in basement membranes of elderly organs is due to its degradation by MMP9 without providing evidence for this conclusion. What they could say is that increased SAA1 in old tissues correlates with decreased collagen type IV which would be consistent with the hypothesis that SAA1 may act to alter the integrity of the basement membrane through MMP9-mediated degradation of collagen type IV.

Response 3: Thank you for your crucial comment. We concur with the reviewer's observation regarding the absence of direct evidence for *in vivo* type IV collagen degradation by MMP9. Regrettably, the analysis of type IV collagen degradation by MMP9 in *in vivo* human tissue models presents certain limitations. Therefore, we conducted a single-cell analysis of type IV collagen mRNA levels in young

and old-aged colon tissues. Utilizing the data available in the public domain (GSE178341), we noted no significant difference in type IV collagen mRNA expression between young and old-aged normal colon tissues (**Fig. 4g**). Fibronectin-positive clusters 19 and 22 exhibited no difference in mRNA gene expression for COL4A1 and A2, and A3 was not detected in either tissue. These findings suggest that despite maintaining collagen type IV mRNA expression in old-aged tissues, the protein level was downregulated, providing indirect evidence of type IV collagen degradation in old-aged tissues.

Additionally, we verified type IV collagen degradation by MMP9 *in vitro*. We infected mid-old fibroblasts with shMMP9 lentivirus and subsequently applied the *in vivo* mimic model as depicted in **Supplementary Fig. S15a**. The results indicated that NHE3 and AQP3 expression was restored upon downregulation of MMP9. We hope this explanation addresses your concerns.

We acknowledge the understanding that basement membrane degradation by MMP9 alone does not solely regulate the functionality of epithelial cells. For example, our previous investigations [Yoon et al., *Theranostics* (2018)] on aging skin have revealed that senescent fibroblasts secrete diverse senescence-associated secretory phenotypes (SASPs), which play a crucial role in modulating the function of epithelial cells. Consequently, we regard MMP9-mediated basement membrane degradation as one of the mechanisms contributing to the regulation of epithelial cell function. We appreciate again your insightful comments.

4 RE Fig. 8: While supplementation of old animals with recombinant SLIT2 is a good approach to directly test the extent to which it acts to rejuvenate tissues and organs, the depth and execution of the analyses are suboptimal, which leads to interpretational problems: the findings do not suggest that SLIT2 protein functions as a rejuvenation molecule in aged mice.

4-1. First, baseline values (prior to SLIT2 supplementation) were not assessed for animals in the study. What was the extent of the hair loss or reduction in the activity level of each of the animals prior to treatment and how much did things improve with treatment (on an animal-to-animal basis)? Also, the studies require a young control group to assess the impact of SLIT2 on animal activity. For instance, SLIT2 is a known modulator of the central nervous system and as such could impact activity.

Response 4-1: Thank you for your valuable feedback. We concur with the reviewer's suggestion. To minimize the animal-to-animal bias, we performed additional experiments with 23-month-old male mice, both before and after rmSLIT2 treatment, as well as on 4-month-old young mice. The revised data have been included in **Fig. 8a-e, Supplementary Fig. S21, S22 and S23**.

Acquiring old-aged mice indeed presents certain limitations. In general, commercially available mice older than 1 year of age are not readily obtainable. Hence, we acquire less than 1-year-old mice and rear them in our research laboratory for approximately 1 year to generate aged mice for our experimental purposes. Thankfully, we were able to procure 23-month-old male C57BL/6N mice (n=18) (previously we used 22 months old C57BL/6J mice; Supplementary Fig. S20). We utilized a total of 18 mice for two separate studies; rejuvenation (n=5 PBS, n=5 rmSLIT2) and longevity (n=4 PBS, n=4 rmSLIT2, as suggested by reviewer 2) experiments. Although the number of mice used for both studies may not have been optimal, we made every effort to maximize our resources.

Despite the same species, C57BL/6N mice displayed a more youthful phenotype compared to C57BL/6J mice, with no observable hair loss in C57BL/6N mice. Thus, in this experiment, we examined various factors including mouse activity, hanging endurance, muscle weight, muscle cell size, body weight and

blood analysis. The results showed that most indices improved in the rmSLIT2 treatment group. Notably, leg muscle mass significantly increased in rmSLIT2-treated mice, including the tibialis anterior and gastrocnemius muscles. We added these data in **Fig. 8c**.

In the previous experiment, the expression of SOX2 slightly increase in the mice treated with rmSLIT2. We also extensively pondered the reasons behind the minimal effect of rmSLIT2 in mice. Ultimately, we hypothesized that this could be attributed to either the single administration dose or the duration of the treatment. However, our *in vitro* data (**Fig. 7**) revealed that a prolonged treatment period is necessary to observe the effects of rmSLIT2. Therefore, we extended the treatment duration of rmSLIT2 in mice from 6 rounds (3 weeks) to 10 rounds (5 weeks) would yield more significant results. We observed an increase in SOX2 expression in stroma-rich various organs including esophagus, stomach, colon, skin and lung (**Fig. 8e**). Furthermore, rmSLIT2 treatment reduced the number of mid-old cells in old-aged mice, as evidenced by a decrease in p21^{Waf1} positive stromal cells in the esophagus, stomach, colon, skin and lung (**Fig. 8d**)

Our data demonstrate that administering rmSLIT2 to aged mice significantly boosts their activity level (**Fig. 8a, b**). At this point, it remains unclear whether this effect is a consequence of rmSLIT2 positively affecting all organs or simply due to its role in maintaining muscle mass, warranting further investigation. While we have not been able to unravel the exact mechanism, we have observed that the administration of rmSLIT2 leads to a reduction in mouse body weight (**Fig. 8a**) and a decrease in intra-abdominal fat upon opening the abdominal cavity (**data below**). In our first experiment, we didn't observe a decrease in body weight in the rmSLIT2 treated group, likely because the treatment was only administered for short-term (6 rounds). However, in the second experiment, there was also no significant weight loss observed during the first three-week treatment period, but after five weeks of treatment, weight loss was evident (significant body weight reduction was observed compared to the start point of rmSLIT2 treatment, and when compared to the PBS group, a trend of body weight reduction was observed; as confirmed by two-tailed Mann-Whitney U test analysis). This suggests that rmSLIT2 could potentially influence fat cell reduction, lipolysis, or beta-oxidation, all of which are intriguing possibilities that warrant further investigation. In conclusion, we speculated that the maintenance of muscle mass and reduction in body weight contribute to increased activity levels and elevated hanging activity in mice.

According to your suggestion, we also evaluated the activity of 4-month-old young mice and found that they had a higher basal activity compared to old-aged mice. However, treating 4-month-old mice with rmSLIT2 for 10 times did not result in any significant difference in activity between the rmSLIT2 treated and non-treated young-aged mouse groups. These findings have been added to **Supplementary Fig. S22**.

Additionally, in response to the suggestion from reviewer 2, we investigated the potential of SLIT2 to induce an increase in lifespan. Unfortunately, we encountered a supply issue with aged mice, resulting in a limited number of mice available for the experiment (rmSLIT2, n = 4; PBS, n = 4). Consequently, we proceeded with the experiment using 23-month-old male mice, administering either PBS or rmSLIT2 (2 µg/mouse) once a week. However, we did not observe a difference in lifespan during the 22-week duration until 28.5 months of age (in each group, three mice have died so far, Survival curves Log-rank Test p=0.519). Two major reasons may explain this phenomenon. Firstly, the limited number of mice used in this study may have influenced the outcome. Secondly, these results suggest that SLIT2 may be associated with functional rejuvenation rather than extending lifespan at both cellular and individual levels. Nevertheless, we are unable to draw a conclusive statement at this stage. We firmly believe that further experiments should be conducted with a larger number of mice to obtain more definitive and reliable results. We hope this response satisfactorily addresses your question.

4-2. Second, histological evaluations of a number of tissues only yielded a significant effect on skin, on the epidermis. It is not clear why the small intestine, liver, lung, and colon are presented in the main figure.

Response 4-2: Thank you for the important feedback. While we cannot provide the exact reasoning, we believe that SLIT2 primarily affects stromal cells rather than directly impacting epithelial cells. In the case of lung, multiple previous studies [Schulte et al., *Front. Physiol.* (2019) and Yazicioglu et al., *Am. J. Physiol. Lung Cell Mol. Physiol.* (2020)] demonstrated that age-related structural and functional alterations in the mouse lung are primarily caused by micromechanical changes, resulting in increased lung parenchymal volume and alveolar airspace. Consequently, aged lungs already exhibit changes in the ECM components in their microenvironment, such as fibrosis, although the exact mechanism of this phenomenon is still not fully understood. However, the number of type II alveolar cells, which is related to surfactant release and differentiation into type I alveolar cells, remains unchanged between young and old mice. Therefore, it is plausible that rmSLIT2 treatment could be resistant to these changes in ECM constituents rather than cellular changes.

Therefore, in the revised version, we focused on whether the number of mid-old cells in the stromal region, as pointed out by the reviewers, decreased when treated with rmSLIT2. We found that the number of mid-old cells decreased in the skin, stomach, colon and esophagus when treated with rmSLIT2. Additionally, there was no significant decrease observed in the small intestine where mid-old cells were less frequently observed. We added these data in **Fig. 8d**.

The thickening of the epidermis in the skin may be due to rejuvenation of the dermis. Our group's previous observations suggest that dermal aging can induce aging in the entire skin including epidermis. Therefore, we believe that dermal rejuvenation may help with epidermal regeneration. However, there is little difference in the organs such as liver and lung tissues between young and aged mice. Therefore, we think it is difficult to find significant structural differences even with the treatment of rmSLIT2. However, we believe that aged mice tissues have functionally become younger rather than structurally different. Specifically, we assumed that the function of epithelial cells would also improve as the stroma becomes younger, and we observed a decrease in the expression of NHE3 in the aged mice colon and SCNN1A in the aged mice lung after treatment with rmSLIT2, which leads us to believe that aged mice tissue function has improved. We added these data in **Supplementary Fig. S23a, S23b**.

4-3. Third, the assessment of SCNN1A and SOX2 expression levels in tissues are not convincing.

Response 4-3: Thank you for pointing it out. We hypothesize that the slight increase in SOX2 expression observed in the previous experiment may be attributed to the short-term rmSLIT2 treatment. Therefore, in the subsequent experiment, we administered 10 rounds of rmSLIT2 as opposed to the initial 6 rounds and witnessed an increase in SOX2 expression across various organs. SOX2 presented strong staining in cells believed to possess stemness. As previously noted, stem cell activity does not appear to decline significantly with aging. Consequently, SOX2 staining was observed in the epithelial cell components of the stomach, esophagus, and lung. In aged mice, stem cells still have their function, hence the expression of SOX2 in stem cell-like cells remains unchanged after rmSLIT2 treatment. However, an increase in the number of SOX2-positive cells was observed in the stroma, where mid-old cells are present. This increase was noted in stroma-rich organs, such as the esophagus, skin, and stomach. These findings are depicted in **Fig. 8e**. Additionally, we have restained for SCNN1A, and these results are included in **Supplementary Fig. S23b**.

4-4. Fourth, a tissue-wide assessment of the extent to which mid-old cell numbers decline (using the markers established in this study) would be the most logical experiment to test the authors' theory that low SLIT2 promotes the formation of “mid-old” cells.

Response 4-4: Thank you for your insightful feedback. We wholeheartedly concur with the reviewer's comments. To explore the decrease in the number of mid-old cells, we administered rmSLIT2 proteins to 23-month-old mice 10 times and examined the quantity of p21^{Waf1}-positive cells in the stromal region. The data showed a significant reduction in the number of p21^{Waf1}-positive cells in the stroma and smooth muscle areas of various tissues, including the esophagus, stomach, colon and skin. However, the small intestine contained a small number of p21^{Waf1}-positive cells, and as a result, rmSLIT2 did not significantly reduce their count in this tissue. We have incorporated these data into **Fig. 8d**. We appreciate your valuable input.

5. RE Fig. 1: p16 staining alone is not a reliable (exclusive) marker of senescence but needs to be stained in combination with other senescence markers to properly define a cell as a senescent cell. The authors do use histone H3K9 trimethyl as a second marker, but it would be more informative if the authors would conduct a double immunofluorescence stain for p16 and histone H3K9 trimethyl (or p21 or HMGB1) to identify senescent cells in tissues. It is well established that senescent cell numbers increase with aging in various tissues and organs. So, the authors would have to conduct a more comprehensive characterization of senescence to substantiate their statement about rates of senescence with aging.

Response 5: Thank you for your valuable feedback. As highlighted by the reviewer, we performed double immunostaining with p21^{Waf1} and p16^{INK4A}. When p16^{INK4A} is stained, most cells are co-stained with p21^{Waf1}. However, in many cells, only p21^{Waf1} is observed to be stained. Thus, we postulate that fully senescent cells are not abundant in aged organs (**Supplementary Fig. S1b lower panel**). To further validate the presence of senescent cells, we conducted senescence associated- β -galactosidase (SA- β -Gal) staining in fresh frozen normal GI tract and lung tissues. As you are aware, the utilization of SA- β -Gal staining in our study presents certain limitations due to its requirement for fresh frozen tissues. Unlike paraffin-embedded tissues, acquiring a substantial number of tissues for analysis was unattainable, consequently rendering it infeasible to stratify the samples into distinct age groups, such as individuals below 40 years old and those above 75 years old. Nonetheless, we were able to procure a subset of GI tract and lung tissues, which were employed for SA- β -Gal staining. The analysis revealed that SA- β -Gal positive cells did not exhibit an increasing tendency according to aging (**Supplementary**

Fig. S1c).

As pointed out by the reviewers, an increase in fully senescent cells in elderly tissues has been reported. However, in our study, we observed an increase in mid-old cells rather than fully senescent cells. To validate our data more precise, we utilized gene analysis from publicly available single-cell data (GSE178341). In the analysis of human colon tissue, minimal expression of p16^{INK4A} was observed, particularly in epithelial cells and fibroblasts, where p16^{INK4A} expression was almost negligible. In contrast, p16^{INK4A} expression was found to increase in immune cells, specifically T cells, in elderly tissues. Therefore, we thought that the previously reported increase in p16^{INK4A} expression in bulk sequencing data is likely due to immune cell origins. Additionally, in single-cell analysis of aged mice (GSE132042), we observed no increase in p16^{INK4A} expression when analyzing whole tissues. When examining lung and colon in detail, p16^{INK4A} expression was also not increased. Supporting data has been included in **Supplementary Fig. S4 and S5**.

[Minor point]

1. The title “The Elderly’s Organs are Ready for Functional Rejuvenation” is not an appropriate title.

Response 1: According to reviewer’s comment, we changed the title as “Functional Rejuvenation of Elderly Organs”

”

Reviewer #2

In this manuscript entitled "The Elderly's Organs are Ready for Functional Rejuvenation", Kim et al. propose the existence of an intermediate cellular state between young/healthy cells and senescent cells, which they term "mid-old" cells, consisting in fibroblasts and smooth muscle cells that were neither proliferative nor fully senescent. Surprisingly, and in contrast to extensive scientific literature, they claim that full-senescent cells are not increased in normal elderly tissue, including from human individuals and mice. The authors provide some evidence of upregulations of pro-inflammatory genes, such as IL1B or SAA1, and downregulation of anti-inflammatory genes, such as SLIT2 or CXCL12, in mid-old cells. They also propose that SAA1 promotes an inflammatory microenvironment via upregulation of MM9 that affect the integrity of epithelial cells by altering the stability of the basement membrane. Finally, the authors suggest that the functional decline of mid-old cells can be rejuvenated by SLIT2 and provide an in vivo validation where mice were treated with recombinant SLIT2 protein.

Overall, this is a voluminous manuscript including a variety of molecular biology analyses, co-culture approaches, mouse model validations, and the use of a remarkable collection of clinical samples. However, the experimental support for many of the conclusions drawn by the authors is still weak, and they often fall into overinterpretations. I consider the proposed findings and mechanistic hypotheses are still preliminary and largely disconnected, and there are major concerns, as outlined below, that should be addressed at this stage.

Response: We deeply appreciate for the reviewer 2 to give us the very insightful comments, which have guided us in strengthening our study. Accordingly, we changed and added additional data in **Fig. 2b-j, 2l, Fig. 3, Fig. 4b-d, 4g, 4j, 4k, Fig. 5a, 5d, 5f-h, Fig. 6c-f, Fig. 7a, 7c, 7g-j, Fig. 8 and Supplementary Fig. S1b-d, Fig. S4a-f, Fig. S5a-c, Fig. S6a-c, Fig. S7a-b, 7d, Fig. S8a, Fig. S12b, Fig. S15a-b, Fig. 19a, Fig. S21, Fig. S22, Fig. S23a-b**. We sincerely hope that our amendments meet your expectations.

[Major concerns]

1. Figure 1 and Suppl Figs. 1-3: The central hypothesis of the authors (i.e. accumulation of non-senescent mid-old stromal cells during ageing) based on the comparison between old-aged and mid-aged tissues seems incomplete and fragmentary. The results largely depend on the efficiency of the p16 antibody and the quality of the images provided do not allow to test any conclusions. It should be noted that immuno histochemistry for p16 is known to be challenging in most tissues. In contrast to cumulative evidence by using both p16 reporter and/or ablation mouse models and clinical samples (e.g. Baker et al. Nature 2011, 2016; Jeon et al. Nat Med 2017; Xue et al. Nat Med 2018; Liu et al. PNAS 2018; Grosse et al. Cell Metab 2020, Martinez-Zumudio et al. Aging Cell, 2021 etc.; for review see Di Micco et al. Nat Rev Mol Cell Biol 2021), the authors state that fully senescent cells (commonly p16-positive) are uncommon in old-aged tissues. A much more comprehensive study should be done to support this conclusion. Importantly, and as the authors must know, there is not a universal biomarker of senescence but a collection of markers defining the senescence programme. As such, senescence assessment in tissues would need of colocalisation analyses to ensure that senescence biomarkers apply to the same cell (either by IF and/or IHC), for instance for p16/Ki67 (to ensure that p16 cells are not proliferative), p16/H3K9m3 or p16/p21. Sup Fig. 1b lacks of quantifications and statistics. Other biomarkers such as SenTraGor (equivalent to SABgal activity for archived materials) and gene expression profiles (Cdkn1a, Cdkn1b, LaminB, IL-6, IL1b, TNF, etc.) would be desirable.

Response 1: Thank you for your very important comment. We fully agree with the reviewer's opinion.

Also, we completely agree that relying solely on p16^{INK4A} immunohistochemistry is not sufficient to support our thesis that fully senescent cells are actually not accumulated in the elderly's tissue.

Therefore, to support our hypothesis, we conducted several additional experiments.

Firstly, we reanalyzed single-cell RNA expression data from publicly available datasets. Since bulk sequencing can encompass various cell types, we utilized single-cell analysis in human normal colon and aged mouse to observe the levels of p16^{INK4A} expression in each cell.

Secondly, we performed SA-β-Gal staining on frozen tissue. SA-β-Gal is considered a major marker of senescence. Therefore, we believed that SA-β-Gal staining data would provide essential information in measuring fully senescent cells in elderly tissues.

Thirdly, we analyzed the expression of p16^{INK4A} and p21^{Waf1} through double immunofluorescence analysis *in vivo*. In our *in vitro* studies, old cells exhibited the expression of both p16^{INK4A} and p21^{Waf1}, whereas mid-old cells only expressed p21^{Waf1}. Therefore, measuring the expression of these two proteins in elderly tissues could provide crucial information.

Lastly, we conducted double immunohistochemistry (IHC) analysis of p16^{INK4A} and Ki67, which revealed that cells expressing p16^{INK4A} did not undergo proliferation.

Firstly, we analyzed the GSE178318 public single-cell RNA-sequencing dataset, which includes colorectal cancer patients' samples, to strengthen our findings in single cell level. By comparing mRNA expression patterns between young and old in normal colon tissue, we found that, similar to our immunohistochemistry analysis, the number of fully senescent cells (p16^{INK4A}-positive cells) in the epithelium and stroma (fibroblasts and smooth muscle cells) was not significantly different between young and aged normal colon tissues. However, we did observe a significant difference in the number of p16^{INK4A}-positive cells in the T-cell population. This observation was confirmed in multiplex immunohistochemistry analysis, suggesting that senescence of the immune system may be another critical aging process, which we plan to investigate in future studies. Additionally, we found that in stromal regions enriched with fibroblasts and smooth muscle cells, fully senescent cells were not significantly accumulated. This finding suggests that mid-old cells, not old cells, may accumulate in the stromal region. Therefore, we thought that the previously reported increase in p16^{INK4A} expression in bulk sequencing data is likely due to immune cell origins. We also attempted to observe this phenomenon in human lung tissue, but we were unable to find an appropriate public dataset that included young patients under 40 years of age because lung diseases generally have later onset than intestinal diseases.

We further validated our hypothesis using single-cell analysis with aged mice model that are publicly available (GSE132042). When we performed the analysis using 3, 18, and 24-month-old mice, we did not observe an increase in *CDKN2A* (p16^{INK4A}) expression in any cell group in the colon. Similar results were observed in the aged mice lung. The difference from human data is that *CDKN2A* (p16^{INK4A}) expression was observed to increase in T cells in the aged human colon, but not in T cells in the aged mice. These results from single-cell analysis using both human and mouse tissues indicate that fully senescent p16^{INK4A}-positive cells do not significantly increase in epithelial cells, fibroblasts, and smooth muscle cells in aged tissues. We added these data in **Supplementary Fig. S4 and S5**.

Secondly, to further validate the presence of senescent cells, we conducted senescence associated-β-galactosidase (SA-β-Gal) staining in fresh frozen normal GI tract and lung tissues. As you know, the utilization of SA-β-Gal staining in our study presents certain limitations due to its requirement for fresh frozen tissues. Unlike paraffin-embedded tissues, acquiring a substantial number of tissues for analysis

was unattainable, consequently rendering it infeasible to stratify the samples into distinct age groups, such as individuals below 40 years old and those above 75 years old. Nonetheless, we were able to procure a subset of GI tract and lung tissues, which were employed for SA- β -Gal staining. The analysis revealed that SA- β -Gal positive cells did not exhibit an increasing tendency according to aging (**Supplementary Fig. S1c**).

Thirdly, we fully agree that relying solely on p16^{INK4A} immunohistochemistry to detect fully senescent cells may not be appropriate. Therefore, as recommended by the reviewer, we stained p16^{INK4A} with another representative senescence marker, p21^{Waf1} *in vivo* and *in vitro*. Our *in vitro* findings showed that p21^{Waf1} is expressed in both mid-old and old cells, making it another good option to detect fully senescent cells when co-stained with p16^{INK4A}. To this end, we performed double immunofluorescence staining analysis to simultaneously detect p21^{Waf1} and p16^{INK4A} double-positive cells *in vivo*. Notably, when p16^{INK4A} was stained, p21^{Waf1} was also observed to be stained. However, it should be noted that not all cells with p21^{Waf1} staining were positive for p16^{INK4A}. These data providing evidence that p16^{INK4A}-positive cells are indeed senescent cells. We added these data in **Supplementary Fig. S1b**.

Lastly, to confirm the non-proliferative nature of p16^{INK4A}-positive cells, we performed double immunohistochemistry staining with p16^{INK4A} and Ki67 *in vitro* and *in vivo*. The results unequivocally showed that cells expressing p16^{INK4A} did not co-express Ki67, indicating their lack of proliferative activity. These findings have been included in **Supplementary Fig. S1e and S6**.

Additionally, we have quantified the Histone H3K9-trimethylation IHC staining and added it to supplementary Fig. S1b.

We hope that it addresses the concerns raised by the reviewer.

2. Fig. 2 and Supp Fig 4. Based on my experience with fibroblasts, mid-old cells (DT5-DT7 days) will contain a percentage of young/healthy cells, really mid-old cells and senescent cells. Again, the assessment of senescence results deficient, as p16/Ki67 or p21/ki67 should be determined in co-staining but not separately (H3K9m3 or HP1gamma would help as heterochromatin marker too). I consider that EdU incorporation assays would be a more refined methodology to assess DNA replication vs Cell Cycle arrest and also to determine the heterogeneity of cell populations across the criteria of the authors to define young, mid-old and old cells.

Response 2: Thank you for your very important comments. As you recommended, we conducted double immunocytochemistry (p21^{Waf1}; green/Ki67; red, p16^{INK4A}; green/Ki67; red) on young, mid-old, and old cells. The data showed that cells expressing p21^{Waf1} or p16^{INK4A} had lower Ki67 expression. Old cells demonstrated p16^{INK4A} expression, however, mid-old cells did not express p16^{INK4A}; instead, they expressed p21^{Waf1}. Furthermore, old cells demonstrated more pronounced upregulation of p21^{Waf1} expression (high intensity in old cells; **Supplementary Fig. S6c upper middle panel**). Additionally, it was observed that in some mid-old cells, p21^{Waf1} expression was absent while Ki67 expression was present. As the reviewer correctly pointed out, this experiment suggests a heterogeneity in cell populations among the mid-old cells. This means that within the DT5 day mid-old cell population, there may be cells with a DT of less than 5 days, as well as cells in a more youthful state. We also had the same concern as the reviewer. We questioned whether the observed increase in proliferation was due to the response of young cells belonging to the mid-old cell population to SLIT2. Therefore, we further investigate whether young cells present in the mid-old cell population respond to SLIT2, we treated

young cells with rhSLIT2. rhSLIT2 did not induce proliferation or any other observable effects in young cells. The data was presented in **Supplementary Fig. S19a**. These data suggest that the cell proliferation mediated by SLIT2 in the mid-old cell population indicates that it was the mid-old cells themselves that underwent proliferation, rather than proliferation of young cells within the mid-old cell population. Additionally, we performed EdU incorporation in young, mid-old, and old cells.

We have included these data in **Supplementary Fig. S6a, S6b, S6c and S19a**.

3. Figure 3. The authors test SAA1 and SLIT2 in colon and lung samples, based on the transcriptomic results of Figure 2 with fibroblasts. But, these fibroblasts are human dermal fibroblasts isolated from foreskin, which do not seem to be an appropriate model. For a lung model the authors should use IMR90 or WI-38 pulmonary fibroblasts, which are well-characterised as a model of senescence in response to multiple triggers.

Response 3: Thank you for your detailed response. We fully concur with the reviewer's comments. The generation of mid-old IMR90 cells required a substantial amount of time due to their high proliferative capacity. Fortunately, we succeeded in creating mid-old cells with a doubling time of over the 4 days through continuous subculture. Similar to skin fibroblasts, these mid-old IMR90 cells showed increased expression of IL1 β , SAA1, and MMP9, alongside elevated p21^{Waf1} expression. Moreover, treating young IMR90 cells with rhSAA1 resulted in augmented MMP9 expression (**Fig. 5d**). Furthermore, the administration of rhSLIT2 (100ng/ml) to mid-old IMR90 cells (with a doubling time of 4 days) led to decreased expression of IL1 β and SAA1 (**data below**). These data suggest that IMR90 cells exhibit properties similar to those of skin fibroblasts. We have included these data in **Fig. 5d**. We have also utilized the mid-old IMR90 cells for a lung mimic *in vitro* model. These cells demonstrated behavior similar to that of fibroblasts, with mid-old IMR90 cells enhancing SCNN1A expression in HPAEpiC. Consequently, we have revised **Fig. 5f**.

4. Figure 4. Figure 4b and 4c indicating higher levels of MMP9 upon rhSAA1 should be validated by Western Blot and/or ELISA. Also, do mid-old or old fibroblasts and smooth muscle cells express more mRNA levels of MMP9? In addition, the design of the co-culture of Figure 4i and 4j does not seem to be representative of colon tissue, as the authors appear to use human dermal fibroblasts and a colon cancer cell line (SW480) rather than epithelial cells (HCoEpiC). Then, the link that the authors make with NH3 protein expression in mouse and human samples in Figure 4K is an over interpretation rather than a demonstration: “As a result, increased water reabsorption and smooth muscle atrophy of muscular mucosa can impair colonic function in the elderly, which could be due to SAA1 expressed in

mid-old fibroblasts”.

Response 4: Thank you for your insightful comments. We completely agree with the reviewer's suggestions. To address these issues, we utilized ELISA (**Fig. 4b, 4c**) and Western blotting (**data below**) to quantify MMP9 protein expression. We have incorporated this data into **Fig. 4b, 4c**. Furthermore, we evaluated MMP9 expression in mid-old and old fibroblasts in comparison to young cells and found a notable upsurge in MMP9 expression in both mid-old and old fibroblasts. However, MMP9 expression was more prominent in mid-old cells than old cells. We've added this information to **Supplementary Fig. S12b**.

In response to the reviewer's comments, we replaced SW480 cells with HCoEpiC in our experiments. Even though the morphology of HCoEpiC did not change significantly, we noticed an increase in APQ3, NHE2, and NHE3 gene expression when co-cultured with mid-old cells. We've presented these findings in **Fig. 4j**. Finally, in compliance with the reviewer's suggestion, we have removed the speculative sentence, “As a result, increased water reabsorption and smooth muscle atrophy of muscular mucosa can impair colonic function in the elderly, which could be due to SAA1 expressed in mid-old fibroblasts.”

5. Figure 5. Following my comments above, experiment 5e should have been done with young and mid-old pulmonary fibroblasts, but not with human dermal fibroblasts. Also, the increased fold induction of SCNN1A seems to be small. Is it also increased in old fibroblasts?

Response 5: In response to the reviewer's comment, we have included data from experiments using old cells. In the colon mimic model with old fibroblasts, we noted an increase in NHE3 expression. However, unlike in mid-old cells, we did not observe an increase in AQP3 and NHE2 expression when using old cells. The reason for this discrepancy is not entirely clear, but it may be due to lower MMP9 expression in old cells compared to mid-old cells. We added these data in **Supplementary Fig. S15b**.

Following the reviewer's suggestion, we have also included data using mid-old IMR90 pulmonary fibroblasts. We have updated this information as new data in **Fig. 5f**. As you mentioned, the change of SCNN1A expression seems to be small. We considered the reason behind the minimal expression of SCNN1A. While we cannot provide a comprehensive explanation, our understanding is that SCNN1A is primarily expressed in type II alveolar cells. However, commercially obtaining solely type II alveolar cells is not feasible, and the primary alveolar epithelial cells (HPAEpiC) we utilized encompass both type I and type II alveolar cells. Consequently, we thought that the observed phenomenon is a result of the significant presence of type I alveolar cells in our samples. Similar findings were also observed when using IMR90 cells, similar to the observations made with skin fibroblasts. Although our explanation may be insufficient, we hope that it addresses the concerns raised by the reviewer.

Furthermore, despite our intention to investigate the fully senescent state using IMR90 cells, we did not observe a transition to a fully senescent phenotype, as IMR90 cells demonstrated vigorous proliferation. As a result, our research was restricted to the examination of mid-old cells, and we were unable to conduct experiments with fully senescent IMR90 cells, as presented in the colon mimic model in Supplementary Fig. S15b. We kindly request the reviewers' understanding regarding this limitation.

6. Figure 6. As stated above, EdU incorporation assays would be more informative and quantitative in Figure 6b. Could the authors provide GSEA of pathways of DNA replication and Cell Division in Figure 6c?

Response 6: In response to the reviewer's comment, we have incorporated the EdU incorporation assay into **Fig. 6c**. In addition, per the reviewer's recommendation, we have included Gene Set Enrichment Analysis (GSEA) in **Fig. 6d and 6e**. Our co-culture data reveals that proteins derived from young cells exert an influence on mid-old cells. Notably, the expression of genes related to cell cycle, DNA replication, and extracellular matrix (ECM) production was enhanced, while the expression of genes associated with inflammation was downregulated.

7. Figure 7. A more concluding and convincing validation would require to generate SLIT2 KO young fibroblasts and mid-old fibroblasts and then supplement the culture with rhSLIT2, while proliferation assays and assessment of inflammatory genes and, most importantly, secreted factors at the level of protein is done (which is currently missing). In fact, the decreased mRNA expression levels of inflammatory genes upon Lenti-SLIT2 in Fig.7c seem really marginal.

Response 7: Thank you for your insightful comment. As recommended, we performed knockdown experiments on SLIT2 in young fibroblasts again and examined the expression of inflammatory genes in mRNA and protein level. We generated two kinds of shSLIT2 lentivirus and infected into young fibroblasts, which led to an upregulation in the expression of inflammation-related genes, including IL1 β and SAA1. We further confirmed these findings using ELISA analysis and included these data in **Fig. 7g**. Infection of shSLIT2 lentivirus into young fibroblasts resulted in an upregulation of inflammation-related gene expression and a concomitant decrease in cell proliferation. This decrease in cell proliferation is supported by findings from cell counting studies and EdU incorporation assays. We added these data in **Fig. 7h**.

Next, we treated the rhSLIT2 protein (100ng/ml) back into the shSLIT2 lentivirus infected young fibroblasts and analyzed the expression of inflammation-related genes *via* real-time PCR. The data showed a downregulation in the expression of these genes. However, rhSLIT2 did not fully rescue the effect of shSLIT2. This data has been added to **Fig. 7i**. Furthermore, our data on SLIT2 downregulation in young cells co-cultured with mid-old cells demonstrated a decrease in cell proliferation. We verified this using cell proliferation and EdU incorporation assays (**Fig. 7j**).

Overexpression of SLIT2 in mid-old fibroblasts was observed to be linked with decreased expression of various inflammation-related genes, as well as reduced p21^{Waf1} expression. In the previous version of the manuscript, only mRNA expression was presented. However, in the revised version, the protein levels of IL1 β and SAA1 were measured using ELISA and included in the presentation of results. We added this data in **Fig. 7c**.

8. Figure 8. The authors show some evidence that the administration of rmSLIT2 increased activity in aged mice. However, a direct (rather than circumstantial) role in organ rejuvenation, if any, is missing. There are not functional tests (e.g. spirometry or gas transfer test) and the SCNN1A immunostaining, or NHE3 or Sox2 immunostaining, represent minor evidence. Also, the effect size is low. Many questions remain pending, for example if the treatment expands the lifespan of the mice or whether it protects mice from damage/inflammation (e.g. mice subjected to irradiation).

Response 8: Thank you for your very important comment. We fully agree with the reviewer's concern. We could not provide direct evidence of rejuvenation after treating rmSLIT2. To address this issue, we conducted additional animal experiments. To minimize animal-to-animal bias, we re-experimented using 23-month-old male mice and analyzed the effects of rmSLIT2 treatment before and after administration. In addition, we also conducted experiments using 4-month-old mice as a comparison. We have updated **Fig. 8** and **Supplementary Fig. S21, S22 and S23**.

As the reviewers are well aware, conducting experiments using aged mice has its limitations. In general, commercially available mice older than 1 year of age are not readily obtainable. Hence, we acquire less than 1-year-old mice and rear them in our research laboratory for approximately 1 year to generate aged mice for our experimental purposes. Thankfully, we were able to procure 23-month-old male C57BL/6N mice (n=18) (previously we used 22 months old C57BL/6J mice; Supplementary Fig. S20). We utilized a total of 18 mice for two separate studies; rejuvenation (n=5 PBS, n=5 rmSLIT2) and longevity (n=4 PBS, n=4 rmSLIT2, as suggested by reviewer 2) experiments. Although the number of mice used for both studies may not have been optimal, we made every effort to maximize our resources. Despite being the same strain, C57BL/6N mice displayed a more youthful phenotype compared to C57BL/6J mice, including an absence of significant hair loss usually observed in aged C57BL/6N mice. We evaluated various factors including mouse activity, hanging endurance, muscle weight, muscle cell size, body weight and conducted blood analysis. The results indicated an improvement in most indices within the rmSLIT2 treatment group. Notably, leg muscle mass, including the tibialis anterior and gastrocnemius muscles, was significantly increased in mice treated with rmSLIT2 (**Fig. 8c**). These results suggest that SLIT2 may have a role in muscle maintenance. Previous publications, along with our findings, suggests that SAA1 induces skeletal and smooth muscle atrophy. Therefore, it is plausible that muscle preservation might be achieved through the SLIT2-mediated inhibition of SAA1. Consequently, hanging endurance and cage activity were improved (**Fig. 8a, 8b**). Moreover, rmSLIT2 treatment reduced the number of mid-old cells in old-aged mice, with a decrease in p21^{Waf1}-positive stromal cells number observed in the esophagus, stomach, colon, and lung, while an increase in SOX2 positive stromal region cells number was observed in rmSLIT2 treated mice (**Fig. 8d, 8e**).

In the previous experiment, the expression of SOX2 and SCNN1A IHC data did show minimal change in the mice treated with rmSLIT2. We also extensively pondered the reasons behind the minimal effect of rmSLIT2 in mice. Ultimately, we hypothesized that this could be attributed to either the single administration dose or the duration of the treatment. However, our *in vitro* data (Fig. 7) revealed that a prolonged treatment period is necessary to observe the effects of rmSLIT2. Therefore, in second *in vivo* mouse experiment, we treated the mice in relatively long-term with 10 rounds of rmSLIT2 compared to the previous experiment (6 rounds) and observed an increase in SOX2 expression in various organs and downregulation of p21^{Waf1}, NHE3 and SCNN1A expression in the lung (**Fig. 8d, 8e and Supplementary Fig. S23a, 23b**).

We appreciate the reviewer's suggestion regarding the use of a spirometer. However, due to limitations in our resources, we were unable to utilize a spirometer to measure lung function of mice. Instead, we attempted blood gas analysis (**Supplementary Fig. S21**). However, due to technical challenges, we analyzed venous blood rather than arterial blood. We deeply understand that this does not accurately reflect the concentrations of pO₂ in the body. Nevertheless, in our analysis, we did not observe any significant differences between the rmSLIT2-treated group and the non-treated group. And we observed that there was no significant difference in the anion gap between rmSLIT2 treated and non-treated group, suggesting the absence of acidosis in the body. This implies that there is no substantial difference in lung or kidney function. Multiple previous studies [Schulte et al., *Front. Physiol.* (2019) and Yazicioglu et al., *Am. J. Physiol. Lung Cell Mol. Physiol.* (2020)] demonstrated that age-related structural and functional alterations in the mouse lung are primarily caused by micromechanical changes, resulting in increased lung parenchymal volume and alveolar airspace. Consequently, aged lungs already exhibit changes in the ECM components in their microenvironment, such as fibrosis, although the exact mechanism of this phenomenon is still not fully understood. However, the number of type II alveolar cells, which is related to surfactant release and differentiation into type I alveolar cells, remains unchanged between young and old mice. Therefore, it is plausible that rmSLIT2 treatment could be resistant to these changes in ECM constituents rather than cellular changes. This observation is consistent with our findings in the rmSLIT2-treatment model in mice, where we observed no functional changes in lung tissues (no significant differences in pCO₂ levels and anion gap in the blood). While we were unable to provide direct evidence, we hope that this addresses the concerns raised by the reviewers.

Our data demonstrate that administering rmSLIT2 to aged mice significantly boosts their activity level (**Fig. 8a**). At this point, it remains unclear whether this effect is a consequence of rmSLIT2 positively affecting all organs or simply due to its role in maintaining muscle mass, warranting further investigation. While we have not been able to unravel the exact mechanism, we have observed that the administration of rmSLIT2 leads to a reduction in mouse body weight (**Fig. 8a**) and a decrease in intra-abdominal fat upon opening the abdominal cavity (**data below**). In our first experiment, we didn't observe a decrease in body weight in the rmSLIT2 treated group, likely because the treatment was only administered for short-term (6 rounds). However, in the second experiment, there was also no significant weight loss observed during the first three-week treatment period, but after five weeks of treatment, weight loss was evident (significant body weight reduction was observed compared to the start point of rmSLIT2 treatment, and when compared to the PBS group, a trend of body weight reduction was observed; as confirmed by two-tailed Mann-Whitney U test analysis). This suggests that rmSLIT2 could potentially influence fat cell reduction, lipolysis, or beta-oxidation, all of which are intriguing possibilities that warrant further investigation. In conclusion, we speculated that the maintenance of muscle mass and reduction in body weight contribute to increased activity levels and elevated hanging ability in mice.

We also evaluated the activity of 4-month-old young mice and found that they had a higher basal activity compared to old-aged mice. However, treating 4-month-old mice with rmSLIT2 for 10 times did not result in any significant difference in activity between the rmSLIT2 treated and non-treated young-aged mouse groups (**Supplementary Fig. S22**).

Additionally, in response to your suggestion, we investigated the potential of SLIT2 to induce an increase in lifespan. Unfortunately, we encountered a supply issue with aged mice, resulting in a limited number of mice available for the experiment (rmSLIT2, n = 4; PBS, n = 4). Consequently, we proceeded with the experiment using 23-month-old male mice, administering either PBS or rmSLIT2 (2 µg/mouse) once a week. However, we did not observe a difference in lifespan during the 22-week duration until 28.5 months of age (in each group, three mice have died so far, Survival curves Log-rank Test p=0.519). Two major reasons may explain this phenomenon. Firstly, the limited number of mice used in this study may have influenced the outcome. Secondly, these results suggest that SLIT2 may be associated with functional rejuvenation rather than extending lifespan at both cellular and individual levels. Nevertheless, we are unable to draw a conclusive statement at this stage. We firmly believe that further experiments should be conducted with a larger number of mice to obtain more definitive and reliable results.

We hope this response satisfactorily addresses your question.

[Other relevant points]

1. There is lack of relevant and more detailed information regarding the clinical samples. A Table is missing including ages, pathological conditions, treatments, etc. Concerningly, the samples seem to come from surgically resected specimens from cancer patients. Even if the tissues were dissected at 5-12 cm from the pathologic regions the physiological tissue microenvironment might be strongly impacted by the clinical conditions of the patient and treatments (e.g. neoadjuvant chemotherapy or radiotherapy).

Response 1: Thank you for your substantial comment. We have included a table detailing the clinical tissue samples used in this study in **Data S4**.

We utilized normal tissues harvested from areas 5-12 cm distant from the lesion in patients. Importantly, only tissues from patients who hadn't received chemotherapy or radiation therapy prior to surgery were used to identify the presence of senescent cells. We are well aware of the complexities mentioned by the reviewer, and we completely concur with your concerns.

As it is widely acknowledged, ethically obtaining normal tissues from individuals who do not have any diseases can be challenging. While designing this study, we contemplated sourcing tissues from individuals who had perished due to accidents, such as traffic incidents. However, we recognized that such tissues may not represent normal conditions since cytokines and other molecules could have been released in significant amounts preceding death. Consequently, we concluded that the most representative normal tissue would be those surgically removed from locations furthest from the lesion. We conducted our study based on this reasoning. We hope this explanation sufficiently addresses your concerns.

2. The quality of the IHC images is poor across the manuscript (Figs. 1, 3, 4, 5 and 8, and Suppl Fig 1, 3, 8 and 9), and convincing magnifications are missing. It is very difficult for the reader to discern between positive or negative cells for the particular markers.

Response 2: We deeply apologize for the poor quality of the images in our previous manuscript. We believe that the conversion of the images to PDF format led to a decrease in their quality. In the revised version, we have included high-resolution figures to address this issue. Furthermore, we have inserted high-magnification images to enable better observation of positive cells. We changed the figure in **Fig. 1, Fig. 3, Fig. 4, Fig. 5, Fig. 8, Supplementary Fig. S1, S3, S4, S9, S10, S11, S20, S23.**

3. Sometimes the criteria of the authors seem biased. For example, the quantification of number of tissues in the graphs, where protein expression status was classified as grade 0, 1+ or 2+. Overall, the number of positive events (positive cells or stained surface) seem very low or marginal, partially explaining the criteria of the authors. In my opinion, a more objective quantification analysis might be obtained by analyzing the relative percentage of positive cells for the representative images. For the reproducibility of the data, all quantifications should have been done in an automated manner. Also, I don't think the calculation of the *p* value by using one-way ANOVA is the most refined approach, but a two-tailed student's *t*-test instead, that might be subjected or not to a correction (e.g. Welch's correction).

Response 3: In response to the reviewer's comment, we have presented the number of immunostaining-positive cells of p21^{Waf1}, SAA1, SLIT2, IL1 β , MMP9, and SCNN1A in percentage rather than categories (**Fig.1, Fig. 3, Fig. 4, Fig. 5, and Fig. 8**). IHC results were presented as the percentage of positive cells in the stromal or epithelial region of three random areas. In the case of type IV collagen and NHE3, an experienced pathologist evaluated the intensity of immunostaining at the region and graded it as weak, moderate and strong (**Fig. 4g, 4k, Fig. 5b, and Supplementary Fig. S23a**). Additionally, we have included the measurement method in the **Materials and Methods** section.

Regarding the concern related to the statistical method for calculating *p* values, we followed the recommendation of the reviewer. For samples with $N < 30$, which did not meet the assumption of normal distribution, we used the Mann-Whitney U test instead of ANOVA or Student's *t*-test. On the other hand, for experiments with $N > 30$, we employed the Student's *t*-test to calculate the statistical differences. Thank you for bringing this up, and we appreciate your thorough review.

4. Figure 2. Could the authors confirm whether the genes presented in Figure 2b are differentially expressed with statistical significance? What is the criteria for the presented list of inflammatory/anti-inflammatory genes? This is not a defined dataset pathway but rather a "partial" list of genes.

Response 4: In alignment with the reviewer's observations, we recognize that the gene list we initially utilized to investigate functional changes in fibroblasts may be somewhat limited and biased. To address this concern, we shifted our analysis to Gene Set Enrichment Analysis (GSEA) rather than employing heatmaps to display gene expression relating to the inflammatory response, extracellular matrix formation, and tissue reorganization (**Fig. 2d**). Although the overall inflammatory response of mid-old cells did not differ from young cells, the specific inflammatory pathway of IL1 β was upregulated in mid-old cells. Additionally, SAA1, an acute response protein, was identified in the IL1 β pathway as a leading-edge gene. These findings suggest that age-related inflammation may be primed by the IL1 β

pathway. However, the degree of inflammation was highly upregulated in old cells compared to young and mid-old cells, with TNF α and IL6-related pathways being major inflammatory responses in old cells (**Fig. 2h**). However, we faced a challenge in identifying suitable gene sets for GSEA related to anti-inflammation, due to the lack of comprehensive gene sets that cover both anti-inflammatory cytokines and chemokines. In order to circumvent this issue, we amalgamated the GSEA gene set titled "GOBP: Negative Regulation of Chemokine Mediated Signaling Pathway", which encompasses anti-inflammatory chemokines, with reference genes from a prior paper [S. M. Opal and V. A. DePalo, *Chest* (2000)], which incorporates anti-inflammatory cytokines. Real-time PCR, performed using the full list of referenced genes, revealed a declining trend in anti-inflammatory genes as senescence advanced (**Fig. 2j**). Simultaneously, we noted that SLIT2, a molecule known to inhibit NF κ B, decreased in expression over time (**Fig. 2j**). Therefore, we concentrated on SLIT2 for subsequent examination of its anti-inflammatory role and potential in promoting functional recovery of mid-old cells.

5. Supp Fig. 16. The authors assume that factors restoring the proliferative capacity of mid-old cells should be p53-p21 pathway inhibitors but why not p16-Rb pathway inhibitors?

Response 5: Thank you for your feedback. We propose that the slowdown of the cell cycle in mid-old cells is mediated by the p53-p21^{Waf1} pathway, as shown by our *in vitro* western blotting and gene expression analyses presented in **Fig. 2a**. In contrast, we do not detect any activity in the p16^{INK4A} signaling pathway in mid-old cells, as confirmed by our western blotting, real-time analysis, and immunocytochemistry data. Consequently, we suggest that the recovery of proliferative ability in mid-old cells may be due to the inhibition of p21^{Waf1}.

6. Figure 6d. The “shortlisting” of Lnc-SBLC, Lnc-SAL RNA1 and Lnc-ROR is unclear. Why do the authors focus on these particular LncRNAs?

Response 6: Thank you for your significant comment. We are indeed interested in identifying secretory products from young cells for potential rejuvenation factors. In the initial planning phase of this study, we classified secreted factors from young cells into three categories: metabolites, RNAs, and proteins. Our analysis of metabolites did not yield any compounds identifiable as associated with rejuvenation (**Supplementary Fig. S17**). We then shifted our focus to RNAs, specifically long noncoding RNAs (LncRNAs), given that three different lncRNAs were reported to have anti-aging effects in previous publications. While we observed the expression of these LncRNAs in young cells, only Lnc-SBLC was found to be secreted *via* exosomes in our experiment (**Fig. 6g**). We undertook further studies on Lnc-SBLC, but as mentioned earlier, RNA products had minimal involvement in cell proliferation and did not involve in anti-inflammation, therefore, it was not considered as major regulators. Therefore, we inferred that proteins were likely the critical rejuvenation factors, which led us to our subsequent study on SLIT2. We hope this explanation addresses your concerns.

Reviewer #1 (Remarks to the Author):

The authors have addressed several comments but did not adequately address others. Despite this shortcoming, I remain very enthusiastic about the central new and interesting findings of the study: (1) the discovery of cells with a "mid-old" phenotype accumulate in tissues and organs; (2) that these cells can impact the tissue microenvironment; (3) that SLIT2 supplementation can revert the mid-old cellular phenotype; and (4) that SLIT2 supplementation counters age-related decreases in activity and muscle mass.

To warrant publication, the authors should more fully address the key concerns that were pointed out. Additionally, I have strong reservations as to the authors' over-interpretation and "hyping" of their data.

Main remaining concerns:

1) While the authors suggest a slowing of sarcopenia and an increase in overall activity but were unable to provide the requested comprehensive, in-depth analysis of aging-associated phenotypes. In the absence of such analyses, it is premature to claim that SLIT2 is a "rejuvenation" factor. What would be justified based on the current data is the conclusion that the mid-old cellular phenotype is reversible, and that this reversal is associated with a slowing or reversal of a few phenotypes associated with aging.

2) As mentioned in the initial review, there is a large body of evidence that p16-positive cells accumulate with aging. Early mouse data by many labs has more recently been confirmed using a p16 reporter mice (Omori et al 2020). I believe that the fact that the authors cannot reproduce firmly established mouse data suggests that their methods are less reliable and more superficial. Indeed, the authors did do the requested co-staining of p16 with p21, but did not do this for all tissues and did not do any quantitation of double positive (senescent) cells. Without such quantitation the p21-p16 double staining is not informative. Furthermore, it is widely accepted that SA-beta-Gal is not a reliable in vivo senescence marker for senescence as many non-senescent cells are beta-Gal positive. If the authors remain of the opinion that it is important to state that p16-senescent cells do not increase with aging in tissues and organs, they will have to provide compelling data to justify this statement.

3) The use of the term rejuvenation is pretentious and not fully justified by the data provided. Why not simply focus on the reversibility of the mid-old cell phenotype and the limited in vivo phenotypic improvements that result? This includes the selection of a more appropriate title and an abstract focusing on the core new findings.

Reviewer #2 (Remarks to the Author):

This is now the revised version of the script entitled "The Elderly's Organs are Ready for Functional Rejuvenation" by Kim et al. The authors have done a comprehensive revision of the manuscript and addressed many of my concerns. In doing so, the revised manuscript is significantly improved in contents and quality of the figures, and now includes better supported conclusions and mechanistic insights.

There are some additional points the authors will need to address:

1. Overall, the authors should tone down the conclusions in the whole manuscript when referring to fully senescent cells and mid-old cells. Distinction between fully senescent cells and mid-old cells are based on the criterion of biomarkers/read-outs used by the authors, which is now more solid but still incomplete in terms of senescence assessment (e.g. SAHF, macromolecular damage or deregulated metabolism regulation, all of them hallmarks of senescence as per Gorgoulis et al. Cell, 2019). Senescence is just a word to describe a very heterogeneous and dynamic process

depending on cellular types, triggers, context, etc. It must be considered that p16 is not a universal (although common) biomarker of senescence and the same applies to SABGAL activity. There are non-senescent cells presenting either p16 expression or SABGAL activity (e.g. osteoclasts). In fact, both makers are often difficult to be assessed in human tissue because of the specificity/sensitivity of the existing antibodies (p16) or because of the requirement of fresh tissue (SABGAL activity). Therefore, I would not be surprised if "mid-old" cells are cells in the dynamic process to be bona fide senescent cells ("mid-senescent" cells) or even another senescent status. In fact, mid-old cells are featured by at least some of the main hallmarks of senescence (1. Expression of cell cycle inhibitors, such as p21, and 2. Secretory phenotype enriched in proinflammatory factors such as IL1B and extracellular matrix remodelling factors, such as MMP9). In other words, p21 and IL1B are very common senescence biomarkers. I think these limitations on how we categorise "aberrant" cellular ecosystems in old tissues should be incorporated into the Discussion.

2. Figure 8. The contents of the Figure have been improved, but the robustness of the conclusions is still very limited. For example, treatment with rmSLIT2 results in a reduction of the mouse body weight (Fig. 8a) and this might be due to a beneficial effect associated to increase activity mice in cages, but also it might just be a weight lost due to toxicities of recombinant protein. The title of the Results section – SLIT 2 as a master rejuvenation factor – must be toned down (e.g. SLIT 2 as a potential rejuvenation factor). Also, the authors must recognise the important limitations of Fig. 8 in the Discussion, including for healthspan, rejuvenation and lifespan, as it is done in the rebuttal letter.

In addition, the validation of p21 in Fig. 8d should be improved or some of the panels modified. The IHC signal of functional p21 should be nuclear, similarly to the SOX2 signal nicely presented in Fig 8e. Could the authors please double check this and reanalyse the quantifications, if required? It is important to ensure they are not quantifying an IHC artefact in some of the fields.

3. In addition to the intriguing mechanistic/functional differences between fully senescent cells and mid-old cells, this study opens some interesting questions. For example, whether mid-old cells can favour their immunoclearance by secreting inflammatory factors or factors associated to inflammatory processes (e.g. IL1B or SAA1) or rather immunosuppression. Also, how do mid-old cells apply to pathological conditions and diseases (e.g. Are they increased in diseased tissues? Are they part of the tumour microenvironment? Is this a process implemented by cancer cells in response to therapy and resulting in cancer resistance?). The reversibility of the process and potential re-entering to the cell cycle is another question that warrants further studies. Perhaps some of these questions are also food for Discussion and a good opportunity for future works.

I congratulate the authors for the exhaustive revision.

Reviewer #1:

The authors have addressed several comments but did not adequately address others. Despite this shortcoming, I remain very enthusiastic about the central new and interesting findings of the study: (1) the discovery of cells with a “mid-old” phenotype accumulates in tissues and organs; (2) that these cells can impact the tissue microenvironment; (3) that SLIT2 supplementation can revert the mid-old cellular phenotype; and (4) that SLITS2 supplementation counters age-related decreases in activity and muscle mass.

To warrant publication, the authors should more fully address the key concerns that were pointed out. Additionally, I have strong reservations as to the authors’ over-interpretation and “hyping” of their data.

Response: We are grateful to the Reviewer for providing us with the opportunity to revise our manuscript. Taking into account the reviewer's comments, we have made an effort to modify our overinterpreted statements and adopt a more restrained tone. We believe that the revised content adequately addresses the concerns raised by the reviewer.

Main remaining concerns:

1 While the authors suggest a slowing of sarcopenia and an increase in overall activity but were unable to provide the requested comprehensive, in-depth analysis of aging-associated phenotypes. In the absence of such analyses, it is premature to claim that SLIT2 is a “rejuvenation” factor. What would be justified based on the current data is the conclusion that the mid-old cellular phenotype is reversible, and that this reversal is associated with a slowing or reversal of a few phenotypes associated aging.

Response: Thank you for Reviewer’s valuable comments. As the reviewer indicated, it is premature to categorize SLIT2 as a rejuvenation factor, and further research should be conducted. Based on our current data, the conclusions we are presenting involve delaying muscle mass reduction and the reversal of mid-old cell phenotypes, such as p21^{Waf1} expression, in aged mouse. Therefore, we have revised the word "rejuvenation" from the manuscript and replaced it with "delay or anti-aging" in the **Title, Abstract, Results and Discussion** session.

2 As mentioned in the initial review, there is a large body of evidence that p16-positive cells accumulate with aging. Early mouse data by many labs has more recently been confirmed using a p16 reporter mice (Omori et al 2020). I believe that the fact that the authors cannot reproduce firmly established mouse data suggests that their methods are less reliable and more superficial. Indeed, the authors did do the requested co-staining of p16 with p21, but did not do this for all tissues and did not do any quantitation

of double positive (senescent) cells. Without such quantitation the p21-p16 double staining is not informative. Furthermore, it is widely accepted that SA-beta-Gal is not a reliable *in vivo* senescence marker for senescence as many non-senescent cells are beta-Gal positive. If the authors remain of the opinion that it is important to state that p16-senescent cells do not increase with aging in tissues and organs, they will have to provide compelling data to justify this statement.

Response: Thank you for the valuable feedback. We apologize for not adequately addressing a point raised by the Reviewer in the previous version of the paper. In our previous manuscript, we presented the single-cell RNA-sequencing data of human and mouse colon. However, we acknowledge that these approaches might not have sufficiently addressed the reviewer's concern. To address this issue, we would like to include the following additional data and provide further explanations.

Finding appropriate markers for senescent cells has been a major challenge in senescence research. Although p16^{INK4A} and p21^{Waf1} are commonly used, we also agree that they don't solely reflect the senescence status. p21^{Waf1}, along with p53, is also involved in the early DNA damage response to external stresses. Therefore, p21^{Waf1}-positive cells may include both senescent and damaged cells in the p21^{Waf1}-positive cell population. On the other hand, p16^{INK4A} can give false positives due to viral infections and insufficiently halt cell proliferation, leading to inaccuracies in senescent cell analysis. Similarly, another widely-used senescence marker SA-β-Gal can also show false positives based on acid-base conditions and certain cell types. Despite these limitations, we believe that p16^{INK4A} remains currently the most reliable marker for detecting the later stages of cellular senescence *in vivo*. Recent transgenic mouse models, such as p16^{INK4A}-tdTomato reporter or the p16^{INK4A}-3MR system, are based on the *CDKN2A* gene rather than other senescence-related genes to address these challenges.

In this paper, we are not trying to deny the broadly accepted concept that the total number of p16^{INK4A}-positive cells increases in both human and aged mice tissue. Several studies have noted an increase in the number of p16^{INK4A}-positive cells in older mice models (Baker et al., *Nature*, 2011, Liu et al., *PNAS*, 2019, and Omori et al., *Cell Metab*, 2020). This aligns with reports indicating a rise in these cells in aged tissues. Hence, we concur that there is indeed an increase in the number of p16^{INK4A}-positive cells in aged tissues. However, despite Omori et al. (*Cell Metab*, 2020) showing an overall increase in tdTomato-positive cells across various organs using a p16^{INK4A}-tdTomato reporter mouse, they did not specify the exact cell types that constituted the tdTomato-positive cells. Additionally, although Grosse et al., using a p16-CRE/R26-mTmG mouse, revealed that CD31-positive endothelial cells and F4/80-positive macrophages comprise the major population of p16^{INK4A}-positive cells in the liver, information about the remaining population and other organs is lacking. Based on these previous studies, we cautiously propose that the majority of the p16^{INK4A}-positive cells originate from circulatory systems, such as immune cells or vascular endothelial cells.

In fact, the emphasis of our work is on the specific cell types within senescent ecosystem. We have primarily focused on epithelial cells, fibroblasts, and smooth muscle cells, which are the major component cells of the organ rather than infiltrated immune cells. To address this issue, we performed further in-depth analysis of single-cell RNA sequencing data (GSE132042) from mouse tissues, as reported in Tabula Muris Consortium et al. (Nature, 2018). p16^{INK4A} expression across various tissues revealed that the changes in the number of p16^{INK4A}-positive cells, as observed with aging, are dependent on the organ type. When analyzing p16^{INK4A} expression in various tissues, we found that certain tissues (liver, tongue, heart, pancreas) showed an increase in p16^{INK4A} expression, while some others (aorta, bladder) showed a decrease. Furthermore, in most tissues, only minimal changes in p16^{INK4A} expression were identified. To gain further insights, we delved into specific cell types within representative tissues, such as lung, colon, skin, and liver, to assess their p16^{INK4A} expression levels. The results indicated a notable increase in p16^{INK4A} expression primarily in immune cells. On the contrary, the cells we primarily focused on, epithelial cells, fibroblasts, and smooth muscle cells, showed either no difference or only a slight increase in p16^{INK4A} expression. We added these data in **Figure S8 a-e**.

To validate these findings in human *in vivo* aged tissues, we attempted to perform single-cell sequencing analysis in human colon, lung, skin, and liver by searching for publicly available datasets. However, unfortunately, we could not find the enough number of patients in both young and old aged tissue samples. Nonetheless, we conducted an additional analysis of the normal regions of single cells in lung, colon, skin, and liver tissues with the limited number of patients. Although they did not align with the initially proposed criteria of old and young groups in the manuscript, we analyzed the lung tissue (51 vs. 75 years old), skin tissue (25 and 27 vs. 53, 69, and 70 years old), colon tissue (35, 35 and 42 vs. 81, 82 and 90 years old), and liver tissue (48 and 48 vs. 64 and 65 years old). We observed that p16^{INK4A}-expressing cells were predominantly found in immune cells rather than in epithelial cells, fibroblasts, or smooth muscle cells in the aged tissue.

In the case of lung data, datasets that include younger samples are scarce due to the late onset of the lung-related diseases. Nonetheless, we managed to find and analyze one dataset that compared samples from individuals aged 51 and 75 years old. The results showed a slight increase in p16^{INK4A}-positive cells (from 0.6% to 0.7%). However, mirroring our mouse findings, these p16^{INK4A}-positive cells were predominantly observed in immune cells, whereas smooth muscle cells and fibroblasts did not express p16^{INK4A}. We also investigated skin and liver tissues. In the skin tissue, when comparing samples from younger individuals (25 and 27 years old) to older individuals (53, 69, and 70 years old), there was a modest increase of approximately 0.1% (from 0.6% to 0.7%) in p16^{INK4A}-expressing cells. The majority of these cells were also identified as immune cells. Regarding liver tissue, our analysis of samples from individuals aged 48 (two individuals) and 64 and 65 years old revealed an increase in p16^{INK4A}-expressing cells, predominantly observed in Kupffer cells. This aligns with the earlier study by Grosse

et al., who used p16^{INK4A} reporter mouse models and found that hepatic F4/80-positive macrophage often exhibit signs of p16^{INK4A}-high phenotype in the livers of aged mice. As demonstrated in the previous version of the manuscript, the colon single-cell data showed that p16^{INK4A} expression is primarily seen in CD3-positive T cells.

Although our analysis was not extensive across a broad range of organs, we have cautiously drawn the following conclusion: with aging, there is an increase in p16^{INK4A}-expressing cells, but this increase is predominantly observed in immune and endothelial cells. Minimal changes were seen in epithelial cells, fibroblasts and smooth muscle cells. We've included relevant data in **Figures S4, S5, S6, S7, and S8** and further discussed these findings in the discussion section.

Moreover, according to the Reviewers' suggestions, we conducted immunofluorescence staining of p16^{INK4A} and p21^{Waf1} with vimentin in aged tissues. The analysis was performed in colon, liver, small intestine, skin, and lung tissues. Unfortunately, we could not analyze thyroid tissue due to the limited presence of stromal fibroblasts for examination. Triple immunofluorescence analysis [p16^{INK4A} (blue), vimentin (green), and p21^{Waf1} (red)] showed that although there were few p16^{INK4A} positive cells present in old-aged stromal tissues, almost p16^{INK4A}/vimentin positive cells were also positive for p21^{Waf1}, but not all p21^{Waf1}-positive cells exhibited p16^{INK4A} expression. These data suggest that fibroblasts in aged tissues show a mid-old cell status characterized by p21^{Waf1} expression rather than fully senescent cells. Relevant data has been included in **Figure S12b**.

Once again, we extend our gratitude to the reviewer for their insightful comments. We believe that addressing these queries has been an essential part of our task to enhance understanding for our readers. We appreciate your patience in reading our lengthy response, and we hope this answer alleviates some of your concerns.

3. The use of the term rejuvenation is pretentious and not fully justified by the data provided. Why not simply focus on the reversibility of the mid-old cell phenotype and the limited in vivo phenotypic improvements that result? This includes the selection of a more appropriate title and an abstract focusing on the core new findings.

Response: In accordance with the comments from the reviewer, we have revised the title and abstract. We have removed the term "rejuvenation" and corrected the content to focus on the reversibility of mid-old cells. We changed the title as "**Mid-Old Cells are Potential Target for Anti-aging Interventions in the Elderly**" And running title is "**Young cell originated SLIT2 is a new anti-aging protein**", respectively.

Reviewer #2:

This is now the revised version of the script entitled "The Elderly's Organs are Ready for Functional Rejuvenation" by Kim et al. The authors have done a comprehensive revision of the manuscript and addressed many of my concerns. In doing so, the revised manuscript is significantly improved in contents and quality of the figures, and now includes better supported conclusions and mechanistic insights.

There are some additional points the authors will need to address:

1. Overall, the authors should tone down the conclusions in the whole manuscript when referring to fully senescent cells and mid-old cells. Distinction between fully senescent cells and mid-old cells are based on the criterion of biomarkers/read-outs used by the authors, which is now more solid but still incomplete in terms of senescence assessment (e.g. SAHF, macromolecular damage or deregulated metabolism regulation, all of them hallmarks of senescence as per Gorgoulis et al. Cell, 2019). Senescence is just a word to describe a very heterogenous and dynamic process depending on cellular types, triggers, context, etc. It must be considered that p16 is not a universal (although common) biomarker of senescence and the same applies to SABGAL activity. There are non-senescent cells presenting either p16 expression or SABGAL activity (e.g. osteoclasts). In fact, both makers are often difficult to be assessed in human tissue because of the specificity/sensitivity of the existing antibodies (p16) or because of the of the requirement of fresh tissue (SABGAL activity). Therefore, I would not be surprised if "mid-old" cells are cells in the dynamic process to be bona fide senescent cells ("mid-senescent" cells) or even another senescent status. In fact, mid-old cells are featured by at least some of the main hallmarks of senescence (1. Expression of cell cycle inhibitors, such as p21, and 2. Secretory phenotype enriched in proinflammatory factors such as IL1B and extracellular matrix remodeling factors, such as MMP9). In other words, p21 and IL1B are very common senescence biomarkers. I think these limitations on how we categorize "aberrant" cellular ecosystems in old tissues should be incorporated into the Discussion.

Response: Thank you for your kind words. We appreciate the valuable comments provided by the reviewer, which have significantly contributed to improving the quality of our manuscript.

As Reviewer's suggestion, the formation of heterochromatin in young, mid-old, and old cells was observed, and there was a gradual increase in SAHF (**Fig. S9d**). However, the proportional differences between young and mid-old cells were much smaller than those between mid-old and old cells. This suggests that the cellular traits of mid-old cells are more similar to young cells rather than old cells.

Additionally, to further distinguish the cellular traits of young, mid-old, and old cells, we conducted gene ontology (GO) pathway analysis of macromolecules (DNA, mRNA, peptide) metabolism using gene set enrichment analysis (GSEA). As expected, we observed a decrease in gene expression related to DNA and mRNA metabolism in senescent cells compared to young and mid-old cells (**Fig. S10c-d**). This finding aligns with a recent study showing a decline in dNTP levels caused by decreased ribonucleotide reductase (RRM2) activity during senescence (Aird, K.M. et al., Cell Reports, 2013). However, we found no significant differences between young and mid-old cells in this aspect. This suggests that nucleotide metabolism, which is essential for cell proliferation, remains active in mid-old cells, indicating that these cells may retain some proliferative capacity and potential for recovery.

In contrast, we made an intriguing observation that gene expression related to peptide metabolism was highest in mid-old cells (**Fig. S10e**). The underlying mechanism behind this phenomenon is still not fully elucidated by the current study, but it is believed to be another characteristic feature of mid-old cells. These results indirectly suggest that protein synthesis is particularly active in mid-old cells, possibly contributing to mid-old cell-specific protein expression. However, further research is needed to elucidate the mechanisms responsible for the sustained higher protein-related metabolism in mid-old cells.

Related data can be found in **Figures S9d and S10c-e**.

2 Figure 8. The contents of the Figure have been improved, but the robustness of the conclusions is still very limited. For example, treatment with rmSLIT2 results in a reduction of the mouse body weight (Fig. 8a) and this might be due to a beneficial effect associated to increase activity mice in cages, but also it might just be a weight lost due to toxicities of recombinant protein. The title of the Results section – SLIT 2 as a master rejuvenation factor – must be toned down (e.g. SLIT 2 as a potential rejuvenation factor). Also, the authors must recognize the important limitations of Fig. 8 in the Discussion, including for healthspan, rejuvenation and lifespan, as it is done in the rebuttal letter.

In addition, the validation of p21 in Fig. 8d should be improved or some of the panels modified. The IHC signal of functional p21 should be nuclear, similarly to the SOX2 signal nicely presented in Fig 8e. Could the authors please double check this and reanalyze the quantifications, if required? It is important to ensure they are not quantifying an IHC artefact in some of the fields.

Response: We fully agree with the points raised by the reviewer. In the current dataset, we cannot exclude the possibility that weight loss is the result of a toxic effect of rmSLIT2 itself. In rodent studies, two of the most important signs indicating a toxic effect of injected drugs are changes in behavior and weight loss (Patricia et al., J Am Assoc Lab Anim Sci, 2011). In our previous manuscript, we provided

a video showing the activity of mice in the cage, comparing young mice treated with PBS (n=5) and rmSLIT2-treated mice (n=5) (**Movie S4**). Additionally, in this revised version, we have included a graph showing the weight change in young mice treated with PBS and rmSLIT2, and we did not observe weight loss in either group. Therefore, it can be inferred that rmSLIT2 does not induce critical toxicity in mice. We have included these data in **Figure S26** in the revised version of the manuscript.

Additionally, we conducted a repeated experiment for the p21^{Waf1} immunohistochemistry (IHC) staining in **Figure 8d**. In the previous version of the manuscript, we observed some cells without nuclear staining for p21^{Waf1}. To clarify the staining pattern of p21^{Waf1}, we performed IHC using another antibody (2990-1, Epitomics) in the revised version. The results from this repeated experiment showed staining in both the nucleus and cytoplasm. Considering that overexpressed p21^{Waf1} can result in cytoplasmic localization (Zoe et al., Breast Cancer Res, 2003 and Roelof et al., J Clin Invest, 2010), we could suggest that p21^{Waf1} is highly expressed once its expression is promoted. In the revised version of the manuscript, we have included this updated **Figure 8d**, which presents the results quantitatively to further support our findings.

Based on the reviewer's comments, we have also changed the titles of each figure. We also included a discussion on the impact of rmSLIT2 protein on lifespan in the discussion section. Thank you again for your valuable comments.

3. In addition to the intriguing mechanistic/functional differences between fully senescent cells and mid-old cells, this study opens some interesting questions. For example, whether mid-old cells can favor their immunoclearance by secreting inflammatory factors or factors associated to inflammatory processes (e.g. IL1B or SAA1) or rather immunosuppression. Also, how do mid-old cells apply to pathological conditions and diseases (e.g. Are they increased in diseased tissues? Are they part of the tumour microenvironment? Is this a process implemented by cancer cells in response to therapy and resulting in cancer resistance?). The reversibility of the process and potential re-entering to the cell cycle is another question that warrants further studies. Perhaps some of these questions are also food for Discussion and a good opportunity for future works.

I congratulate the authors for the exhaustive revision.

Response: We are deeply grateful for your intriguing suggestions for future research. The heterogeneous nature of senescent cells is a topic that also greatly interests us. We believe there could be additional subtypes, particularly in relation to proliferation capacity. Therefore, the concept of 'mid-old' cells remains somewhat broad, potentially encompassing this heterogeneity. A deeper understanding of this complex population, both *in vitro* and *in vivo*, will require further investigation.

Additionally, we conjecture that there might be functionally specialized subtypes of senescent cells related to immune clearance, immune reaction inhibition, or SASP production. The systematic understanding of this 'senescence ecosystem' poses yet another significant question.

Furthermore, Jeon et al. (Nature Med, 2016) highlighted the negative effects of age-related senescent cells in pathologic conditions, specifically in osteoarthritis. We also propose that age-related senescent cells could have negative implications in various pathological conditions, not just osteoarthritis. In fact, we are currently examining another pathological condition - cancer. It will indeed be intriguing to explore whether the concept of 'mid-old' cells can be applied to cancer cells or other cells within the cancer environment, such as cancer-associated fibroblasts. We have discussed it in the **Discussion** section.

Once again, we sincerely thank you for your valuable insights.

Reviewer #1 (Remarks to the Author):

See attachment.

Reviewer #1 Attachment on the following page

The authors have not satisfactorily addressed major concerns listed under point 1 regarding their analysis of and statements on senescent cells. Please see below for the rationale behind these major concerns. My comments to the authors responses are in italics.

1. As mentioned in the initial review, there is a large body of evidence that p16-positive cells accumulate with aging. Early mouse data by many labs has more recently been confirmed using a p16 reporter mice (Omori et al 2020). I believe that the fact that the authors cannot reproduce firmly established mousedata suggests that their methods are less reliable and more superficial. Indeed, the authors did do the requested co-staining of p16 with p21, but did not do this for all tissues and did not do any quantitation of double positive (senescent) cells. Without such quantitation the p21-p16 double staining is not informative. Furthermore, it is widely accepted that SA-beta-Gal is not a reliable *in vivo* senescence marker for senescence as many non-senescent cells are beta-Gal positive. If the authors remain of the opinion that it is important to state that p16-senescent cells do not increase with aging in tissues and organs, they will have to provide compelling data to justify this statement.

Response: Thank you for the valuable feedback. We apologize for not adequately addressing a pointraised by the Reviewer in the previous version of the paper. In our previous manuscript, we presentedthe single-cell RNA-sequencing data of human and mouse colon. However, we acknowledge that theseapproaches might not have sufficiently addressed the reviewer's concern. To address this issue, we would like to include the following additional data and provide further explanations.

Finding appropriate markers for senescent cells has been a major challenge in senescence research. Although p16^{INK4A} and p21^{Waf1} are commonly used, we also agree that they don't solely reflect the senescence status. p21^{Waf1}, along with p53, is also involved in the early DNA damage response to external stresses. Therefore, p21^{Waf1}-positive cells may include both senescent and damaged cells in the p21^{Waf1}-positive cell population. On the other hand, p16^{INK4A} can give false positives due to viral infections and insufficiently halt cell proliferation, leading to inaccuracies in senescent cell analysis. Similarly, another widely-used senescence marker SA-β-Gal can also show false positives based on acid-base conditions and certain cell types. Despite these limitations, we believe that p16^{INK4A} remainscurrently the most reliable marker for detecting the later stages of cellular senescence *in vivo*. Recenttransgenic mouse models, such as p16^{INK4A}-tdTomato reporter or the p16^{INK4A}-3MR system, are basedon the *CDKN2A* gene rather than other senescence-related genes to address these challenges.

In this paper, we are not trying to deny the broadly accepted concept that the total number of p16^{INK4A}-positive

cells increases in both human and aged mice tissue. Several studies have noted an increase in the number of p16^{INK4A}-positive cells in older mice models (Baker et al., Nature, 2011, Liu et al., PNAS, 2019, and Omori et al., Cell Metab, 2020). This aligns with reports indicating a rise in these cells in aged tissues. Hence, we concur that there is indeed an increase in the number of p16^{INK4A}-positive cells in aged tissues. However, despite Omori et al. (Cell Metab, 2020) showing an overall increase in tdTomato-positive cells across various organs using a p16^{INK4A}-tdTomato reporter mouse, they did not specify the exact cell types that constituted the tdTomato-positive cells.

Additionally, although Grosse et al., using a p16-CRE/R26-mTmG mouse, revealed that CD31-positive endothelial cells and F4/80-positive macrophages comprise the major population of p16^{INK4A}-positive cells in the liver, information about the remaining population and other organs is lacking.

Based on these previous studies, we cautiously propose that the majority of the p16^{INK4A}-positive cells originate from circulatory systems, such as immune cells or vascular endothelial cells.

Several papers using the reporter mice have been published which include specifications of the Tomato+ p16+ cell types. These cell types that were positive included epithelial cells and smooth muscle cells. See for instance Omori 2020.

In fact, the emphasis of our work is on the specific cell types within senescent ecosystem. We have primarily focused on epithelial cells, fibroblasts, and smooth muscle cells, which are the major component cells of the organ rather than infiltrated immune cells. To address this issue, we performed further in-depth analysis of single-cell RNA sequencing data (GSE132042) from mouse tissues, as reported in Tabula Muris Consortium et al. (Nature, 2018). p16^{INK4A} expression across various tissues revealed that the changes in the number of p16^{INK4A}-positive cells, as observed with aging, are dependent on the organ type. When analyzing p16^{INK4A} expression in various tissues, we found that certain tissues (liver, tongue, heart, pancreas) showed an increase in p16^{INK4A} expression, while some others (aorta, bladder) showed a decrease. Furthermore, in most tissues, only minimal changes in p16^{INK4A} expression were identified. To gain further insights, we delved into specific cell types within representative tissues, such as lung, colon, skin, and liver, to assess their p16^{INK4A} expression levels. The results indicated a notable increase in p16^{INK4A} expression primarily in immune cells. On the contrary, the cells we primarily focused on, epithelial cells, fibroblasts, and smooth muscle cells, showed either no difference or only a slight increase in p16^{INK4A} expression. We added these data in **Figure S8 a-e**.

The problem with the use of single-cell RNA sequencing is that this approach - in general - has rather poor transcript coverage, which is particularly problematic when using it to study transcripts of low abundance, such

as p16 transcripts. It is well known that absolute p16 transcript levels are low, even in senescent cells, and that the relatively long half-life of rare p16 transcripts positively impacts protein expression.

To validate these findings in human *in vivo* aged tissues, we attempted to perform single-cell sequencing analysis in human colon, lung, skin, and liver by searching for publicly available datasets. However, unfortunately, we could not find the enough number of patients in both young and old aged tissue samples. Nonetheless, we conducted an additional analysis of the normal regions of single cells in lung, colon, skin, and liver tissues with the limited number of patients. Although they did not align with the initially proposed criteria of old and young groups in the manuscript, we analyzed the lung tissue (51 vs. 75 years old), skin tissue (25 and 27 vs. 53, 69, and 70 years old), colon tissue (35, 35 and 42 vs. 81,82 and 90 years old), and liver tissue (48 and 48 vs. 64 and 65 years old). We observed that p16^{INK4A}-expressing cells were predominantly found in immune cells rather than in epithelial cells, fibroblasts, or smooth muscle cells in the aged tissue.

Apart from the problems with the analysis mentioned by the authors themselves, the added concern that single-cell RNA sequencing is not a solid method to detect p16-positive (senescent) cells makes these human data results virtually uninterpretable.

In the case of lung data, datasets that include younger samples are scarce due to the late onset of the lung-related diseases. Nonetheless, we managed to find and analyze one dataset that compared samples from individuals aged 51 and 75 years old. The results showed a slight increase in p16^{INK4A}-positive cells (from 0.6% to 0.7%). However, mirroring our mouse findings, these p16^{INK4A}-positive cells were predominantly observed in immune cells, whereas smooth muscle cells and fibroblasts did not express p16^{INK4A}. We also investigated skin and liver tissues. In the skin tissue, when comparing samples from younger individuals (25 and 27 years old) to older individuals (53, 69, and 70 years old), there was a modest increase of approximately 0.1% (from 0.6% to 0.7%) in p16^{INK4A}-expressing cells. The majority of these cells were also identified as immune cells. Regarding liver tissue, our analysis of samples from individuals aged 48 (two individuals) and 64 and 65 years old revealed an increase in p16^{INK4A}-expressing cells, predominantly observed in Kupffer cells. This aligns with the earlier study by Grosse et al., who used p16^{INK4A} reporter mouse models and found that hepatic F4/80-positive macrophage often exhibit signs of p16^{INK4A}-high phenotype in the livers of aged mice. As demonstrated in the previous version of the manuscript, the colon single-cell data showed that p16^{INK4A} expression is primarily seen in CD3-positive T cells.

Although our analysis was not extensive across a broad range of organs, we have cautiously drawn the following conclusion: with aging, there is an increase in p16^{INK4A}-expressing cells, but this increase is predominantly observed in immune and endothelial cells. Minimal changes were seen in epithelial cells, fibroblasts and smooth

muscle cells. We've included relevant data in **Figures S4, S5, S6, S7, and S8** and further discussed these findings in the discussion section.

Moreover, according to the Reviewers' suggestions, we conducted immunofluorescence staining of p16^{INK4A} and p21^{Waf1} with vimentin in aged tissues. The analysis was performed in colon, liver, small intestine, skin, and lung tissues. Unfortunately, we could not analyze thyroid tissue due to the limited presence of stromal fibroblasts for examination. Triple immunofluorescence analysis [p16^{INK4A} (blue), vimentin (green), and p21^{Waf1} (red)] showed that although there were few p16^{INK4A} positive cells present in old-aged stromal tissues, almost p16^{INK4A}/vimentin positive cells were also positive for p21^{Waf1}, but not all p21^{Waf1}-positive cells exhibited p16^{INK4A} expression. These data suggest that fibroblasts in aged tissues show a mid-old cell status characterized by p21^{Waf1} expression rather than fully senescent cells. Relevant data has been included in **Figure S12b**.

Going back to the original point, the authors claim that there is no increase in p16-positive senescent cells and based this on the use of p16 as a marker for senescence. To identify p16-positive cells as senescent, the standard in the field is to show that p16-positive cells co-express other marker of senescence. One of these markers is p21, but there are others if one wanted to do a thorough characterization. In response the authors conducted a co-staining for p16 and p21. However, they failed to quantify the p16-p21 double positive cells, leaving us still without any evidence for their conclusion that p16-positive senescent cells do not increase with aging. This was pointed out in the re-review of the paper and the authors in response performed quantitation of p16-p21 positive fibroblast-like (vimentin+) cells in various aged tissues. However, the necessary comparison of p16-p21 positive cells in young versus old tissue was not included, which is a necessity to be able to draw the conclusion that there is not an age-related increase in senescent cells.

This together with the abovementioned major concerns regarding the identification of fully senescent cells in this paper, leads me to conclude that publication of this aspect of the study is not warranted. Also, the authors' narrowing the analysis in subsequent revisions - by focusing on fibroblasts only and not including epithelial and VSMC - is a major concern.

Again, as I mentioned before: "If the authors remain of the opinion that it is important to state that p16-senescent cells do not increase with aging in tissues and organs, they will have to provide compelling data to justify this statement."

If the authors could concentrate the manuscript on what I perceive as the most interesting and novel findings related to mid-old cells and remove all the premature and/or non-insightful studies on senescent cells that would be a major step forward.

Other points:

Abstract:

“However, the present study identified that fibroblasts and smooth muscle cells which are the major constituents of organ stroma were neither proliferative nor fully senescent in tissues of the elderly, which we termed “mid-old status” cells. “

“However, the present study identified a subset of fibroblasts and smooth muscle cells which are the major constituents of organ stroma were neither proliferative nor fully senescent in tissues of the elderly, which we termed “mid-old status” cells.

“We provided the functional reverse of mid-old cells rather than elimination of senescent cells as a new concept about anti-aging”.

“Our data identify functional reversion of mid-old cells as a novel experimental towards preventing or ameliorating aspects of aging-related tissue dysfunction”.

Reviewer #2 (Remarks to the Author):

The authors have addressed now all my points. The revised manuscript has been significantly improved throughout the rounds of revision and the conclusions are supported by the presented data. I congratulate the authors for the submitted work.

Answer to reviewers

To the reviewer 1

We appreciate your re-consideration of our paper despite the various points of contention. In accordance with the reviewer's recommendation, we have removed all the content related to p16^{INK4A}-positive cells that is still not entirely conclusive from the manuscript. We have restructured the paper to focus on the presence and characterization of mid-old cells. We are grateful for your valuable suggestions once again and we hope that our revised manuscript will meet your approval.

1. Several papers using the reporter mice have been published which include specifications of the Tomato+ p16+ cell types. These cell types that were positive included epithelial cells and smooth muscle cells. See for instance Omori 2020.

Response: We appreciate the reviewer's detailed comments. We have thoroughly reviewed the paper published by *Omori et al. in Cell Metabolism in 2020*. Omori et al. observed an increase in p16^{INK4A}-positive cells in the liver, skin, kidney, and lung of 12-month-old mice. They used single-cell RNA-seq to identify which cell types express p16^{INK4A} among the constituent organs. Although we did not analyze the kidney in previous version of manuscript, Omori et al. observed an increase in p16^{INK4A}-positive cells in kidney tubular cells and endothelial cells.

As the reviewer pointed out, our data did not cover all tissues, and there is still controversial content included. Therefore, we have removed the analysis of p16^{INK4A}-positive cells in elderly tissues from our paper (Previous version manuscript Figure 1, S1, S2, and S3).

2. The problem with the use of single-cell RNA sequencing is that this approach - in general – has rather poor transcript coverage, which is particularly problematic when using it to study transcripts of low abundance, such as p16 transcripts. It is well known that absolute p16 transcript levels are low, even in senescent cells, and that the relatively long half-life of rare p16 transcripts positively impacts protein expression.

Apart from the problems with the analysis mentioned by the authors themselves, the added concern that single-cell RNA sequencing is not a solid method to detect p16-positive (senescent) cells makes these human data results virtually uninterpretable.

Response: We completely agree with the reviewer's comment that the single-cell RNA-Seq lacks the necessary depth to detect p16^{INK4A} positive cells in elderly human tissues and aged mice. Consequently, we have removed the single cell RNA-seq data and related supplementary figure from the manuscript (Previous version manuscript Figure S4~S8).

4. Going back to the original point, the authors claim that there is no increase in p16-positive senescent cells and based this on the use of p16 as a marker for senescence. To identify p16-positive cells as senescent, the standard in the field is to show that p16-positive cells co-express other marker of senescence. One of these markers is p21, but there are others if one wanted to do a thorough characterization. In response the authors conducted a co-staining for p16 and p21. However, they failed to quantitate the p16-p21 double positive cells, leaving us still without any evidence for their conclusion that p16-positive senescent cells do not increase with aging. This was pointed out in the re-review of the paper and the authors in response performed quantitation of p16-p21 positive fibroblast-like (vimentin+) cells in various aged tissues. However, the necessary comparison of p16-p21 positive cells in young versus old tissue was not included, which is a necessity to be able to draw the conclusion that there is not an age-related increase in senescent cells.

This together with the abovementioned major concerns regarding the identification of fully senescent cells in this paper, leads me to conclude that publication of this aspect of the study is not warranted. Also, the authors' narrowing the analysis in subsequent revisions - by focusing on fibroblasts only and not including epithelial and VSMC - is a major concern.

Again, as I mentioned before: "If the authors remain of the opinion that it is important to state that p16-senescent cells do not increase with aging in tissues and organs, they will have to provide compelling data to justify this statement."

If the authors could concentrate the manuscript on what I perceive as the most interesting and novel findings related to mid-old cells and remove the all the premature and/or non-insightful studies on senescent cells that would be a major step forward.

Response: We appreciate reviewer's the constructive feedback. The main concern raised by reviewer 1 was the validity and reliability of our analysis of senescent cells based on p16^{INK4A} expression. We acknowledge that this analysis was not sufficiently rigorous and conclusive, and that there are several caveats and limitations associated with the use of p16^{INK4A} as a marker of senescence. Therefore, we have decided to remove this part (the section of the manuscript containing data indicating that the proportion of p16^{INK4A}-positive senescent cells does not significantly increase with age compared to young individuals) of our manuscript entirely, and focus on the characterization of mid-old cells, which are a novel and distinct cell population that we have identified in elderly human tissues. We removed the Figure 1 and the related Figures S1~S8 and S12b. We have restructured the paper to focus on the presence of mid-old cells.

As a result, we have made corrections to the introduction and discussion sections. Additionally, we have included new data in **Figure S4** that shows the absence of mid-old cell markers expression in colon and lung epithelial cells.

Regarding the mid-old cell-related gene expression in vascular smooth muscle cells (VSMCs), as it was noted that it was expressed in *in vivo* elderly tissue, we conducted experiments by culturing and generating mid-old status cells using primary pulmonary artery smooth muscle cells (PASM). When comparing doubling time 2 (DT2) and DT6, we observed that mid-old cell markers expression in DT6 were increased similarly to mid-old fibroblasts. We have included relevant data in **Figure S5**.

Thank you again for the thorough review.

Other points:

Abstract:

“However, the present study identified a subset of fibroblasts and smooth muscle cells which are the major constituents of organ stroma were neither proliferative nor fully senescent in tissues of the elderly, which we termed “mid-old status” cells.

“We provided the functional reverse of mid-old cells rather than elimination of senescent cells as a new concept about anti-aging”.

“Our data identify functional reversion of mid-old cells as a novel experimental towards preventing or ameliorating aspects of aging-related tissue dysfunction”.

Response: As pointed out by the reviewer, we have made the corrections in Abstract.

To the reviewer 2

The authors have addressed now all my points. The revised manuscript has been significantly improved throughout the round of revision and the conclusion are supported by the presented data. I congratulate the authors for the submitted work.

Response: We are grateful for the reviewer's meticulous review. By addressing the points raised, the completeness of the paper has been enhanced. Once again, we express our sincere gratitude.

Reviewer #1 (Remarks to the Author):

I commend the authors for deciding to focus on the discovery and characterization of mid-old cells. I have no further reservations regarding the publication of the authors' very interesting and innovative study.

Minor point:

Great modified title. The abstract also has been appropriately modified and reads very well but a word seems to be lacking in the last sentence: "...novel experimental (?approach?) towards "...